# PLANETALIGN: A COMPREHENSIVE PYTHON LIBRARY FOR BENCHMARKING NETWORK ALIGNMENT

**Qi Yu**[1], **Zhichen Zeng**[1], **Yuchen Yan**[1], **Zhining Liu**[1], **Baoyu Jing**[1], **Ruizhong Qiu**[1]
**Ariful Azad**[2], **Hanghang Tong**[1]
[1]University of Illinois Urbana-Champaign, [2] Texas A&M University
[1]{qiyu6,zhichenz,yucheny5,liu326,baoyuj2,rq5,htong}@illinois.edu
[2]ariful@tamu.edu

## ABSTRACT

Network alignment (NA) aims to identify node correspondence across different networks and serves as a critical cornerstone behind various downstream multi-network learning tasks. Despite growing research in NA, there lacks a comprehensive library that facilitates the systematic development and benchmarking of NA methods. In this work, we introduce PLANETALIGN, a comprehensive Python library for network alignment that features a rich collection of built-in datasets, methods, and evaluation pipelines with easy-to-use APIs. Specifically, PLANETALIGN integrates 18 datasets and 14 NA methods with extensible APIs for easy use and development of NA methods. Our standardized evaluation pipeline encompasses a wide range of metrics, enabling a systematic assessment of the effectiveness, scalability, and robustness of NA methods. Through extensive comparative studies, we reveal practical insights into the strengths and limitations of existing NA methods. We hope that PLANETALIGN can foster a deeper understanding of the NA problem and facilitate the development and benchmarking of more effective, scalable, and robust methods in the future. The source code of PLANETALIGN is available at https://github.com/yq-leo/PlanetAlign.

## 1 INTRODUCTION

Multi-sourced and multi-layer networks are becoming ubiquitous across a wide range of domains in the era of big data and AI (Lin et al., 2024; Wei et al., 2026), ranging from social network analysis (Shao et al., 2023; Qiu et al., 2023; Liu et al., 2024d; Rácz & Zhang, 2024; Peng et al., 2025), anti-money laundering (Zhang et al., 2021; Bei et al., 2023), bio-informatics (Zeng et al., 2023b; Hu et al., 2024; Zare Mirak-Abad & Ghorbanali, 2025; Zeng et al., 2024b), to knowledge graph fusion (Yan et al., 2021a; Chen et al., 2024b; Liu et al., 2025; Ai et al., 2025), infrastructure network modeling (Qiu et al., 2022; Wang et al., 2023; Xu et al., 2024a; Zou et al., 2025b; Lin et al., 2025b;a; He et al., 2026; Lin et al., 2026), and unstructured adaptation (Xu et al., 2024b; Bao et al., 2024; Liu et al., 2024c; Wu et al., 2024; Zou et al., 2025a; Li et al., 2025; Bao et al., 2025; Zeng et al., 2025c;d; 2026). Identifying the same node across different networks, i.e., network alignment (NA), enables joint learning across multiple networks and serves as the key cornerstone of multi-network tasks. For example, aligning users across online social networks improves personalized services, e.g., cross-domain recommendation (Liu et al., 2023a; Yu et al., 2025; Zeng et al., 2025a;b; Yoo et al., 2025a;b). Aligning suspicious accounts from different transaction networks facilitates the detection of fraudulent activity (Zhang et al., 2019b; Du et al., 2021; Yan et al., 2024b;a;c; Qiu et al., 2024; Chen et al., 2026). In protein interaction networks, alignment of proteins across different species uncovers hidden biological homologies (Clark & Kalita, 2014; Hu et al., 2024). In knowledge graphs (KGs), merging incomplete KGs based on aligned entities helps construct unified knowledge bases (Yan et al., 2021a; Liu et al., 2023b; 2024a; Chen et al., 2024b).

Despite growing interest in NA, there lacks a comprehensive benchmark to provide standardized evaluation of NA methods on different datasets from various aspects. The absence of such benchmarks leaves the genuine performance and usefulness of existing NA methods an open research question, hindering the standardization of research in the NA community. Although prior efforts, which are

summarized in Table 1, have been made in benchmarking NA methods (Clark & Kalita, 2014; Cao & Yu, 2016; Sun et al., 2020; Trung et al., 2020; Döpmann, 2013), they suffer from at least one of the following limitations: (1) *limited datasets* within a single domain, e.g. biological networks (Clark & Kalita, 2014) or social networks (Cao & Yu, 2016); (2) *limited methods* exclusively focusing on a single category, e.g., consistency-based methods (Döpmann, 2013) or embedding-based methods (Sun et al., 2020), while ignoring the most recent line of works, e.g., optimal transport (OT) based methods; (3) *limited and inconsistent evaluation* from a single aspect, e.g. effectiveness (Clark & Kalita, 2014; Cao & Yu, 2016), without standardized dataset splits and evaluation metrics (Clark & Kalita, 2014; Cao & Yu, 2016; Sun et al., 2020; Trung et al., 2020; Döpmann, 2013).

In response to these limitations, we introduce PLANETALIGN, an open-source PyTorch-based library designed for unified evaluation and streamlined development of NA methods, which features the following key design. **Firstly**, PLANETALIGN includes 18 different public datasets spanning 6 different domains which can be directly downloaded through a simple API call, including social networks (Zhang & Philip, 2015; Zhang & Tong, 2016), publication networks (Tang et al., 2008; Yang et al., 2016; Leskovec et al., 2007), biological networks (Stark et al., 2006; De Domenico et al., 2015b; Zitnik & Leskovec, 2017; Park et al., 2010), knowledge graphs (Sun et al., 2017), infrastructure networks (Yan et al., 2022; Zhu et al., 2021; Song et al., 2020), and communication networks (Zhang et al., 2017; Kunegis, 2013), covering both real-world and synthetic scenarios (*Limitation #1*). The wide range of datasets built into PLANETALIGN allows comprehensive evaluation of NA methods on different types of networks, e.g., plain and attributed networks, fostering in-depth understanding of the applicability of NA methods to different domains. **Secondly**, PLANETALIGN features efficient implementations of 14 different NA methods including consistency-based (Singh et al., 2008; Zhang & Tong, 2016), embedding-based (Liu et al., 2016; Heimann et al., 2018; Chu et al., 2019; Zhang et al., 2020; 2021; Gao et al., 2021; Yan et al., 2021b; Liu et al., 2023a), and OT-based methods (Zeng et al., 2023a; 2024a; Tang et al., 2023; Yu et al., 2025), covering traditional and state-of-the-art baselines (*Limitation #2*). With easy-to-use APIs, PLANETALIGN allows streamlined comparison between NA methods across diverse benchmark settings. **Thirdly**, PLANETALIGN highlights a comprehensive list of evaluation metrics and benchmarking tools (*Limitation #3*). For evaluation metrics, we include the most classical effectiveness metrics, Hits@K and MRR, under different pairwise alignment settings. We also include time and memory overheads for evaluating the efficiency and scalability of NA methods. For benchmarking tools, we enforce consistent dataset split through a unified API design to ensure reproducibility. PLANETALIGN also provides a rich collection of APIs and utility functions that allows fair and reproducible benchmarking across key dimensions of NA performance. **Finally**, PLANETALIGN designs extensible APIs and implements efficient utility functions, allowing users to streamline the implementation of customized NA methods and the integration of customized datasets with minimal efforts. Specifically, our API design allows customized datasets and NA methods to be built upon carefully designed base classes and integrated into PLANETALIGN's pipeline with only a few lines of code. PLANETALIGN further provides commonly used utility functions for NA algorithms, such as random walk with restart (RWR) embedding, anchor-based embedding, etc. Empowered by the aforementioned features, PLANETALIGN addresses the limitations of existing NA benchmarks comprehensively.

Based on PLANETALIGN, we conduct comprehensive experiments to evaluate the effectiveness, scalability, robustness, and sensitivity to supervision of 14 built-in NA methods across 18 built-in datasets, revealing both theoretical and practical insights into the strength and limitations of existing NA methods. We also compare PLANETALIGN's implementation of NA algorithms with their official implementation, which shows that our implementations achieve up to 3 times speed-up while maintaining similar effectiveness performance, demonstrating the superiority of PLANETALIGN.

In summary, we introduce a unified, comprehensive, and efficient library PLANETALIGN featuring a wide range of built-in datasets and NA methods, as well as extensible and easy-to-use utility functions and APIs, facilitating the benchmarking and development of NA methods.

## 2  PROBLEM DEFINITION

An illustration of NA problems are shown in Figure 1. Given two input networks $\mathcal{G}_1 = \{\mathcal{V}_1, \mathbf{A}_1, \mathbf{X}_1, \mathbf{E}_1\}$, $\mathcal{G}_2 = \{\mathcal{V}_2, \mathbf{A}_2, \mathbf{X}_2, \mathbf{E}_2\}$ and a set of anchor node pairs $\mathcal{L} = \{(x, y) | x \in \mathcal{V}_1, y \in \mathcal{V}_2\}$ indicating pre-alignment, where $\mathcal{V}_1, \mathcal{V}_2$ denote the node sets, $\mathbf{A}_1, \mathbf{A}_2$ denote

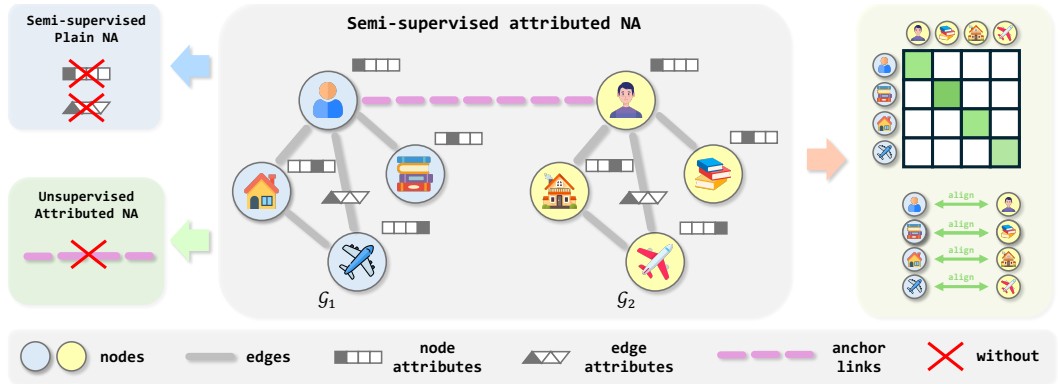

Figure 1: An illustration of NA problems.

the graph [1] adjacency matrices, $\mathbf{X}_1, \mathbf{X}_2$ denote the node attribute matrices, and $\mathbf{E}_1, \mathbf{E}_2$ denote the edge attribute matrices, the *semi-supervised attributed network alignment* task aims to discover node-level correspondence across two networks inferred from an output alignment matrix $\mathbf{S}$, where $\mathbf{S}(x, y)$ indicates the likelihood of alignment between node $x \in \mathcal{V}_1$ and node $y \in \mathcal{V}_2$. If neither node attributes $\mathbf{X}_1, \mathbf{X}_2$ nor edge attributes $\mathbf{E}_1, \mathbf{E}_2$ are available, this becomes the *semi-supervised plain network alignment* task; If no anchor node pairs are available, i.e., $|\mathcal{L}| = 0$, this becomes the *unsupervised attributed network alignment* task.

## 3 RELATED WORK

### 3.1 NETWORK ALIGNMENT METHODS

Existing NA methods can be classified into three categories: consistency-based, embedding-based, and OT-based approaches (Zhang & Tong, 2020). Consistency-based methods are among the earliest approaches, formulated as optimization problems which assume structural and/or attribute consistency between node neighborhoods across networks (Singh et al., 2008; Bayati et al., 2009; Zhang & Tong, 2016; Zhang et al., 2019a). Although recent works on NA have largely moved beyond consistency principles, consistency-based methods remain important baselines for benchmarking purposes.

Embedding-based and OT-based methods represent more recent advances in the NA community. For embedding-based methods, nodes are mapped into a shared low-dimensional space and aligned based on embedding similarity (Liu et al., 2016; Heimann et al., 2018; Chu et al., 2019; Zhang et al., 2020; Gao et al., 2021; Yan et al., 2021b; Zhang et al., 2021; Liu et al., 2023a; Zheng et al., 2025). By leveraging advances in deep representation learning, embedding-based methods have shown strong performance and remain an active research direction. For OT-based methods, they formulate the NA problem as an optimization problem minimizing the total effort of transporting the node distribution of one graph to another under a set of pre-defined or learnable cost functions (Tang et al., 2023; Zeng et al., 2023a; 2024a; Yu et al., 2025). Recent OT-based methods consistently achieve state-of-the-art performance across these three categories of NA methods, making them a promising direction for future research. PLANETALIGN includes representative methods from all three kinds of methods, providing a comprehensive benchmarking library.

### 3.2 NETWORK ALIGNMENT LIBRARIES

There are five existing benchmarks/libraries for NA, and we include a comprehensive comparison based on the inclusion of datasets, NA methods, and evaluation dimensions, in Table 1. Specifically, SGAPBSA (Döpmann, 2013) and CAPABN (Clark & Kalita, 2014) mainly focus on benchmarking traditional consistency-based NA methods on biological networks. ASNets (Cao & Yu, 2016) benchmarks the effectiveness of both consistency-based and embedding-based methods on social networks, but leaves the scalability and robustness of these methods an open research question.

---

[1] In this work, the terms 'network' and 'graph' are used interchangeably.

Table 1: Comparison with existing NA benchmarks/libraries (Clark & Kalita, 2014; Cao & Yu, 2016; Sun et al., 2020; Trung et al., 2020; Döpmann, 2013). We denote whether a specific type of networks, methods, and evaluations is included in the benchmark/library.

| Benchmark/Library | | SGAPBSA | CAPABN | ASNets | NAB | OpenEA | PLANETALIGN (ours) |
|---|---|---|---|---|---|---|---|
| **Networks** | Social | ✗ | ✗ | ✓ | ✓ | ✗ | ✓ |
| | Communication | ✗ | ✗ | ✗ | ✗ | ✗ | ✓ |
| | Publication | ✗ | ✗ | ✗ | ✗ | ✗ | ✓ |
| | Biological | ✓ | ✓ | ✗ | ✗ | ✗ | ✓ |
| | Knowledge | ✗ | ✗ | ✗ | ✗ | ✓ | ✓ |
| | Infrastructure | ✗ | ✗ | ✗ | ✗ | ✗ | ✓ |
| **Methods** | Consistency-based | ✓ | ✓ | ✓ | ✓ | ✗ | ✓ |
| | Embedding-based | ✗ | ✗ | ✓ | ✓ | ✓ | ✓ |
| | OT-based | ✗ | ✗ | ✗ | ✗ | ✗ | ✓ |
| **Evaluations** | Effectiveness | ✓ | ✓ | ✓ | ✓ | ✓ | ✓ |
| | Scalability | ✓ | ✗ | ✗ | ✓ | ✓ | ✓ |
| | Robustness | ✗ | ✗ | ✗ | ✓ | ✗ | ✓ |

NAB (Trung et al., 2020) comprehensively evaluates the effectiveness, scalability, and robustness of both consistency-based and embedding-based methods. However, NAB only includes social networks, lacking comprehensive datasets on other domains where NA also plays an important part. OpenEA (Sun et al., 2020) focuses on benchmarking embedding-based methods on knowledge graphs, ignoring networks in other domains. In addition, none of the existing NA libraries includes *OT-based* methods, which have emerged as the most recent and effective line of work in the NA community.

## 4 DESIGN OF PLANETALIGN

In this section, we introduce the design features of PLANETALIGN, which includes comprehensive built-in datasets and NA methods (Section 4.1), unified and easy-to-use APIs (Section 4.2), as well as standardized and diverse benchmarking tools (Section 4.3).

### 4.1 COMPREHENSIVE DATASETS AND METHODS

PLANETALIGN collects and curates 18 NA datasets across 6 different domains, covering social networks, publication networks, biological networks, knowledge graphs, infrastructure networks, and communication networks. PLANETALIGN also implements 14 existing NA methods across all 3 categories, including consistency-based, embedding-based, and OT-based methods. An overview of built-in datasets and NA methods in PLANETALIGN is summarized in Figure 2.

**Dataset Collection and Synthesis.** We collect 11 real-world datasets from existing NA works and synthesize 7 additional datasets across 6 distinct domains. We follow the most classical method to synthesize NA datasets from a single network, where we insert 10% noisy edges into and delete 15% existing edges from the original network to create two permuted networks (Yang et al., 2016; Zhang et al., 2020; Yan et al., 2021b; Zhang et al., 2021; Zeng et al., 2023a; Yu et al., 2025).

Specifically, for *social networks*, where NA is used to align the same user for personalized recommendation (Zhang & Philip, 2015; Cao & Yu, 2016; Liu et al., 2016; Chen et al., 2024a), PLANETALIGN includes 4 real-world datasets: Foursquare-Twitter (Zhang & Philip, 2015), Douban (Zhang & Tong, 2016), Flickr-LastFM (Zhang & Tong, 2016), and Flickr-MySpace (Zhang & Tong, 2016); for *publication networks*, where NA is used for author disambiguation (Li et al., 2021), PLANETALIGN includes the most representative real-world dataset ACM-DBLP (Tang et al., 2008), and synthesizes 2 additional datasets from Cora (Yang et al., 2016) and ArXiv (Leskovec et al., 2007); for *biological networks*, where NA uncovers hidden biological homologies by aligning proteins of different species (Singh et al., 2008; Clark & Kalita, 2014; Faisal et al., 2015), PLANETALIGN includes 1 real-world dataset SacchCere (Stark et al., 2006; De Domenico et al., 2015b) and 2 synthetical datasets PPI (Zitnik & Leskovec, 2017) and GGI (Park et al., 2010). For *knowledge graphs*, where NA is used for knowledge fusion (Sun et al., 2020; Liu et al., 2023b; Chen et al., 2023), PLANETALIGN includes 3 variants of a real-world dataset DBP15K (Sun et al., 2017), namely DBP15K ZH-EN, JA-EN, and FR-EN. For *infrastructure networks*, where NA plays an important role in cross layer

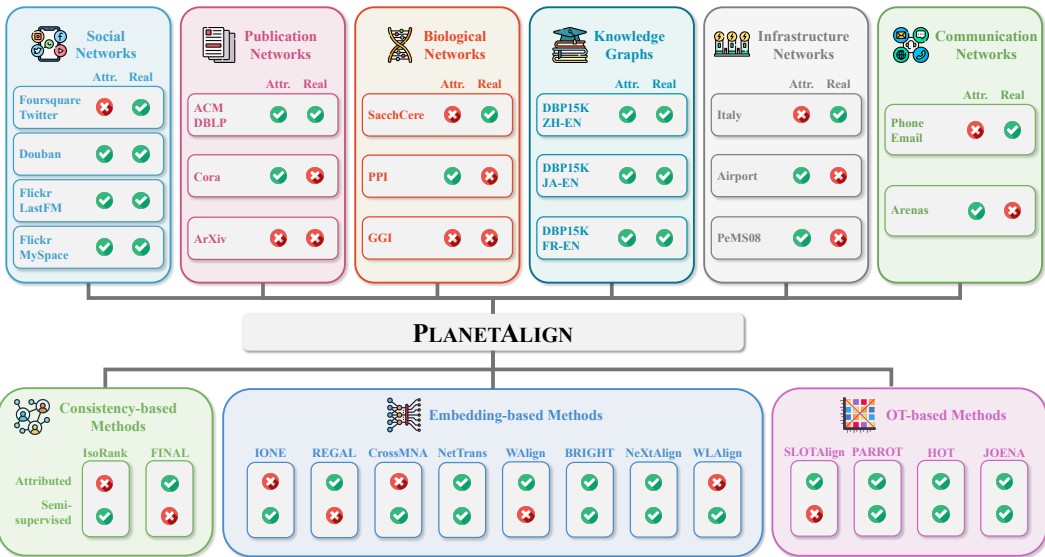

Figure 2: **An overview of built-in datasets and NA methods in PLANETALIGN.** For built-in *datasets*, we indicate if they consist of attributed (Attr.) or plain networks, and if they are real-world (Real) or synthetic datasets. For built-in *NA methods*, we indicate if they are designed for attributed or plain NA tasks, and if they are semi-supervised or unsupervised methods.

dependency inference (Yan et al., 2022), PLANETALIGN includes 1 real-world dataset Italy (Yan et al., 2022), and 2 synthetic datasets Airport (Zhu et al., 2021) and PeMS08 (Song et al., 2020). For *communication networks*, PLANETALIGN includes 1 real-world dataset Phone-Email (Zhang et al., 2017) and 1 synthetic dataset Arenas (Kunegis, 2013). Detailed dataset statistics can be found in Appendix A.

**Baseline Implementations.** We implement 14 existing NA methods based on a unified API, including 2 representative consistency-based methods, 8 embedding-based methods, and 4 OT-based methods. Specifically, for *consistency-based* methods, PLANETALIGN includes IsoRank (Singh et al., 2008) and FINAL (Zhang & Tong, 2016); for *embedding-based* methods, PLANETALIGN includes IONE (Liu et al., 2016), REGAL (Heimann et al., 2018), CrossMNA (Chu et al., 2019), NetTrans (Zhang et al., 2020), WAlign (Gao et al., 2021), BRIGHT (Yan et al., 2021b), NeXtAlign (Zhang et al., 2021), and WLAlign (Liu et al., 2023a); for *OT-based* methods, PLANETALIGN includes SLOTAlign (Tang et al., 2023), PARROT (Zeng et al., 2023a), HOT (Zeng et al., 2024a), and JOENA (Yu et al., 2025). Detailed introductions of built-in NA methods can be found in Appendix B.

## 4.2 UNIFIED AND EASY-TO-USE APIs

PLANETALIGN is carefully designed to provide unified and easy-to-use APIs to streamline the implementation, training, and evaluation of NA algorithms on customizable datasets. An example usage of PLANETALIGN is shown in Figure 3. We also provide detailed documentation at https://planetalign.readthedocs.io/en/latest/, covering both quick-start tutorials and in-depth documentations of API usage of the major components of PLANETALIGN.

Specifically, to train and evaluate an NA algorithm on a specific dataset, the user of PLANETALIGN will first define a `PlanetAlign.data.BaseData` object and a `Model` object inherited from the base class `PlanetAlign.algorithm.BaseModel`. For built-in datasets, PLANETALIGN provides downloading options and reproducible train/test split with a customized training ratio; for built-in algorithms, PLANETALIGN provides hyperparameter options upon definition of the algorithm, and a unified API as PyTorch for GPU/CPU offloading. Both built-in dataset and algorithm objects can be defined neatly in a single line of code. PLANETALIGN also provides unified and intuitive base classes for defining customized datasets and algorithms, as shown in Figure 3.

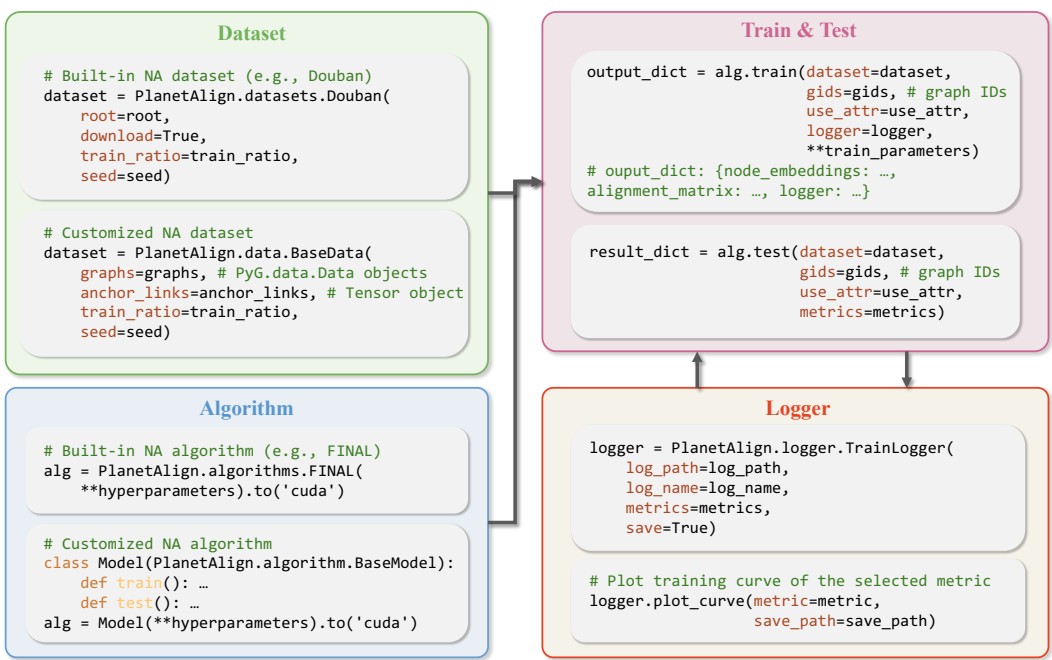

Figure 3: Example usage of PLANETALIGN for benchmarking NA. Users begin by initializing a dataset and algorithm objects, along with a logger for training-time monitoring and visualization. The training and evaluation can then be performed through simple API calls with user-defined parameters, providing substantial flexibility in controlling the training and evaluation workflows.

Before training an NA algorithm, the user has an option to initialize a `PlanetAlign.Logger` object used to log the training process of the algorithm. Then, the user can simply call the `.train` method of the algorithm object with the dataset and logger object, IDs of graphs to be aligned, and additional configuration of training, e.g., training epochs, learning rate, etc., to start the training. Training outputs, including node embeddings, alignment matrix, and training performance are returned by the `.train` after the training process ends, providing fine-grained intermediate results of alignment that users can readily leverage for downstream tasks, such as cross-layer dependency inference (Yan et al., 2022), knowledge integration (Yan et al., 2021a), and cross-KG modality fusion (Chen et al., 2023).

Finally, after the training process, the user can call the `.test` method of the algorithm object with customized options of evaluation metrics. The optional logger object, which records and logs comprehensive data during training, also provides a rich collection of APIs for visualizing the evolution of different metrics along training, e.g., training loss, time and memory usage, etc.

### 4.3 STANDARDIZED AND DIVERSE BENCHMARKING TOOLS

**Standard Evaluation Metrics.** PLANETALIGN provides low-level utility functions for computing standard and widely adopted evaluation metrics in the NA tasks, with custom options for alignment directions, such as left-to-right for pairwise alignment scenarios where the nodes in $\mathcal{G}_1$ is aligned to $\mathcal{G}_2$, and vise versa. Specifically, PLANETALIGN includes the following metrics:

- **Hits@K**. In the case of aligning $\mathcal{G}_1$ to $\mathcal{G}_2$, Hits@K refers to the proportion of nodes in $\mathcal{G}_1$ whose correct alignment in $\mathcal{G}_2$ is ranked within the top-$K$ candidates by a NA algorithm. Formally,

$$\text{Hits@K} = \frac{1}{N} \sum_{i=1}^{N} \mathbb{1}\{\text{rank}_i \leq K\},$$

where $N$ is the number of nodes in $\mathcal{G}_1$, $\text{rank}_i$ is the rank of the correct alignment for the $i$-th node in $\mathcal{G}_1$, and $\mathbb{1}\{\cdot\}$ is the indicator function. Note that in NA, Precision@K (Trung et al., 2020) is equivalent to Hits@K.

- **Mean Reciprocal Rank (MRR)**. MRR refers to the average reciprocal of the rank at which the correct alignment appears in the candidate list. Formally, In the case of aligning $\mathcal{G}_1$ to $\mathcal{G}_2$,

$$\text{MRR} = \frac{1}{N} \sum_{i=1}^{N} \frac{1}{\text{rank}_i},$$

where $N$ is the number of nodes in $\mathcal{G}_1$, $\text{rank}_i$ is the rank of the correct alignment for the $i$-th node in $\mathcal{G}_1$. Note that in NA, Mean Average Precision (MAP) (Trung et al., 2020) is equivalent to MRR.

**Diverse Benchmark Settings.** The design of PLANETALIGN enables diverse and reproducible benchmarking of existing NA algorithms with minimal efforts, providing a rich collection of APIs and built-in utility functions that allow users to easily configure, run, and evaluate experiments along key dimensions of NA performance, including effectiveness, scalability, and robustness.

Specifically, for *effectiveness*, PLANETALIGN supports custom training ratios and generates consistent, reproducible train/test splits by a user-defined random seed, ensuring fair comparisons of different NA algorithms. The APIs also supports experiments that evaluate the sensitivity of different NA methods with respect to the amount of supervision, providing valuable insights into their applicability to various supervision regimes. PLANETALIGN further provides unified utility functions to selectively introduce or remove supervision, enabling side-by-side comparisons between supervised and unsupervised algorithms under the same setting; for *scalability*, PLANETALIGN includes built-in logging functionalities that automatically track the runtime and memory usage during training and inference, allowing consistent and transparent evaluation of the efficiency of NA algorithms across datasets of varying sizes; for *robustness*, PLANETALIGN provides utility functions for injecting edge-level, attribute-level, and supervision noise into input graphs, allowing comprehensive evaluation of the robustness of NA methods under diverse graph noises or data inconsistencies.

## 5 EXPERIMENTS

Based on PLANETALIGN, we carry out extensive experiments to benchmark a wide range of NA algorithms across four key dimensions: effectiveness (Section 5.2), scalability (Section 5.3), robustness (Appendix D.1), and sensitivity to supervision (Appendix D.2). Additionally, we compare our implementation to the official implementation of built-in NA algorithms of PLANETALIGN to validate the correctness and efficiency our library (Appendix D.3).

### 5.1 EXPERIMENTAL SETUP

**Datasets and methods.** We benchmarks the performance of 14 NA algorithms on 18 NA datasets built into PLANETALIGN, as shown in Figure 2. Detailed dataset statistics and a brief introduction to each algorithm can be found at Appendix A and B, respectively.

**Metrics.** To evaluate effectiveness, we report Hits@K and MRR introduced in Section 4.3. All reported Hits@K and MRR are averaged results from both alignment directions. To evaluate scalability, we report the runtime and peak memory usage.

**Additional Setup.** For each experiments, we run 5 times and report the mean and standard deviation of the results. Additional experimental setup, such as the machine used to run the experiments and hyperparameter settings, are detailed in Appendix C.

### 5.2 EFFECTIVENESS RESULTS

We first evaluate the effectiveness of existing NA algorithms on plain networks under a semi-supervised setting with 20% training ratio. Datasets are randomly split for training and testing by a fixed random seed to ensure fair and reproducible comparison. Table 2 shows the averaged results on all 18 datasets group by their categories. Detailed results on plain and attributed NA datasets can be found in Appendix D.

We can see from Table 2 that OT-based methods, particularly PARROT (Zeng et al., 2023a) and JOENA (Yu et al., 2025), consistently achieve SOTA alignment performance in Hits@K and MRR across all datasets, demonstrating the effectiveness of optimal transport in aligning distributional

Table 2: Effectiveness and efficiency results of NA algorithms on plain networks with a training ratio of 20%. We group the 18 datasets in PLANETALIGN by their categories and report the averaged Hits@1, Hits@10, MRR (in %), inference time (Time), and peak memory usage (Memory). Cells that contain the 1st/2nd/3rd best results are highlighted in red/blue/green, respectively. Detailed results for each dataset can be found in Appendix D.5.

| Dataset | Social | | | Publication | | | Biological | | | Knowledge | | | Infrastructure | | | Communication | | |
|---|---|---|---|---|---|---|---|---|---|---|---|---|---|---|---|---|---|---|
| Metrics | Hits@1 | Hits@10 | MRR | Hits@1 | Hits@10 | MRR | Hits@1 | Hits@10 | MRR | Hits@1 | Hits@10 | MRR | Hits@1 | Hits@10 | MRR | Hits@1 | Hits@10 | MRR |
| IsoRank | 4.2 | 19.6 | 9.2 | 18.9 | 59.1 | 31.4 | 21.6 | 45.1 | 29.4 | 11.5 | 50.0 | 23.4 | 14.2 | 43.9 | 24.1 | 22.1 | 44.5 | 30.2 |
| FINAL | 4.9 | 22.3 | 10.1 | 22.3 | 68.6 | 37.3 | 22.9 | 56.6 | 34.1 | 13.9 | 55.4 | 27.3 | 15.1 | 53.6 | 28.0 | 21.7 | 57.7 | 33.9 |
| IONE | 7.9 | 20.0 | 12.1 | 28.7 | 63.6 | 40.1 | 46.1 | 60.5 | 51.0 | 4.7 | 20.3 | 10.0 | 29.6 | 51.2 | 37.0 | 50.4 | 69.0 | 56.7 |
| REGAL | 0.3 | 2.2 | 1.2 | 1.8 | 7.8 | 3.9 | 1.0 | 5.3 | 2.6 | 0.8 | 2.9 | 1.6 | 2.8 | 13.4 | 6.7 | 45.3 | 49.5 | 47.2 |
| CrossMNA | 1.2 | 11.1 | 4.5 | 13.2 | 58.1 | 27.2 | 40.2 | 58.7 | 46.5 | 2.7 | 28.8 | 10.7 | 14.6 | 30.9 | 20.2 | 22.8 | 53.3 | 33.9 |
| NetTrans | 7.2 | 21.8 | 11.9 | 40.7 | 77.3 | 52.7 | 34.2 | 57.5 | 41.8 | 28.8 | 62.8 | 39.7 | 29.3 | 59.5 | 39.6 | 45.2 | 62.6 | 51.8 |
| WAlign | 4.2 | 8.5 | 6.1 | 31.2 | 49.6 | 37.8 | 20.3 | 27.3 | 22.9 | 19.4 | 28.2 | 22.8 | 29.1 | 47.3 | 35.5 | 49.6 | 56.0 | 52.1 |
| BRIGHT | 5.1 | 17.0 | 9.0 | 40.4 | 74.0 | 51.8 | 30.5 | 48.0 | 36.5 | 30.4 | 61.7 | 40.9 | 29.9 | 57.0 | 39.5 | 50.9 | 62.3 | 55.0 |
| NeXtAlign | 7.1 | 19.5 | 11.3 | 43.2 | 76.9 | 54.7 | 25.9 | 44.8 | 32.8 | 27.5 | 59.9 | 38.3 | 28.0 | 55.1 | 37.8 | 29.6 | 51.4 | 37.2 |
| WLAlign | 7.6 | 14.8 | 10.1 | 35.9 | 58.1 | 43.2 | 41.2 | 50.7 | 44.3 | 25.9 | 44.2 | 31.7 | 29.5 | 42.8 | 34.1 | 34.1 | 49.1 | 39.5 |
| PARROT | 12.6 | 26.3 | 17.2 | 66.6 | 88.6 | 74.4 | 61.6 | 73.4 | 65.5 | 66.0 | 87.2 | 73.1 | 51.8 | 69.2 | 57.8 | 63.3 | 86.7 | 71.3 |
| SLOTAlign | 0.9 | 4.0 | 2.2 | 50.7 | 65.5 | 56.1 | 48.6 | 54.5 | 50.7 | 1.5 | 5.8 | 3.1 | 53.2 | 60.8 | 55.7 | 49.3 | 52.6 | 50.9 |
| HOT | 5.3 | 16.0 | 5.2 | 38.1 | 65.6 | 23.7 | 25.4 | 37.9 | 15.2 | 33.9 | 61.2 | 21.4 | 32.1 | 52.1 | 19.5 | 52.1 | 66.2 | 28.5 |
| JOENA | 18.7 | 35.1 | 24.4 | 73.2 | 92.1 | 80.2 | 63.7 | 72.9 | 66.8 | 66.3 | 87.8 | 73.0 | 62.9 | 75.0 | 67.2 | 66.3 | 89.0 | 74.3 |

| Dataset | Social | | Publication | | Biological | | Knowledge | | Infrastructure | | Communication | |
|---|---|---|---|---|---|---|---|---|---|---|---|---|
| Metrics | Time(s) | Memory(GB) | Time(s) | Memory(GB) | Time(s) | Memory(GB) | Time(s) | Memory(GB) | Time(s) | Memory(GB) | Time(s) | Memory(GB) |
| IsoRank | 25.17 | 3.54 | 57.99 | 6.89 | 13.01 | 2.67 | $1.54 \times 10^2$ | 15.97 | 0.28 | 0.66 | 0.17 | 0.80 |
| FINAL | 5.91 | 5.39 | 6.75 | 10.06 | 1.88 | 3.54 | 18.10 | 24.37 | 0.10 | 0.80 | 0.11 | 0.88 |
| IONE | $6.34 \times 10^3$ | 1.94 | $1.43 \times 10^4$ | 4.16 | $1.41 \times 10^4$ | 1.93 | $1.50 \times 10^4$ | 8.75 | $9.41 \times 10^3$ | 0.90 | $8.10 \times 10^3$ | 1.02 |
| REGAL | 9.38 | 1.16 | 16.14 | 3.18 | 7.28 | 1.55 | 30.83 | 5.96 | 0.76 | 0.81 | 1.17 | 0.77 |
| CrossMNA | $3.06 \times 10^2$ | 1.40 | $1.16 \times 10^3$ | 3.16 | $5.33 \times 10^2$ | 1.58 | $1.11 \times 10^3$ | 6.14 | 59.03 | 0.98 | $5.78 \times 10^2$ | 0.81 |
| NetTrans | $1.56 \times 10^2$ | 8.88 | $5.14 \times 10^2$ | 8.57 | $1.08 \times 10^2$ | 3.90 | $3.65 \times 10^2$ | 21.90 | 6.46 | 1.28 | 12.37 | 1.54 |
| WAlign | 0.61 | 2.65 | 9.41 | 9.88 | 2.46 | 3.86 | 10.05 | 15.40 | 0.12 | 1.43 | 0.20 | 1.11 |
| BRIGHT | 21.76 | 3.00 | $1.26 \times 10^2$ | 5.66 | 33.55 | 3.24 | $3.81 \times 10^2$ | 11.53 | 0.28 | 1.14 | 0.33 | 1.12 |
| NeXtAlign | 40.89 | 3.75 | $1.55 \times 10^2$ | 7.82 | 19.35 | 3.57 | $2.62 \times 10^3$ | 13.57 | 0.29 | 1.34 | 2.44 | 0.99 |
| WLAlign | $7.41 \times 10^2$ | 2.17 | $2.69 \times 10^3$ | 11.98 | $1.15 \times 10^3$ | 4.21 | $3.24 \times 10^3$ | 28.33 | $4.25 \times 10^2$ | 1.35 | $6.99 \times 10^2$ | 0.95 |
| PARROT | 76.76 | 6.26 | $2.99 \times 10^2$ | 11.68 | 84.63 | 3.98 | $8.95 \times 10^2$ | 28.47 | 0.76 | 0.85 | 0.82 | 0.90 |
| SLOTAlign | 46.31 | 6.40 | $5.64 \times 10^2$ | 11.55 | 67.66 | 3.96 | $1.03 \times 10^4$ | 28.27 | 3.85 | 1.03 | 1.23 | 0.80 |
| HOT | $4.01 \times 10^2$ | 3.85 | $7.89 \times 10^2$ | 7.75 | $7.08 \times 10^2$ | 4.44 | $2.08 \times 10^3$ | 18.43 | 8.95 | 2.12 | 7.04 | 4.46 |
| JOENA | 58.73 | 4.89 | 30.60 | 2.73 | 3.65 | 1.39 | $6.61 \times 10^2$ | 26.47 | 0.05 | 1.02 | 0.49 | 0.86 |

structures. Embedding-based methods such as IONE (Liu et al., 2016), NetTrans (Zhang et al., 2020), and BRIGHT (Yan et al., 2021b) can be effective in aligning some networks. However, their strong performances are not consistent across different datasets, potentially due to the space disparity issue (Yan et al., 2021b; Zhang et al., 2021). Consistency-based methods, while occasionally perform well on certain datasets, usually outperformed by best-performing embedding-based and OT-based methods, suggesting that relying solely on consistency principles may lead to sub-optimal alignment.

In addition to empirical observations, we further provide theoretical analysis into the superior performance of OT-based alignment methods. Compared with consistency-based methods, which are restricted by local consistency principles, OT-based methods go beyond local assumptions by solving a globally constrained optimization problem. Compared with embedding-based methods, which infer alignment from noisy embedding similarities, OT-based methods directly learns a robust alignment matrix from the transportation cost, thanks to the marginal constraints that naturally encourage one-to-one node alignment. Empowered by constrained optimization and informative transportation cost encoded by powerful graph proximity measures or learnable node embeddings, OT-based methods learns *robust*, *deterministic*, and *global-structure-aware* alignment.

> **Takeaway #1: Optimal transport demonstrates significant potentials in NA.**
>
> *Best-performing OT-based methods consistently outperform consistency and embedding-based approaches by a significant margin in alignment performance across diverse domains, demonstrating the power of constrained optimization and informative transport cost which lead to robust, deterministic, and global-structure-aware alignment.*

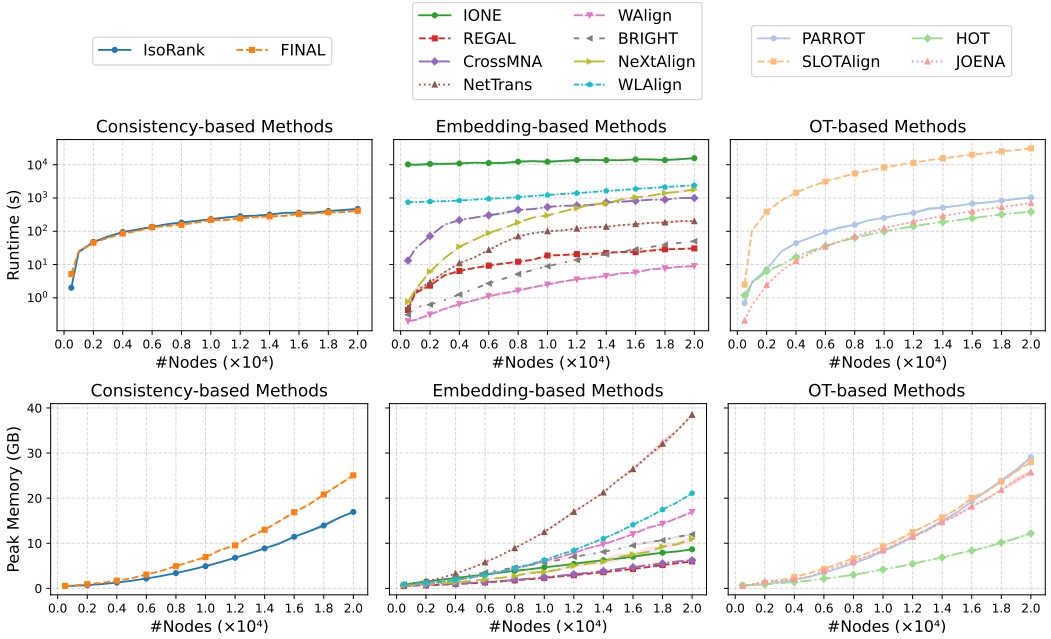

Figure 4: Scalability results on ER graphs. The **x-axis** shows the number of nodes in the ER graphs (in $10^4$), and the **y-axis** of the 1st/2nd row shows the inference time and peak memory usage, respectively.

## 5.3 Efficiency and Scalability Results

We also include the efficiency results of NA algorithms on plain networks under a semi-supervised setting with a 20% training ratio in Table 2. To further evaluate the scalability of NA algorithms, we conduct another set of experiments on synthetic graphs generated by the Erdős–Rényi (ER) (Erdos et al., 1960) model with a fixed average node degree of 10 under the same semi-supervised setting, and record the inference time and peak memory usage as the number of nodes increases in Figure 4.

Embedding-based methods typically face a two-out-of-three trade-off among effectiveness, time efficiency, and memory efficiency. Specifically, WAlign (Gao et al., 2021) and BRIGHT (Yan et al., 2021b) are among the most scalable algorithms in terms of inference time thanks to simple neural network (NN) structures which only requires a forward pass during inference. However, they tends to be less scalable in memory usage due to overheads of NN parameters. In terms of memory usage, CrossMNA (Chu et al., 2019) and IONE (Liu et al., 2016) achieve the best scalability as they learn low-dimensional embeddings without using NN. However, they tends to be less scalable in time since their transductive embeddings requires retraining for different networks (Hamilton et al., 2017). REGAL (Heimann et al., 2018) achieve both time and memory scalability by decomposition on sampled embedding matrices but is less effective compared to other NA methods.

> **Takeaway #2: Embedding-based methods face a two-out-of-three trade-off among effectiveness, time efficiency, and memory efficiency.**
>
> *Embedding-based methods face trade-off among transductive embeddings for memory efficiency, inductive embeddings for time efficiency, and learning-based approaches for effectiveness.*

Consistency-based methods (Singh et al., 2008; Zhang & Tong, 2016), on the other hand, scale moderately in terms of both time and memory usage. OT-based methods, in general, share similar scalability results as consistency-based methods since the optimizations of both kinds of methods involve matrix operations of quartic complexity. Although the original OT problem is non-convex and computationally expensive to solve by gradient descent (Tang et al., 2023), PARROT (Zeng et al., 2023a) and JOENA (Yu et al., 2025) solve the OT problem efficiently by convex approximation (Peyré et al., 2019) and proximal point methods (Xu et al., 2019a). HOT (Zeng et al., 2024a) further utilizes a hierarchical OT framework for cluster-level alignment to scale efficiently to large networks.

> **Takeaway #3: OT-based methods requires efficient optimization methods to scale similarly as consistency-based methods.**
>
> *OT-based methods requires efficient optimization of OT problem, e.g, convex approximation, to scale moderately like consistency-based methods in terms of both time and memory usage.*

## 5.4 ROBUSTNESS AND SENSITIVITY RESULTS

We evaluate the robustness of NA methods under various types of graph noises, as well as their sensitivity to different levels of supervision. Our key findings are twofold. *First*, different NA methods show distinct sensitivities to different types of graph noises, suggesting that effective integration of different alignment techniques can potentially improve the overall robustness of NA algorithms. *Second*, current NA algorithms remain sensitive to supervision, underscoring the need for future research on self-supervised and active alignment approaches. Due to space constraints, detailed experimental results, analysis, and key takeaways are provided in Appendix D.1 and D.2.

## 6 CONCLUSION

In this paper, we introduce PLANETALIGN, a comprehensive library that facilitates the benchmarking and development of network alignment methods. PLANETALIGN highlights a collection of 18 different public datasets spanning 6 different domains, along with a unified and efficient implementation of 14 different NA algorithms of 3 different categories. With a comprehensive list of evaluation metrics, benchmarking tools, and utility functions implemented as easy-to-use APIs, PLANETALIGN not only enables fair and reproducible benchmarking of NA algorithms but also facilitates the development of new NA methods. Through extensive benchmark, we reveal practical insights into the strengths and limitations of existing NA methods which guides the development of future NA algorithms.

## 7 LIMITATIONS AND FUTURE WORK

While we introduce a comprehensive library for benchmarking NA, PLANETALIGN could be potentially improved from the following two directions. *First*, although PLANETALIGN features a rich collection of baselines, some variants of NA methods that targets a specific kind of network remain uncovered, e.g., entity alignment approaches (Chen et al., 2023; Liu et al., 2023b; Yan et al., 2021a) for aligning knowledge graphs. *Second*, PLANETALIGN focuses primarily on benchmarking pairwise NA problems. Although multi-network alignment methods are included in PLANETALIGN (Chu et al., 2019; Zeng et al., 2024a), benchmarking under a simultaneous multi-network alignment setting remains underexplored at this stage.

As for future work, we will continuously expand PLANETALIGN to incorporate new NA datasets, algorithms, benchmark settings, and utility functions. Specifically, *for NA datasets*, we plan to include 1) multi-network alignment datasets which consist of more than two networks, such as multi-layered version of ArXiv (De Domenico et al., 2015a), Twitter (Omodei et al., 2015), and SacchCere (De Domenico et al., 2015b), 2) dynamic networks which evolves over time, such as synthetic datasets from (Vijayan et al., 2017) and (Yan et al., 2021a), and 3) cross-domain datasets which consist of networks from different domains, such as text-image network constructed by GOT (Chen et al., 2020); *for NA algorithms*, we plan to introduce 1) domain-specific alignment algorithms, such as entity alignment methods DualMatch (Liu et al., 2023b) and MEAformer (Chen et al., 2023), 2) multi-network alignment algorithms such as MrMine (Du & Tong, 2019), 3) dynamic NA algorithms such as DynaMAGNA++ (Vijayan et al., 2017) and DINGA (Yan et al., 2021a); *for benchmark settings*, we plan to add additional evaluation metrics for measuring the uncertainty of the alignment (Zhou et al., 2021), which are critical for developing active or self-improving NA methods highlighted as important future directions in our paper; for utility functions, our immediate goal is to introduce scalability tools to allow easy acceleration of NA algorithms built upon PLANETALIGN, including distributed training APIs, sparse and low-rank matrix optimizations, and low-rank (Scetbon & Cuturi, 2022) & sliced OT (Liu et al., 2024b) optimization tools.

## ACKNOWLEDGMENTS

This work is supported by NSF (2316233) and AFOSR (FA9550-24-1-0002). The content of the information in this document does not necessarily reflect the position or the policy of the Government, and no official endorsement should be inferred. The U.S. Government is authorized to reproduce and distribute reprints for Government purposes notwithstanding any copyright notation here on.

## ETHICS STATEMENT

Our library uses only publicly available datasets and conducts evaluation in a transparent and responsible manner in accordance with the code of ethics of ICLR. The research does not involve human subjects, animal studies, or any other procedures that may raise ethical concerns.

## REPRODUCIBILITY STATEMENT

To ensure reproducibility, for **datasets** in PLANETALIGN, we include their detailed statistics and description in Appendix A. For **experimental setup**, we include detailed description of adopted evaluation metrics, machines, dataset splits, and hyperparameter settings in Section 5.1 and Appendix C. The **source code** of PLANETALIGN is available at https://github.com/yq-leo/PlanetAlign, with detailed **documentation** at https://planetalign.readthedocs.io/en/latest/.

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

# TABLE OF APPENDIX

# A DATASETS DETAILS

## A.1 DATASET STATISTICS

Table 3: Dataset Statistics.

| Domain | Networks | # nodes | # edges | # node attr. | # edge attr. | Type |
|---|---|---|---|---|---|---|
| Social Networks | Foursquare | 5,313 | 54,233 | 0 | 0 | Real-world |
| | Twitter | 5,120 | 130,575 | 0 | 0 | |
| | Douban(online) | 3,906 | 8,164 | 538 | 2 | Real-world |
| | Douban(offline) | 1,118 | 1,511 | 538 | 2 | |
| | Flickr | 12,974 | 16,149 | 3 | 3 | Real-world |
| | Lastfm | 15,436 | 16,319 | 3 | 3 | |
| | Flickr | 6,714 | 7,333 | 3 | 3 | Real-world |
| | Myspace | 10,733 | 11,081 | 3 | 3 | |
| Communication Networks | Phone | 1,000 | 41,191 | 0 | 0 | Real-world |
| | Email | 1,003 | 4,628 | 0 | 0 | |
| | Arenas1 | 1,135 | 10,902 | 50 | 0 | Synthetic |
| | Arenas2 | 1,135 | 10,800 | 50 | 0 | |
| Publication Networks | ACM | 9,872 | 39,561 | 17 | 0 | Real-world |
| | DBLP | 9,916 | 44,808 | 17 | 0 | |
| | Cora1 | 2,708 | 6,334 | 1,433 | 0 | Synthetic |
| | Cora2 | 2,708 | 4,542 | 1,433 | 0 | |
| | ArXiv1 | 18,722 | 217,921 | 0 | 0 | Synthetic |
| | ArXiv2 | 18,722 | 168,394 | 0 | 0 | |
| Biological Networks | SacchCere1 | 5,928 | 66,150 | 0 | 0 | Real-world |
| | SacchCere2 | 5,042 | 29,599 | 0 | 0 | |
| | PPI1 | 3,480 | 117,429 | 50 | 0 | Synthetic |
| | PPI2 | 3,480 | 90,741 | 50 | 0 | |
| | GGI1 | 10,403 | 115,755 | 0 | 0 | Synthetic |
| | GGI2 | 10,403 | 89,448 | 0 | 0 | |
| Knowledge Graphs | DBP15K_ZH | 19,388 | 70,414 | 300 | 0 | Real-world |
| | DBP15K_EN | 19,572 | 95,142 | 300 | 0 | |
| | DBP15K_JA | 19,814 | 77,214 | 300 | 0 | Real-world |
| | DBP15K_EN | 19,780 | 93,484 | 300 | 0 | |
| | DBP15K_FR | 19,661 | 105,997 | 300 | 0 | Real-world |
| | DBP15K_EN | 19,993 | 115,722 | 300 | 0 | |
| Infrastructure Networks | Italy1 | 349 | 416 | 0 | 0 | Real-world |
| | Italy2 | 349 | 435 | 0 | 0 | |
| | Airport1 | 1,190 | 14,958 | 4 | 0 | Synthetic |
| | Airport2 | 1,190 | 11,560 | 4 | 0 | |
| | PeMS08-1 | 170 | 301 | 3 | 0 | Synthetic |
| | PeMS08-2 | 170 | 233 | 3 | 0 | |

## A.2 DATASET DESCRIPTIONS

Detailed datasets descriptions are introduced as follows

- **Foursquare-Twitter** (Zhang & Philip, 2015). A pair of online social networks, Foursquare and Twitter. Nodes represent users and an edge exists between two users if they have follower/followee relationships. Both networks are plain networks. There are 1,609 common users across two networks.
- **Douban** (Zhang & Tong, 2016). A pair of online-offline social networks constructed from Douban. Nodes represent users and edges represent user interactions on the website. The location of a user is treated as the node attribute, and the contact/friend relationship are treated as the edge attributes. There are 1,118 common user across the two networks.

- **Flickr-LastFM** (Zhang & Tong, 2016). A pair of social networks from Flickr and LastFM. Nodes in both networks represent users, and edges represent friend / following relationships in Flickr and LastFM, respectively. The gender of a user is treated as the node attributes (male, female, unknown), and the level of people a user is connected to is treated as the edge attributes (e.g., leader with leader). There are 452 common users across two networks.
- **Flickr-MySpace** (Zhang & Tong, 2016). A pair of social networks from Flickr and MySpace. Nodes in both networks represent users, and edges represent friend / following relationships. The gender of a user is treated as the node attributes (male, female, unknown), and the level of people a user is connected to is treated as the edge attributes (e.g., leader with leader). There are 267 common users across two networks.
- **ACM-DBLP** (Tang et al., 2008). A pair of undirected co-authorship networks, ACM and DBLP. Nodes represent authors and edges an edge exists between two authors if they co-author at least one paper. Node attributes are available in both networks. There are 6,325 common authors across two networks.
- **Cora** (Yang et al., 2016). A pair of networks synthesized from the citation network Cora. Each network Nodes represent publications and an edge exists between two publications if they have a citation relationship. The two networks are noisy permutations of the original network generated by randomly inserting 10% edges (Cora1) and deleting 15% edges (Cora2) from the original network, respectively. There are in total 2,708 common nodes across two networks.
- **ArXiv** (Leskovec et al., 2007). A pair of networks synthesized from the Arxiv ASTRO-PH (Astro Physics) collaboration network (Leskovec et al., 2007). Nodes represent authors and an edge exists between two authors if they have co-authored a paper. The two networks are noisy permutations of the original network generated by randomly inserting 10% edges (ArXiv1) and deleting 15% edges (ArXiv2) from the original network, respectively. Node and edge attributes are not available. There are in total 18,722 common nodes across two networks.
- **SacchCere** (Stark et al., 2006; De Domenico et al., 2015b). A pair of direct interaction layer and association layer from the SacchCere multiplex GPI network. The SacchCere network consider different kinds of protein and genetic interactions for Saccharomyces Cerevisiae in BioGRID (Stark et al., 2006), a public database that archives and disseminates genetic and protein interaction data from humans and model organisms. There are in total 1,337 common nodes across two layers of networks.
- **PPI** (Zitnik & Leskovec, 2017). A pair of networks synthesized from the protein-protein interaction (PPI) network (Zitnik & Leskovec, 2017), where nodes represent human proteins and edges represent physical interaction between proteins in a human cell. The immunological signatures are included as node features. The two networks are noisy permutations of the original network generated by randomly inserting 10% edges (PPI1) and deleting 15% edges (PPI2) from the original network, respectively. There are in total 3,980 common nodes across two networks.
- **GGI** (Park et al., 2010). A pair of networks synthesized from the human gene-gene interaction (PPI) network from IsoBase (Park et al., 2010). Nodes represent human genes and edges represent gene-gene interactions. The two networks are noisy permutations of the original network generated by randomly inserting 10% edges (GGI1) and deleting 15% edges (GGI2) from the original network, respectively. There are in total 10,403 common nodes across two networks.
- **DBP15K ZH-EN, JA-EN, FR-EN** (Sun et al., 2017). Pairs of Chinese, Japanese, and French to English version of multi-lingual DBpedia networks. The node attributes are given by pre-trained and aligned monolingual word embeddings (Xu et al., 2019b). There are 15,000 pairs of aligned entities in DBP15K ZH-EN (Chinese to English), JA-EN (Japanese to English), and FR-EN (French to English), respectively.
- **Italy** (Yan et al., 2022). A pair of power grid networks from two regions in Italy. Nodes represent power stations and edges represent the existence of power transfer lines. Node attributes are derived from node labels. There are in total 377 common nodes across two networks inferred from the ground-truth cross-layer dependencies.
- **Airport** (Zhu et al., 2021). A pair of networks synthesized from the American air-traffic network (Ribeiro et al., 2017). Nodes represent airports and an edge exists between two aiports if there are commercial flights between them. The level of activity in each airport is used as node attributes. The two networks are noisy permutations of the original network generated by randomly inserting 10% edges (Airport1) and deleting 15% edges (Airport2) from the original network, respectively. There are in total 1,190 common nodes across two networks.
- **PeMS08** (Song et al., 2020). A pair of traffic networks synthesized from the Performance Measurement System (PeMS) Data Source. Nodes represent sensors and edges indicate traffic flow

correlation. Node attributes are averaged across all time interval. The two networks are noisy permutations of the original network generated by randomly inserting 10% edges (PeMS08-1) and deleting 15% edges (PeMS08-2) from the original network, respectively. There are in total 170 common nodes across two networks.

- **Phone-Email** (Zhang et al., 2017). A pair of communication networks among people via phone or email. Nodes represent people and an edge exists between two people if they communicate via phone or email at least once. Phone network includes 1,000 nodes and 41,191 edges. Email network includes 1,003 nodes and 4,627 edges. Both networks are plain networks. There are 1,000 common people across two networks.
- **Arenas** (Kunegis, 2013). A pair of networks synthesized from the email communication network Arenas at the University Rovira i Virgili. Nodes are users and each edge represents that at least one email was sent. The two networks are noisy permutations of each other. There are in total 1,135 common nodes across two networks.

## B    NETWORK ALIGNMENT METHODS

### B.1    CONSISTENCY-BASED METHODS

- **IsoRank** (Singh et al., 2008). IsoRank is originally designed for global alignment of multiple PPI networks. It is built upon neighborhood topology consistency which assumes that the neighbors of aligned anchor nodes should be aligned as well, and is formulated as an eigenvalue problem. (Yan et al., 2021b) reveals that the formulation of IsoRank can be considered as conducting random walk propagation of anchor links on the product graph to achieve topology consistency.
- **FINAL** (Zhang & Tong, 2016). FINAL interprets the alignment consistency principles as an optimization problem and introduces additional consistency principles at node/edge attribute levels to handle attributed network alignment.

### B.2    EMBEDDING-BASED METHODS

- **IONE** (Liu et al., 2016). IONE modeled the follower/followee-ship of different nodes as input/output context vectors to learn proximity-preserving node embeddings, and solve the node embedding and network alignment problem based on a unified framework.
- **REGAL** (Heimann et al., 2018). REGAL designs an embedding learning methods called xNetMF which learns powerful node embeddings by matrix factorization on a linear combination between cross-network structural and attribute similarity matrix. Based on xNetMF embeddings, REGAL infer node-level alignment of two networks based on Euclidean distance of nodes in the embedding space.
- **CrossMNA** (Chu et al., 2019). CrossMNA leverages cross-network structural information to learn inter and intra network embeddings simultaneously. By comparing inter network embeddings across different networks, CrossMNA is capable of aligning multiple networks at the same time.
- **NetTrans** (Zhang et al., 2020). NetTrans approach the network alignment problem from a cross-network transformation perspective. It learns the transformation of both network structure and node attributes at different resolutions to identify node-level alignment.
- **WAlign** (Gao et al., 2021). WAlign learns node embeddings by a lightweight GCN model to capture both local and global graph patterns and proposes a Wasserstein distance discriminator to minimize the Wasserstein distance between node embeddings across different graphs.
- **BRIGHT** (Yan et al., 2021b). BRIGHT first generate positional node embeddings by random walk with restart (RWR) (Tong et al., 2006) against anchor links. To handle plain network alignment, BRIGHT-U learns position-aware embeddings by transforming RWR embeddings through a shared MLP. To handle attributed network alignment, BRIGHT-A use a shared GCN model for transforming node attributes and concatenates the output embeddings with RWR vectors before feeding into the shared MLP.
- **NeXtAlign** (Zhang et al., 2021). NeXtAlign designs a spatial GCN model and learns node embeddings that balance the alignment consistency and disparity by crafting the sampling strategy.
- **WLAlign** (Liu et al., 2023a). WLAlign proposes a cross-network Weisfeiler-Lehman relabeling scheme to learn embeddings that preserves long-range connectivity to the anchor pairs on plain networks.

### B.3 OT-BASED METHODS

- **SLOTAlign** (Tang et al., 2023). SLOTAlign utilizes a parameter-free GCN model to encode graph structure. By integrating output embeddings of multiple layers of GCN through a learnable linear combination, SLOTAlign encode the Gromov-Wasserstein distance between two networks via the learned embeddings and optimize the embedding and optimal transport problem alternatively to infer alignment.
- **PARROT** (Zeng et al., 2023a). PARROT encodes a position-aware transportation cost by random walk with restart (RWR) (Tong et al., 2006) on separate and product graphs, and integrate consistency principle at node, edge, and neighborhood levels into the optimal transport formulation. Then, it solves the resulting optimization problem efficiently via constrained proximal point methods to infer node-level alignment.
- **HOT** (Zeng et al., 2024a). HOT proposes a hierarchical multi-marginal optimal transport framework which first decomposes multiple networks to aligned clusters via the fused Gromov-Wasserstein (FGW) barycenter (Peyré et al., 2016) and then aligns node in aligned clusters simultaneous by solving optimal transport problem within clusters.
- **JOENA** (Yu et al., 2025). JOENA transforms the transport plan of optimal transport into an adaptive sampling strategy via a learnable transformation to learn node embeddings and alignment in a mutual beneficial manner.

## C DETAILED EXPERIMENTAL SETUP

**Machine.** All experiments are conducted on a computing server equipped with dual Intel® Xeon® Gold 6240R CPUs and 4 NVIDIA Tesla V100-SXM2 GPUs with 32GB memory each.

**Dataset split and hyperparameters.** To mitigate the randomness introduced by a single random dataset split, we report the average metrics of 5 different dataset split based on 5 randomly selected seeds. All NA methods are evaluated under the same dataset splits to ensure a fair comparison. For each dataset split, we run a NA algorithm 5 times and report the average metrics. Hyperparameters are tuned with a fixed budget of 5 per key parameter based on the default values and hyperparameter study in the original papers. Detailed hyperparameter search spaces can be found in Table 4.

Table 4: Hyperparameter search spaces.

| NA Method | Search Parameters |
|---|---|
| **IsoRank** | $\alpha \in \{0.1, 0.3, 0.5, 0.7, 0.9\}$ |
| **FINAL** | $\alpha \in \{0.1, 0.3, 0.5, 0.7, 0.9\}$ |
| **IONE** | out_dim$\in \{32, 64, 100, 128, 256\}$ |
| **REGAL** | k$\in \{1, 5, 10, 15, 20\}$, num_layers$\in \{1, 2, 3, 4, 5\}$, $\alpha \in \{0.001, 0.005, 0.01, 0.05, 0.1\}$ |
| **CrossMNA** | $d_1 \in \{10, 50, 100, 150, 200\}$, $d_2 \in \{10, 50, 100, 150, 200\}$ |
| **NetTrans** | $\alpha \in \{0.01, 0.1, 1, 10, 100\}$, $\beta \in \{0.01, 0.1, 1, 10, 100\}$, $\gamma \in \{0.01, 0.1, 1, 10, 100\}$, $L \in \{1, 2, 3, 4, 5\}$ |
| **WAlign** | $h \in \{128, 256, 512, 1024, 2048\}$, $\alpha \in \{0.01, 0.02, 0.04, 0.06, 0.08\}$ |
| **BRIGHT** | $\beta \in \{0.05, 0.1, 0.15, 0.2, 0.25\}$, out_dim$\in \{32, 64, 128, 256, 512\}$, neg_sample_size$\in \{100, 300, 500, 700, 900\}$ |
| **NeXtAlign** | $\beta \in \{0.05, 0.1, 0.15, 0.2, 0.25\}$, out_dim$\in \{32, 64, 128, 256, 512\}$, neg_sample_size$\in \{100, 300, 500, 700, 900\}$ |
| **WLAlign** | out_dim$\in \{32, 64, 128, 256, 512\}$, neg_sample_size$\in \{20, 40, 60, 80, 100\}$ |
| **PARROT** | $\eta \in \{0.1, 0.5, 1, 5, 10\}$, $\lambda_e \in \eta\lambda_e^{\text{default}}$, $\lambda_n \in \eta\lambda_n^{\text{default}}$, $\lambda_a \in \eta\lambda_a^{\text{default}}$, $\lambda_p \in \eta\lambda_p^{\text{default}}$ |
| **SLOTAlign** | $\epsilon \in \{0.001, 0.005, 0.01, 0.05, 0.1\}$, step_size$\in \{1, 2, 3, 4, 5\}$ |
| **HOT** | $\epsilon \in \{0.001, 0.005, 0.01, 0.05, 0.1\}$, $\alpha \in \{0.1, 0.3, 0.5, 0.7, 0.9\}$ |
| **JOENA** | $\alpha \in \{0.1, 0.3, 0.5, 0.7, 0.9\}$, $\gamma_p \in \{0.001, 0.005, 0.01, 0.05, 0.1\}$, $\lambda_0 \in \{0.1, 0.5, 1.0, 1.5, 2.0\}$ |

## D ADDITIONAL EXPERIMENTAL RESULTS

### D.1 ROBUSTNESS RESULTS

To benchmark the robustness of existing NA algorithms, we conduct controlled experiments to study the impact of edge, attribute, and supervision noises to alignment performance, offering practical insights into the development of robust NA methods.

**Edge noise.** We introduce edge-level noise to simulate real-world edge perturbation (Jin et al., 2020). Specifically, the p% edge noise level is defined as randomly adding/deleting p% edges in

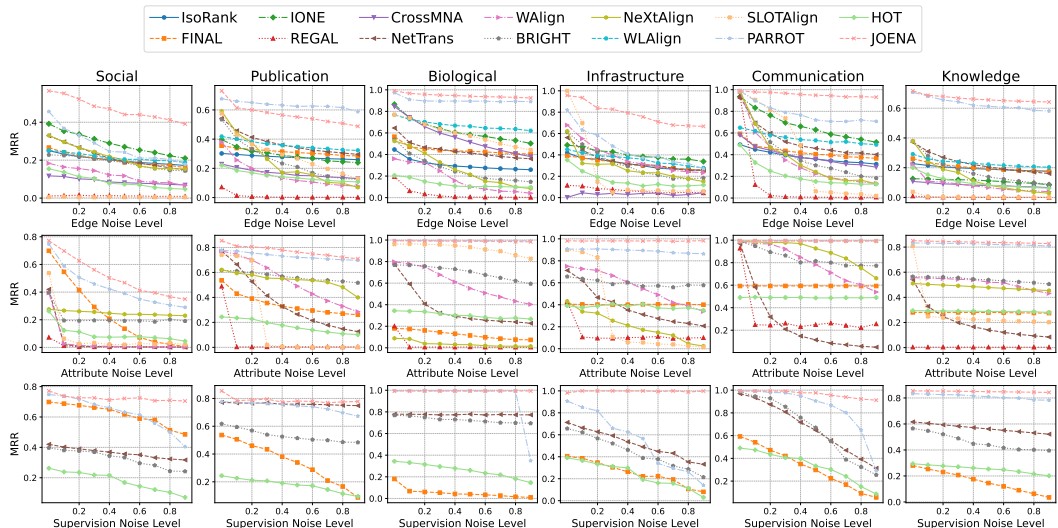

Figure 5: Robustness results of NA methods under different levels of edge, attribute, and supervision noises on representative datasets across 6 domains. The **x-axis** of the plots in the 1st/2nd/3rd row shows the noise level of edge/attribute/supervision, respectively, and the **y-axis** shows the MRR.

the second network to be aligned (Tang et al., 2023; Zeng et al., 2023a). We conduct evaluations of all NA methods under a semi-supervised (20% training ratio) plain NA setup to avoid potential interference of node/edge attributes.

**Attribute noise.** We introduce attribute-level noise to simulate real-world attribute perturbation (Zheng et al., 2021). Specifically, the p% attribute noise level is defined as randomly perturbing [2] p% node and edge attributes in the second network to be aligned (Zeng et al., 2023a). We conduct evaluations of attributed NA methods under a semi-supervised attributed NA setting with a training ratio of 20%.

**Supervision noise.** We introduce supervision noise to evaluate the robustness of semi-supervised NA methods against noisy anchor node pairs (Yan et al., 2021b; Tang et al., 2023). Specifically, the p% supervision noise is defined as randomly setting p% anchor nodes in the second graph to non-anchor nodes. To ensure fair comparison, we only evaluate the robustness of semi-supervised attributed NA methods [3] against supervision noise under a semi-supervised attributed NA setting with a training ratio of 20%.

**Analysis.** Robustness results on five representative datasets are shown in Figure 5. Firstly, for **edge noise**, consistency-based methods, including IsoRank (Singh et al., 2008) and FINAL (Zhang & Tong, 2016), are among the most robust methods with the slightest performance drop across all datasets. Embedding-based methods (Liu et al., 2016; Heimann et al., 2018; Chu et al., 2019; Zhang et al., 2020; Gao et al., 2021; Yan et al., 2021b; Zhang et al., 2021; Liu et al., 2023a), while slightly less robust than consistency-based approaches, generally show descent performance degradation ratio as edge noise level increases. OT-based methods (Zeng et al., 2023a; Tang et al., 2023; Zeng et al., 2024a; Yu et al., 2025), on the other hand, differ significantly in terms of robustness to edge-level noisel, indicating that although OT can reduce the negative effect of graph noises by marginal constraint (Zeng et al., 2023a; Yu et al., 2025), they require careful design of the transportation costs to avoid noise amplification during optimization. Nevertheless, OT-based methods PARROT (Zeng et al., 2023a) and JOENA (Yu et al., 2025) consistently outperforms all other NA algorithms in alignment performance across different noise levels.

---

[2] We flip binary attributes and add standard gaussian noise into normalized continuous attributes.
[3] We include FINAL (Zhang & Tong, 2016) which has a semi-supervised version in its original paper.

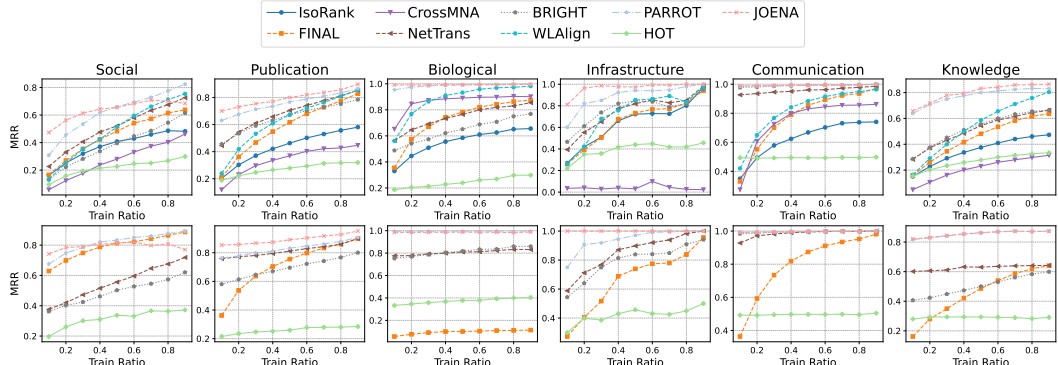

Figure 6: Sensitivity results of semi-supervised NA algorithms to different levels of supervision. The two rows of plots correspond to plain (upper row) and attributed (lower row) NA settings respectively. The **x-axis** shows the training ratio (i.e., supervision level), and the **y-axis** shows the MRR.

Secondly, for **attribute noise**, PARROT (Zeng et al., 2023a) and JOENA (Yu et al., 2025) becomes the most robust algorithms across all datasets. While both methods are OT-based, PARROT integrates consistency principles which further improve its robustness, and JOENA adopts an embedding-encoded OT cost learned via a MLP for robust alignment. Consistency-based methods remain robust to attribute noise on most datasets. Embedding-based methods are generally more sensitive to attribute noise than edge noise and suffer from significant performance drop under high attribute noise level, which highlights the need for more robust embedding learning approaches, potentially through the integration of optimal transport or consistency principles.

Finally, for **supervision noise**, the performance of most NA algorithms degrades significantly as the noise level increase, indicating that the effectiveness of existing semi-supervised NA methods rely heavy on the quality of anchor node pairs even when node/edge attributes are available. Nevertheless, JOENA (Yu et al., 2025) consistently shows the mildest performance drop across all datasets, demonstrating the power of effective combination of embedding and OT-based methods. Future methods may explore more effective integration of consistency, embedding, and OT-based approaches to better handle different kinds of real-world noise.

> **Takeaway #4: Different NA methods are sensitive to different kinds of noises. Effective integration of different NA techniques could be a way out.**
>
> *Different NA methods may be sensitive to different kinds of real-world noises. Integrating different NA techniques effectively, such as consistency principles, embedding learning, and optimal transport, could potentially improve the overall robustness of NA algorithms.*

## D.2 SENSITIVITY TO SUPERVISION RESULTS

To comprehensively evaluate the impact of supervision on the performance of NA algorithms, we conduct a set of experiments to study the sensitivity of semi-supervised NA methods to different levels of supervision. Specifically, we gradually increase the training ratio and report the MRR of semi-supervised NA methods on five representative datasets under both plain and attributed NA settings. The results are presented in Figure 6.

**Analysis.** Firstly, the performance of NA algorithms generally shows a growing trend as the training ratio increases, with only a few exceptions such as JOENA (Yu et al., 2025) on Douban (Zhang & Tong, 2016), potentially due to overfitting on training data or the presence of noisy anchor pairs from real-world datasets. Nonetheless, most NA methods benefit significantly from increased supervision, demonstrating its importance to the effectiveness of NA algorithms.

Secondly, the use of attribute information typically improves the performance of NA algorithms under low supervision. However, the performance gap between plain and attributed settings narrows as the training ratio increases. For example, PARROT (Zeng et al., 2023a) achieve an MRR of

approximately 0.7/0.3 on Douban with/without attribute information under 10% training ratio, whereas the performance rises to about 0.95/0.9 under a 90% training ratio. This suggests that while attributes can help in low-supervision scenarios, increasing supervision remains crucial even in the presence of node and edge attributes in graphs. Combined with our previous robustness study against supervision noise, we present the following findings:

> **Takeaway #5: Supervision greatly affect the effectiveness of NA algorithms.**
>
> *The quality and quantity of supervision greatly affect the performance of NA algorithms even in the presence of node and edge attributes, suggesting that self-supervised and active learning methods that discover high-quality anchors could be a promising directions for NA research.*

### D.3  COMPARISON WITH OFFICIAL IMPLEMENTATIONS

Table 5: Performance and runtime comparison with official implementations averaged against all datasets. Δ represents the absolute difference between official and PLANETALIGN's implementation.

| Metrics | MRR | | | Training Runtime(s) | | | |
|---|---|---|---|---|---|---|---|
| Version | Official | PLANETALIGN | Δ | Official | PLANETALIGN | Δ | Speedup |
| REGAL | 0.079 | **0.080** | +0.001 | 20 | **14** | -6 | 1.43 |
| CrossMNA | 0.220 | **0.222** | +0.002 | 298 | **210** | -88 | 1.42 |
| NetTrans | 0.373 | **0.374** | +0.001 | 919 | **817** | -102 | 1.12 |
| WAlign | **0.271** | 0.270 | -0.001 | 79 | **68** | -11 | 1.16 |
| BRIGHT | 0.362 | **0.362** | +0.000 | 768 | **619** | -149 | 1.24 |
| NeXtAlign | 0.391 | **0.391** | +0.000 | 1319 | **1234** | -85 | 1.07 |
| WLAlign | **0.328** | 0.322 | -0.006 | 4018 | **1276** | -2742 | 3.15 |
| SLOTAlign | **0.200** | 0.200 | -0.001 | 891 | **821** | -70 | 1.09 |
| HOT | **0.173** | 0.172 | -0.001 | 239 | **226** | -13 | 1.06 |
| JOENA | **0.583** | 0.583 | +0.000 | 691 | **679** | -12 | 1.02 |

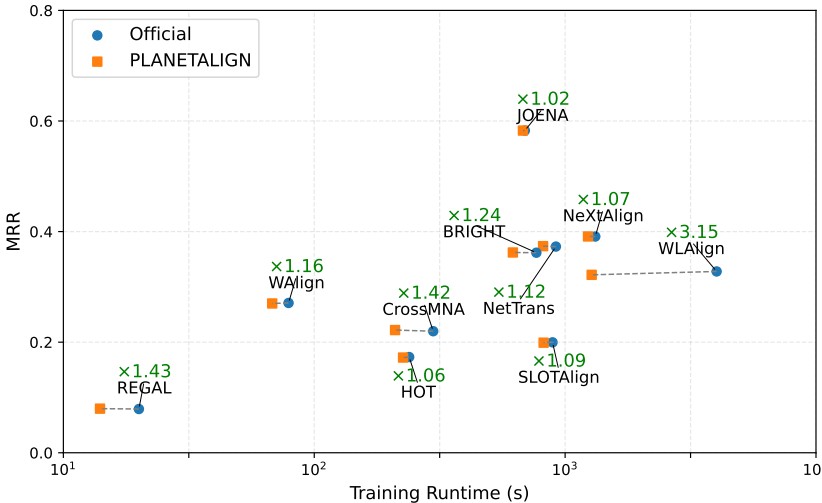

Figure 7: Performance and runtime comparison between official and PLANETALIGN's implementations. The **x-axis** shows the average training runtime, and the **y-axis** shows the average MRR of different NA algorithms across 18 datasets. The average speedup in runtime of each method are shown in green.

We conduct comparative experiments between the official and PLANETALIGN's implementations of NA algorithms to validate the correctness and efficiency of PLANETALIGN. To ensure a fair comparison, we only include algorithms that have official Python implementations to eliminate the efficiency discrepancy of different programming languages. All training parameters, including training epochs, are set as default in the official code. We report the average MRR and training

runtime of official and PLANETALIGN's implementation across all 18 datasets in PLANETALIGN in Table 5 and Figure 7

We can see that PLANETALIGN's implementation show comparable performance across all baselines while achieving up to 3 times speed-up over official implementations, demonstrating the correctness and efficiency of our implementation of existing NA methods.

## D.4 SCALABILITY RESULTS ON LARGE GRAPHS

To further demonstrate the efficiency of our implementation on large-scale networks, we compare the training runtime with the official implementation on ER networks of 50K, 75K, and 100K nodes with an average node degree of 5 per network. As we can see in Table 6, PLANETALIGN consistently outperform official implementations with up to 2.7 times speed-up.

Additionally, we can also see from Table 6 that most existing NA methods cannot scale to million-node networks due to their intrinsic complexity constraints, highlighting more efficient NA methods as an important line of future works. PLANETALIGN intentionally preserves the theoretical complexity of existing methods to ensure fair and consistent benchmarking.

Table 6: Runtime (s) comparison with official implementations on large-scale ER networks of 50K, 75K, and 100K nodes with average node degrees of 5. OOM represents out-of-memory.

| # Nodes | 50K | | | 75K | | | 100K | | |
|---|---|---|---|---|---|---|---|---|---|
| Version | Official | PLANETALIGN | Speedup | Official | PLANETALIGN | Speedup | Official | PLANETALIGN | Speedup |
| REGAL | 334 | **214** | 1.56 | 583 | **390** | 1.49 | $1.12\times10^3$ | **731** | 1.53 |
| CrossMNA | $4.12\times10^3$ | $\mathbf{3.02\times10^3}$ | 1.36 | $7.36\times10^3$ | $\mathbf{5.13\times10^3}$ | 1.43 | $1.37\times10^4$ | $\mathbf{8.91\times10^3}$ | 1.54 |
| WAlign | $1.65\times10^3$ | $\mathbf{1.29\times10^3}$ | 1.28 | $3.12\times10^3$ | $\mathbf{2.54\times10^3}$ | 1.23 | OOM | OOM | OOM |
| BRIGHT | $2.82\times10^4$ | $\mathbf{2.15\times10^4}$ | 1.31 | $8.95\times10^4$ | $\mathbf{6.90\times10^4}$ | 1.29 | $3.47\times10^5$ | $\mathbf{2.49\times10^5}$ | 1.40 |
| NeXtAlign | $6.55\times10^4$ | $\mathbf{6.08\times10^4}$ | 1.08 | $3.68\times10^5$ | $\mathbf{3.41\times10^5}$ | 1.08 | OOM | OOM | OOM |
| WLAlign | $2.46\times10^5$ | $\mathbf{9.12\times10^4}$ | 2.70 | OOM | OOM | OOM | OOM | OOM | OOM |
| SLOTAlign | $3.01\times10^4$ | $\mathbf{2.75\times10^4}$ | 1.10 | OOM | OOM | OOM | OOM | OOM | OOM |
| HOT | $3.14\times10^3$ | $\mathbf{2.87\times10^3}$ | 1.10 | $6.17\times10^3$ | $\mathbf{5.74\times10^3}$ | 1.07 | $1.08\times10^4$ | $\mathbf{9.43\times10^3}$ | 1.15 |
| JOENA | $2.31\times10^3$ | $\mathbf{2.28\times10^4}$ | 1.01 | $8.30\times10^4$ | $\mathbf{8.12\times10^4}$ | 1.02 | OOM | OOM | OOM |

## D.5 DETAILED EFFECTIVENESS RESULTS

Detailed effectiveness results on *plain* networks with a training ratio of 20% are shown in Table 7 and 8. Detailed effectiveness results on *attributed* networks with a training ratio of 20% are shown in Table 9.

## E STATEMENT OF LLM USAGE

In this paper, LLMs were used exclusively for formatting assistance and language polishing. At no point were LLMs involved significantly in research ideation and/or writing to the extent that they could be considered as a contributor. Therefore, the use of LLMs does not impact the core methodology, the scientific rigorousness, or the originality of this research.

Table 7: Detailed effectiveness results (Part I of II) on plain networks with a training ratio of 20%. The 1st/2nd/3rd best results are marked in red/blue/green respectively. Time denotes the inference time and Mem. denotes the peak memory usage.

| Dataset | Foursquare-Twitter | | | | | Phone-Email | | | | |
|---|---|---|---|---|---|---|---|---|---|---|
| Metrics | Hits@1 | Hits@10 | MRR | Time(s) | Mem.(GB) | Hits@1 | Hits@10 | MRR | Time(s) | Mem.(GB) |
| IsoRank | $.0241_{\pm.0000}$ | $.1487_{\pm.0000}$ | $.0645_{\pm.0000}$ | $5.70_{\pm.60}$ | $1.60_{\pm.10}$ | $.0431_{\pm.0000}$ | $.2156_{\pm.0000}$ | $.1100_{\pm.0000}$ | $0.25_{\pm.12}$ | $0.59_{\pm.00}$ |
| FINAL | $.0474_{\pm.0000}$ | $.2407_{\pm.0000}$ | $.1062_{\pm.0000}$ | $0.96_{\pm.03}$ | $2.30_{\pm.00}$ | $.0494_{\pm.0000}$ | $.2725_{\pm.0000}$ | $.1257_{\pm.0000}$ | $0.10_{\pm.04}$ | $0.63_{\pm.01}$ |
| IONE | $.0202_{\pm.0052}$ | $.0985_{\pm.0051}$ | $.0481_{\pm.0045}$ | $1.10_{\pm.02}\times10^4$ | $1.20_{\pm.00}$ | $.0941_{\pm.0032}$ | $.4037_{\pm.0120}$ | $.1952_{\pm.0031}$ | $2.30_{\pm.06}\times10^3$ | $0.80_{\pm.00}$ |
| REGAL | $.0001_{\pm.0002}$ | $.0027_{\pm.0010}$ | $.0025_{\pm.0003}$ | $7.80_{\pm.10}$ | $1.10_{\pm.00}$ | $.0012_{\pm.0000}$ | $.0097_{\pm.0011}$ | $.0076_{\pm.0003}$ | $1.30_{\pm.20}$ | $0.60_{\pm.00}$ |
| CrossMNA | $.0162_{\pm.0034}$ | $.1011_{\pm.0061}$ | $.0456_{\pm.0039}$ | $1.10_{\pm.06}\times10^3$ | $1.00_{\pm.00}$ | $.0305_{\pm.0029}$ | $.2163_{\pm.0086}$ | $.0968_{\pm.0021}$ | $1.10_{\pm.27}\times10^3$ | $0.62_{\pm.00}$ |
| NetTrans | $.0809_{\pm.0043}$ | $.2470_{\pm.0074}$ | $.1347_{\pm.0048}$ | $5.90_{\pm.86}\times10^2$ | $3.50_{\pm1.20}$ | $.0216_{\pm.0027}$ | $.2546_{\pm.0020}$ | $.1020_{\pm.0016}$ | $2.40_{\pm1.70}\times10^1$ | $0.85_{\pm.02}$ |
| WAlign | $.0039_{\pm.0004}$ | $.0150_{\pm.0005}$ | $.0095_{\pm.0003}$ | $1.30_{\pm.04}$ | $2.30_{\pm.00}$ | $.0206_{\pm.0018}$ | $.1235_{\pm.0093}$ | $.0585_{\pm.0009}$ | $0.29_{\pm.08}$ | $0.89_{\pm.01}$ |
| BRIGHT | $.0537_{\pm.0027}$ | $.1784_{\pm.0012}$ | $.0923_{\pm.0019}$ | $9.20_{\pm.20}$ | $1.40_{\pm.00}$ | $.0476_{\pm.0033}$ | $.2516_{\pm.0028}$ | $.1186_{\pm.0029}$ | $0.29_{\pm.01}$ | $0.95_{\pm.01}$ |
| NeXtAlign | $.0387_{\pm.0040}$ | $.1420_{\pm.0163}$ | $.0745_{\pm.0075}$ | $7.90_{\pm.25}\times10^1$ | $2.30_{\pm.00}$ | $.0570_{\pm.0045}$ | $.3012_{\pm.0116}$ | $.1411_{\pm.0036}$ | $4.60_{\pm.10}$ | $0.86_{\pm.01}$ |
| WLAlign | $.0924_{\pm.0016}$ | $.2103_{\pm.0036}$ | $.1325_{\pm.0009}$ | $1.20_{\pm.14}\times10^3$ | $3.00_{\pm.00}$ | $.0764_{\pm.0012}$ | $.2669_{\pm.0047}$ | $.1412_{\pm.0014}$ | $7.50_{\pm1.10}\times10^2$ | $1.00_{\pm.00}$ |
| PARROT | $.1203_{\pm.0000}$ | $.2908_{\pm.0000}$ | $.1776_{\pm.0000}$ | $1.60_{\pm.02}\times10^1$ | $2.60_{\pm.10}$ | $.2887_{\pm.0000}$ | $.7331_{\pm.0000}$ | $.4374_{\pm.0000}$ | $0.76_{\pm.09}$ | $0.69_{\pm.00}$ |
| SLOTAlign | $.0291_{\pm.0000}$ | $.1172_{\pm.0000}$ | $.0614_{\pm.0000}$ | $1.60_{\pm.03}\times10^2$ | $2.50_{\pm.00}$ | $.0075_{\pm.0000}$ | $.0525_{\pm.0000}$ | $.0283_{\pm.0000}$ | $2.34_{\pm.30}$ | $0.75_{\pm.01}$ |
| HOT | $.0518_{\pm.0030}$ | $.1627_{\pm.0044}$ | $.0457_{\pm.0018}$ | $1.10_{\pm.21}\times10^2$ | $1.70_{\pm.10}$ | $.0775_{\pm.0025}$ | $.3273_{\pm.0069}$ | $.0801_{\pm.0020}$ | $5.30_{\pm.60}$ | $0.72_{\pm.01}$ |
| JOENA | $.2673_{\pm.0069}$ | $.4478_{\pm.0083}$ | $.3304_{\pm.0083}$ | $2.80_{\pm.05}\times10^1$ | $2.50_{\pm.06}$ | $.3468_{\pm.0029}$ | $.7809_{\pm.0037}$ | $.4973_{\pm.0018}$ | $0.76_{\pm.08}$ | $0.72_{\pm.01}$ |

| Dataset | ACM-DBLP | | | | | SacchCere1-SacchCere2 | | | | |
|---|---|---|---|---|---|---|---|---|---|---|
| Metrics | Hits@1 | Hits@10 | MRR | Time(s) | Mem.(GB) | Hits@1 | Hits@10 | MRR | Time(s) | Mem.(GB) |
| IsoRank | $.1641_{\pm.0003}$ | $.6329_{\pm.0008}$ | $.3023_{\pm.0002}$ | $4.70_{\pm1.70}\times10^1$ | $4.30_{\pm.10}$ | $.0335_{\pm.0009}$ | $.2284_{\pm.0016}$ | $.0976_{\pm.0002}$ | $0.95_{\pm.10}$ | $1.50_{\pm.60}$ |
| FINAL | $.2082_{\pm.0000}$ | $.6893_{\pm.0000}$ | $.3612_{\pm.0000}$ | $3.90_{\pm.10}$ | $6.60_{\pm.10}$ | $.0467_{\pm.0000}$ | $.2379_{\pm.0000}$ | $.1083_{\pm.0000}$ | $0.54_{\pm.03}$ | $1.70_{\pm.60}$ |
| IONE | $.2515_{\pm.0028}$ | $.7267_{\pm.0075}$ | $.3979_{\pm.0023}$ | $1.30_{\pm.05}\times10^4$ | $2.60_{\pm.13}$ | $.0458_{\pm.0037}$ | $.2100_{\pm.0055}$ | $.0992_{\pm.0016}$ | $8.30_{\pm.04}\times10^3$ | $1.30_{\pm.60}$ |
| REGAL | $.0357_{\pm.0022}$ | $.1367_{\pm.0035}$ | $.0700_{\pm.0030}$ | $1.30_{\pm.10}\times10^1$ | $1.90_{\pm.00}$ | $.0023_{\pm.0008}$ | $.0087_{\pm.0007}$ | $.0063_{\pm.0006}$ | $3.40_{\pm.10}$ | $1.20_{\pm.00}$ |
| CrossMNA | $.0742_{\pm.0034}$ | $.6108_{\pm.0031}$ | $.2290_{\pm.0025}$ | $5.50_{\pm.13}\times10^2$ | $1.90_{\pm.00}$ | $.0046_{\pm.0007}$ | $.1452_{\pm.0041}$ | $.0492_{\pm.0023}$ | $1.80_{\pm.13}\times10^2$ | $0.98_{\pm.00}$ |
| NetTrans | $.4148_{\pm.0018}$ | $.8107_{\pm.0009}$ | $.5429_{\pm.0012}$ | $1.10_{\pm.34}\times10^2$ | $1.50_{\pm.50}\times10^1$ | $.0523_{\pm.0017}$ | $.2534_{\pm.0038}$ | $.1150_{\pm.0020}$ | $1.20_{\pm.30}\times10^1$ | $2.20_{\pm.20}$ |
| WAlign | $.2871_{\pm.0018}$ | $.5538_{\pm.0065}$ | $.3797_{\pm.0021}$ | $2.80_{\pm.30}$ | $4.90_{\pm.10}$ | $.0207_{\pm.0006}$ | $.0516_{\pm.0032}$ | $.0336_{\pm.0019}$ | $0.28_{\pm.01}$ | $1.90_{\pm.00}$ |
| BRIGHT | $.4052_{\pm.0011}$ | $.7957_{\pm.0011}$ | $.5346_{\pm.0013}$ | $6.80_{\pm.10}\times10^1$ | $3.80_{\pm.00}$ | $.0353_{\pm.0015}$ | $.2188_{\pm.0062}$ | $.0915_{\pm.0021}$ | $4.60_{\pm.00}$ | $1.60_{\pm.00}$ |
| NeXtAlign | $.4670_{\pm.0019}$ | $.8401_{\pm.0017}$ | $.5915_{\pm.0011}$ | $1.70_{\pm.04}\times10^2$ | $4.00_{\pm.00}$ | $.0292_{\pm.0013}$ | $.2075_{\pm.0075}$ | $.0886_{\pm.0040}$ | $1.50_{\pm.40}$ | $1.30_{\pm.10}$ |
| WLAlign | $.3152_{\pm.0012}$ | $.6446_{\pm.0008}$ | $.4183_{\pm.0008}$ | $1.40_{\pm.04}\times10^3$ | $7.00_{\pm.00}$ | $.0535_{\pm.0011}$ | $.1639_{\pm.0014}$ | $.0888_{\pm.0007}$ | $5.80_{\pm.99}\times10^2$ | $1.50_{\pm.00}$ |
| PARROT | $.5749_{\pm.0000}$ | $.8784_{\pm.0000}$ | $.6766_{\pm.0000}$ | $1.30_{\pm.04}\times10^2$ | $7.80_{\pm.00}$ | $.0645_{\pm.0000}$ | $.2720_{\pm.0000}$ | $.1312_{\pm.0000}$ | $5.90_{\pm.20}$ | $1.90_{\pm.60}$ |
| SLOTAlign | $.4914_{\pm.0000}$ | $.7174_{\pm.0000}$ | $.5707_{\pm.0000}$ | $9.90_{\pm.16}\times10^2$ | $7.60_{\pm.26}$ | $.0000_{\pm.0000}$ | $.0023_{\pm.0000}$ | $.0028_{\pm.0000}$ | $1.00_{\pm.00}$ | $2.10_{\pm.10}$ |
| HOT | $.3261_{\pm.0040}$ | $.6787_{\pm.0053}$ | $.2210_{\pm.0026}$ | $4.30_{\pm.15}\times10^2$ | $5.00_{\pm.00}$ | $.0344_{\pm.0023}$ | $.1993_{\pm.0039}$ | $.0564_{\pm.0012}$ | $1.20_{\pm.11}\times10^3$ | $1.50_{\pm.10}$ |
| JOENA | $.6149_{\pm.0458}$ | $.9062_{\pm.0219}$ | $.7136_{\pm.0389}$ | $3.50_{\pm.50}\times10^1$ | $3.00_{\pm2.00}$ | $.0589_{\pm.0026}$ | $.2284_{\pm.0062}$ | $.1129_{\pm.0027}$ | $0.31_{\pm.01}$ | $1.10_{\pm.00}$ |

| Dataset | DBP15K_ZH-EN | | | | | Italy1-Italy2 | | | | |
|---|---|---|---|---|---|---|---|---|---|---|
| Metrics | Hits@1 | Hits@10 | MRR | Time(s) | Mem.(GB) | Hits@1 | Hits@10 | MRR | Time(s) | Mem.(GB) |
| IsoRank | $.1092_{\pm.0001}$ | $.4878_{\pm.0001}$ | $.2276_{\pm.0000}$ | $1.60_{\pm.16}\times10^2$ | $1.60_{\pm.00}\times10^1$ | $.0424_{\pm.0009}$ | $.2076_{\pm.0009}$ | $.0954_{\pm.0003}$ | $0.49_{\pm.13}$ | $0.60_{\pm.00}$ |
| FINAL | $.1327_{\pm.0000}$ | $.5347_{\pm.0000}$ | $.2634_{\pm.0000}$ | $1.80_{\pm.00}\times10^1$ | $2.40_{\pm.00}\times10^1$ | $.0430_{\pm.0000}$ | $.2202_{\pm.0000}$ | $.0958_{\pm.0000}$ | $0.13_{\pm.01}$ | $0.84_{\pm.00}$ |
| IONE | $.0633_{\pm.0033}$ | $.2512_{\pm.0080}$ | $.1258_{\pm.0047}$ | $1.50_{\pm.03}\times10^4$ | $8.70_{\pm.00}$ | $.0245_{\pm.0027}$ | $.1570_{\pm.0149}$ | $.0679_{\pm.0028}$ | $6.20_{\pm.03}\times10^3$ | $0.84_{\pm.00}$ |
| REGAL | $.0036_{\pm.0005}$ | $.0168_{\pm.0008}$ | $.0091_{\pm.0006}$ | $3.00_{\pm.10}\times10^1$ | $5.90_{\pm.00}$ | $.0033_{\pm.0020}$ | $.0265_{\pm.0041}$ | $.0142_{\pm.0023}$ | $0.97_{\pm.00}$ | $0.81_{\pm.00}$ |
| CrossMNA | $.0303_{\pm.0012}$ | $.2816_{\pm.0046}$ | $.1077_{\pm.0017}$ | $9.90_{\pm.26}\times10^2$ | $6.00_{\pm.00}$ | $.0023_{\pm.0015}$ | $.1447_{\pm.0105}$ | $.0476_{\pm.0024}$ | $3.70_{\pm.70}\times10^1$ | $0.98_{\pm.01}$ |
| NetTrans | $.2625_{\pm.0011}$ | $.6022_{\pm.0010}$ | $.3717_{\pm.0009}$ | $5.00_{\pm.22}\times10^2$ | $3.00_{\pm1.30}\times10^1$ | $.0503_{\pm.0019}$ | $.2248_{\pm.0027}$ | $.1110_{\pm.0020}$ | $0.62_{\pm.15}$ | $1.10_{\pm.00}$ |
| WAlign | $.1856_{\pm.0103}$ | $.2823_{\pm.0170}$ | $.2231_{\pm.0127}$ | $9.10_{\pm.44}$ | $1.50_{\pm.00}\times10^1$ | $.0609_{\pm.0014}$ | $.1580_{\pm.0034}$ | $.0938_{\pm.0020}$ | $0.09_{\pm.00}$ | $1.10_{\pm.00}$ |
| BRIGHT | $.2715_{\pm.0007}$ | $.5938_{\pm.0015}$ | $.3789_{\pm.0007}$ | $3.20_{\pm.03}\times10^2$ | $1.10_{\pm.00}\times10^1$ | $.0904_{\pm.0025}$ | $.2566_{\pm.0045}$ | $.1443_{\pm.0010}$ | $0.33_{\pm.03}$ | $1.00_{\pm.00}$ |
| NeXtAlign | $.2695_{\pm.0098}$ | $.5981_{\pm.0095}$ | $.3790_{\pm.0104}$ | $2.60_{\pm.03}\times10^3$ | $1.30_{\pm.10}\times10^1$ | $.0861_{\pm.0048}$ | $.2580_{\pm.0029}$ | $.1466_{\pm.0045}$ | $0.11_{\pm.04}$ | $1.20_{\pm.00}$ |
| WLAlign | $.2349_{\pm.0006}$ | $.4122_{\pm.0006}$ | $.2911_{\pm.0001}$ | $2.90_{\pm.26}\times10^3$ | $2.70_{\pm.00}\times10^1$ | $.0404_{\pm.0048}$ | $.1536_{\pm.0051}$ | $.0778_{\pm.0037}$ | $2.40_{\pm.86}\times10^2$ | $1.40_{\pm.00}$ |
| PARROT | $.6334_{\pm.0000}$ | $.8528_{\pm.0000}$ | $.7074_{\pm.0000}$ | $8.70_{\pm.10}\times10^2$ | $2.80_{\pm.00}\times10^1$ | $.0993_{\pm.0000}$ | $.2848_{\pm.0000}$ | $.1655_{\pm.0000}$ | $0.90_{\pm.00}$ | $0.83_{\pm.00}$ |
| SLOTAlign | $.0188_{\pm.0000}$ | $.0725_{\pm.0000}$ | $.0381_{\pm.0000}$ | $3.06_{\pm.01}\times10^4$ | $2.80_{\pm.00}\times10^1$ | $.0149_{\pm.0000}$ | $.0613_{\pm.0000}$ | $.0334_{\pm.0000}$ | $0.12_{\pm.00}$ | $1.00_{\pm.20}$ |
| HOT | $.3143_{\pm.0038}$ | $.5832_{\pm.0058}$ | $.2006_{\pm.0020}$ | $2.10_{\pm.08}\times10^3$ | $2.00_{\pm.00}\times10^1$ | $.0639_{\pm.0114}$ | $.2408_{\pm.0306}$ | $.0586_{\pm.0046}$ | $1.10_{\pm.10}\times10^1$ | $2.10_{\pm.00}$ |
| JOENA | $.6476_{\pm.0037}$ | $.8496_{\pm.0034}$ | $.7170_{\pm.0037}$ | $6.70_{\pm.08}\times10^2$ | $2.60_{\pm.00}\times10^1$ | $.1010_{\pm.0049}$ | $.2930_{\pm.0161}$ | $.1697_{\pm.0061}$ | $0.06_{\pm.00}$ | $1.00_{\pm.10}$ |

| Dataset | Douban | | | | | Flickr-LastFM | | | | |
|---|---|---|---|---|---|---|---|---|---|---|
| Metrics | Hits@1 | Hits@10 | MRR | Time(s) | Mem.(GB) | Hits@1 | Hits@10 | MRR | Time(s) | Mem.(GB) |
| IsoRank | $.1351_{\pm.0003}$ | $.5179_{\pm.0004}$ | $.2517_{\pm.0001}$ | $0.59_{\pm.06}$ | $0.95_{\pm.03}$ | $.0028_{\pm.0000}$ | $.0815_{\pm.0000}$ | $.0305_{\pm.0000}$ | $9.20_{\pm.22}\times10^1$ | $8.13_{\pm.21}$ |
| FINAL | $.1458_{\pm.0000}$ | $.5676_{\pm.0000}$ | $.2692_{\pm.0000}$ | $0.34_{\pm.03}$ | $1.14_{\pm.04}$ | $.0028_{\pm.0000}$ | $.0732_{\pm.0000}$ | $.0253_{\pm.0000}$ | $1.56_{\pm.04}\times10^1$ | $1.29_{\pm.02}\times10^1$ |
| IONE | $.2777_{\pm.0046}$ | $.6312_{\pm.0098}$ | $.3936_{\pm.0040}$ | $9.08_{\pm1.57}\times10^3$ | $0.89_{\pm.04}$ | $.0113_{\pm.0033}$ | $.0437_{\pm.0044}$ | $.0239_{\pm.0036}$ | $4.09_{\pm1.41}\times10^3$ | $3.77_{\pm.21}$ |
| REGAL | $.0025_{\pm.0000}$ | $.0198_{\pm.0034}$ | $.0105_{\pm.0006}$ | $2.57_{\pm.19}$ | $0.82_{\pm.00}$ | $.0042_{\pm.0026}$ | $.0340_{\pm.0062}$ | $.0177_{\pm.0023}$ | $1.78_{\pm.06}\times10^1$ | $1.65_{\pm.16}$ |
| CrossMNA | $.0187_{\pm.0044}$ | $.3241_{\pm.0054}$ | $.1172_{\pm.0049}$ | $6.34_{\pm.24}\times10^1$ | $0.78_{\pm.01}$ | $.0061_{\pm.0008}$ | $.0091_{\pm.0030}$ | $.0078_{\pm.0011}$ | $3.18_{\pm3.40}\times10^1$ | $2.46_{\pm.01}$ |
| NetTrans | $.2030_{\pm.0020}$ | $.6018_{\pm.0020}$ | $.3291_{\pm.0015}$ | $1.54_{\pm.44}$ | $1.92_{\pm.40}$ | $.0052_{\pm.0012}$ | $.0204_{\pm.0018}$ | $.0115_{\pm.0009}$ | $2.75_{\pm2.01}\times10^1$ | $2.27_{\pm1.15}\times10^1$ |
| WAlign | $.1480_{\pm.0018}$ | $.2381_{\pm.0032}$ | $.1834_{\pm.0019}$ | $0.20_{\pm.06}$ | $1.31_{\pm.01}$ | $.0094_{\pm.0041}$ | $.0423_{\pm.0021}$ | $.0290_{\pm.0010}$ | $0.55_{\pm.03}$ | $4.51_{\pm.08}$ |
| BRIGHT | $.1202_{\pm.0007}$ | $.4361_{\pm.0031}$ | $.2218_{\pm.0013}$ | $2.64_{\pm.07}$ | $1.14_{\pm.01}$ | $.0259_{\pm.0015}$ | $.0492_{\pm.0037}$ | $.0357_{\pm.0010}$ | $5.44_{\pm.14}\times10^1$ | $6.14_{\pm.04}$ |
| NeXtAlign | $.2154_{\pm.0062}$ | $.5701_{\pm.0144}$ | $.3305_{\pm.0084}$ | $1.86_{\pm.08}$ | $1.66_{\pm.00}$ | $.0260_{\pm.0030}$ | $.0541_{\pm.0075}$ | $.0375_{\pm.0023}$ | $6.11_{\pm.29}\times10^1$ | $6.85_{\pm.07}$ |
| WLAlign | $.2028_{\pm.0021}$ | $.3505_{\pm.0019}$ | $.2517_{\pm.0015}$ | $6.58_{\pm.41}\times10^2$ | $0.95_{\pm.02}$ | $.0080_{\pm.0027}$ | $.0224_{\pm.0012}$ | $.0141_{\pm.0018}$ | $5.37_{\pm3.57}\times10^2$ | $3.21_{\pm.02}$ |
| PARROT | $.3469_{\pm.0000}$ | $.6832_{\pm.0000}$ | $.4563_{\pm.0000}$ | $2.73_{\pm.71}$ | $1.31_{\pm.02}$ | $.0276_{\pm.0000}$ | $.0608_{\pm.0000}$ | $.0417_{\pm.0000}$ | $2.51_{\pm.08}\times10^2$ | $1.51_{\pm.02}\times10^1$ |
| SLOTAlign | $.0000_{\pm.0000}$ | $.0078_{\pm.0000}$ | $.0048_{\pm.0000}$ | $7.01_{\pm.32}$ | $1.40_{\pm.03}$ | $.0041_{\pm.0000}$ | $.0235_{\pm.0000}$ | $.0145_{\pm.0000}$ | $8.44_{\pm4.05}$ | $1.54_{\pm.04}\times10^1$ |
| HOT | $.1509_{\pm.0059}$ | $.4600_{\pm.0131}$ | $.1545_{\pm.0026}$ | $3.95_{\pm1.54}\times10^1$ | $1.37_{\pm.00}$ | $.0091_{\pm.0037}$ | $.0157_{\pm.0036}$ | $.0063_{\pm.0015}$ | $1.01_{\pm.02}\times10^1$ | $7.97_{\pm.18}$ |
| JOENA | $.4457_{\pm.0038}$ | $.8091_{\pm.0026}$ | $.5657_{\pm.0022}$ | $0.41_{\pm.06}$ | $1.19_{\pm.03}$ | $.0290_{\pm.0000}$ | $.0994_{\pm.0062}$ | $.0558_{\pm.0023}$ | $2.02_{\pm.07}\times10^2$ | $1.41_{\pm.02}\times10^1$ |

| Dataset | Flickr-MySpace | | | | | Arenas | | | | |
|---|---|---|---|---|---|---|---|---|---|---|
| Metrics | Hits@1 | Hits@10 | MRR | Time(s) | Mem.(GB) | Hits@1 | Hits@10 | MRR | Time(s) | Mem.(GB) |
| IsoRank | $.0047_{\pm.0000}$ | $.0374_{\pm.0000}$ | $.0225_{\pm.0002}$ | $2.39_{\pm2.35}$ | $3.47_{\pm.04}$ | $.3981_{\pm.0000}$ | $.6751_{\pm.0000}$ | $.4940_{\pm.0000}$ | $0.09_{\pm.00}$ | $1.02_{\pm.63}$ |
| FINAL | $.0000_{\pm.0000}$ | $.0093_{\pm.0000}$ | $.0048_{\pm.0000}$ | $6.74_{\pm.28}$ | $5.21_{\pm.01}$ | $.3849_{\pm.0000}$ | $.8811_{\pm.0000}$ | $.5523_{\pm.0000}$ | $0.11_{\pm.01}$ | $1.13_{\pm.63}$ |
| IONE | $.0056_{\pm.0027}$ | $.0266_{\pm.0058}$ | $.0170_{\pm.0023}$ | $1.19_{\pm.64}\times10^3$ | $1.88_{\pm.07}$ | $.9130_{\pm.0069}$ | $.9763_{\pm.0042}$ | $.9393_{\pm.0060}$ | $1.39_{\pm.14}\times10^4$ | $1.25_{\pm.63}$ |
| REGAL | $.0061_{\pm.0013}$ | $.0322_{\pm.0042}$ | $.0173_{\pm.0017}$ | $9.36_{\pm.10}$ | $1.06_{\pm.14}$ | $.9053_{\pm.0052}$ | $.9803_{\pm.0008}$ | $.9354_{\pm.0016}$ | $1.04_{\pm.02}$ | $0.94_{\pm.11}$ |
| CrossMNA | $.0070_{\pm.0037}$ | $.0107_{\pm.0021}$ | $.0090_{\pm.0032}$ | $2.93_{\pm3.03}\times10^1$ | $1.36_{\pm.01}$ | $.4261_{\pm.0175}$ | $.8492_{\pm.0120}$ | $.5812_{\pm.0158}$ | $5.61_{\pm.10}\times10^1$ | $1.00_{\pm.00}$ |
| NetTrans | $.0000_{\pm.0000}$ | $.0009_{\pm.0013}$ | $.0010_{\pm.0002}$ | $5.42_{\pm3.96}$ | $7.38_{\pm3.86}$ | $.8827_{\pm.0013}$ | $.9983_{\pm.0004}$ | $.9348_{\pm.0010}$ | $0.74_{\pm.05}$ | $2.22_{\pm.43}$ |
| WAlign | $.0056_{\pm.0013}$ | $.0444_{\pm.0016}$ | $.0240_{\pm.0008}$ | $0.40_{\pm.05}$ | $2.49_{\pm.04}$ | $.9723_{\pm.0003}$ | $.9974_{\pm.0003}$ | $.9837_{\pm.0002}$ | $0.12_{\pm.00}$ | $1.34_{\pm.15}$ |
| BRIGHT | $.0037_{\pm.0027}$ | $.0154_{\pm.0043}$ | $.0089_{\pm.0012}$ | $2.08_{\pm.04}\times10^1$ | $3.32_{\pm.07}$ | $.9700_{\pm.0005}$ | $.9954_{\pm.0006}$ | $.9817_{\pm.0003}$ | $0.38_{\pm.01}$ | $1.29_{\pm.03}$ |
| NeXtAlign | $.0047_{\pm.0017}$ | $.0140_{\pm.0024}$ | $.0089_{\pm.0014}$ | $2.16_{\pm.06}\times10^1$ | $4.20_{\pm.01}$ | $.5347_{\pm.0412}$ | $.7269_{\pm.0301}$ | $.6034_{\pm.0089}$ | $0.28_{\pm.02}$ | $1.11_{\pm.08}$ |
| WLAlign | $.0000_{\pm.0000}$ | $.0070_{\pm.0055}$ | $.0043_{\pm.0021}$ | $5.67_{\pm3.01}\times10^2$ | $1.51_{\pm.02}$ | $.6051_{\pm.0022}$ | $.7152_{\pm.0010}$ | $.6491_{\pm.0009}$ | $6.47_{\pm.89}\times10^2$ | $0.89_{\pm.07}$ |
| PARROT | $.0093_{\pm.0000}$ | $.0164_{\pm.0000}$ | $.0114_{\pm.0000}$ | $3.73_{\pm.08}\times10^1$ | $6.04_{\pm.06}$ | $.9780_{\pm.0000}$ | $.9999_{\pm.0000}$ | $.9886_{\pm.0000}$ | $0.88_{\pm.05}$ | $1.10_{\pm.63}$ |
| SLOTAlign | $.0021_{\pm.0000}$ | $.0135_{\pm.0000}$ | $.0075_{\pm.0000}$ | $9.78_{\pm.13}$ | $6.31_{\pm.04}$ | $.9785_{\pm.0000}$ | $.9999_{\pm.0000}$ | $.9889_{\pm.0000}$ | $0.12_{\pm.00}$ | $0.86_{\pm.01}$ |
| HOT | $.0014_{\pm.0013}$ | $.0033_{\pm.0039}$ | $.0005_{\pm.0005}$ | $4.46_{\pm.44}\times10^2$ | $4.37_{\pm.23}$ | $.9653_{\pm.0048}$ | $.9958_{\pm.0020}$ | $.4901_{\pm.0020}$ | $8.78_{\pm1.28}$ | $8.19_{\pm1.36}$ |
| JOENA | $.0071_{\pm.0000}$ | $.0476_{\pm.0035}$ | $.0261_{\pm.0006}$ | $4.49_{\pm.07}$ | $1.79_{\pm.11}$ | $.9795_{\pm.0006}$ | $.9999_{\pm.0000}$ | $.9894_{\pm.0003}$ | $0.22_{\pm.35}$ | $1.01_{\pm.00}$ |

Table 8: Detailed effectiveness results (Part II of II) on plain networks with a training ratio of 20%. The 1st/2nd/3rd best results are marked in red/blue/green respectively. Time denotes the inference time and Mem. denotes the peak memory usage.

| Dataset | Cora | | | | | ArXiv | | | | |
|---|---|---|---|---|---|---|---|---|---|---|
| Metrics | Hits@1 | Hits@10 | MRR | Time(s) | Mem.(GB) | Hits@1 | Hits@10 | MRR | Time(s) | Mem.(GB) |
| IsoRank | $.1793_{\pm.0000}$ | $.5877_{\pm.0000}$ | $.3127_{\pm.0001}$ | $0.98_{\pm0.30}$ | $1.17_{\pm0.15}$ | $.2247_{\pm.0001}$ | $.5537_{\pm.0000}$ | $.3281_{\pm.0000}$ | $1.26_{\pm0.17}\times10^{2}$ | $1.52_{\pm0.06}\times10^{1}$ |
| FINAL | $.2065_{\pm.0000}$ | $.6442_{\pm.0000}$ | $.3488_{\pm.0000}$ | $0.25_{\pm.01}$ | $1.38_{\pm0.15}$ | $.2551_{\pm.0000}$ | $.7250_{\pm.0000}$ | $.4102_{\pm.0000}$ | $1.61_{\pm0.02}\times10^{1}$ | $2.22_{\pm0.06}\times10^{1}$ |
| IONE | $.3415_{\pm.0080}$ | $.6611_{\pm.0079}$ | $.4536_{\pm.0078}$ | $1.34_{\pm0.04}\times10^{4}$ | $1.08_{\pm0.15}$ | $.2688_{\pm.0086}$ | $.5195_{\pm.0093}$ | $.3517_{\pm.0091}$ | $1.66_{\pm0.17}\times10^{4}$ | $8.79_{\pm0.57}$ |
| REGAL | $.0158_{\pm.0019}$ | $.0789_{\pm.0051}$ | $.0383_{\pm.0032}$ | $2.41_{\pm0.07}$ | $0.86_{\pm0.02}$ | $.0026_{\pm.0003}$ | $.0185_{\pm.0012}$ | $.0098_{\pm.0005}$ | $3.30_{\pm0.22}\times10^{1}$ | $6.79_{\pm0.10}$ |
| CrossMNA | $.0358_{\pm.0045}$ | $.4574_{\pm.0080}$ | $.1740_{\pm.0052}$ | $5.80_{\pm0.17}\times10^{1}$ | $0.89_{\pm0.01}$ | $.2875_{\pm.0015}$ | $.6742_{\pm.0007}$ | $.4115_{\pm.0012}$ | $2.87_{\pm0.06}\times10^{3}$ | $6.68_{\pm0.06}$ |
| NetTrans | $.3703_{\pm.0023}$ | $.7238_{\pm.0017}$ | $.4891_{\pm.0020}$ | $1.38_{\pm0.28}$ | $1.88_{\pm0.26}$ | $.4359_{\pm.0015}$ | $.7831_{\pm.0006}$ | $.5503_{\pm.0010}$ | $1.43_{\pm0.11}\times10^{3}$ | $8.84_{\pm0.09}$ |
| WAlign | $.4176_{\pm.0041}$ | $.5650_{\pm.0048}$ | $.4753_{\pm.0039}$ | $0.23_{\pm0.04}$ | $1.85_{\pm0.06}$ | $.2308_{\pm.0041}$ | $.3684_{\pm.0073}$ | $.2785_{\pm.0049}$ | $2.52_{\pm0.08}\times10^{1}$ | $2.29_{\pm0.01}\times10^{1}$ |
| BRIGHT | $.3839_{\pm.0019}$ | $.6966_{\pm.0025}$ | $.4934_{\pm.0013}$ | $3.61_{\pm0.05}$ | $1.88_{\pm0.02}$ | $.4216_{\pm.0006}$ | $.7263_{\pm.0010}$ | $.5270_{\pm.0007}$ | $3.06_{\pm0.02}\times10^{2}$ | $1.13_{\pm0.00}\times10^{1}$ |
| NeXtAlign | $.4096_{\pm.0106}$ | $.7212_{\pm.0087}$ | $.5192_{\pm.0087}$ | $3.07_{\pm0.12}$ | $1.67_{\pm0.00}$ | $.4189_{\pm.0067}$ | $.7461_{\pm.0098}$ | $.5291_{\pm.0023}$ | $2.91_{\pm0.07}\times10^{2}$ | $1.78_{\pm0.02}\times10^{1}$ |
| WLAlign | $.2754_{\pm.0011}$ | $.4398_{\pm.0010}$ | $.3349_{\pm.0002}$ | $7.02_{\pm0.43}\times10^{2}$ | $1.25_{\pm0.03}$ | $.4873_{\pm.0007}$ | $.6593_{\pm.0010}$ | $.5425_{\pm.0006}$ | $5.98_{\pm0.08}\times10^{3}$ | $2.77_{\pm0.00}\times10^{1}$ |
| PARROT | $.6961_{\pm.0000}$ | $.8639_{\pm.0000}$ | $.7599_{\pm.0000}$ | $6.02_{\pm0.99}$ | $1.73_{\pm0.12}$ | $.7259_{\pm.0000}$ | $.9169_{\pm.0000}$ | $.7948_{\pm.0000}$ | $7.60_{\pm0.22}\times10^{2}$ | $2.55_{\pm0.06}\times10^{1}$ |
| SLOTAlign | $.6654_{\pm.0000}$ | $.7621_{\pm.0000}$ | $.7044_{\pm.0000}$ | $1.79_{\pm0.01}$ | $1.96_{\pm0.10}$ | $.3642_{\pm.3152}$ | $.4853_{\pm.4175}$ | $.4068_{\pm.3501}$ | $7.01_{\pm5.21}\times10^{2}$ | $2.51_{\pm0.09}\times10^{1}$ |
| HOT | $.4173_{\pm.0071}$ | $.6413_{\pm.0084}$ | $.2481_{\pm.0036}$ | $3.83_{\pm0.07}\times10^{1}$ | $2.36_{\pm0.58}$ | $.3994_{\pm.0014}$ | $.6475_{\pm.0022}$ | $.2417_{\pm.0009}$ | $1.90_{\pm0.18}\times10^{3}$ | $1.59_{\pm0.18}\times10^{1}$ |
| JOENA | $.8238_{\pm.0033}$ | $.9212_{\pm.0005}$ | $.8646_{\pm.0021}$ | $0.50_{\pm0.01}$ | $1.00_{\pm0.04}$ | $.7578_{\pm.0011}$ | $.9370_{\pm.0006}$ | $.8271_{\pm.0006}$ | $5.63_{\pm0.50}\times10^{1}$ | $4.19_{\pm0.00}$ |

| Dataset | PPI | | | | | GGI | | | | |
|---|---|---|---|---|---|---|---|---|---|---|
| Metrics | Hits@1 | Hits@10 | MRR | Time(s) | Mem.(GB) | Hits@1 | Hits@10 | MRR | Time(s) | Mem.(GB) |
| IsoRank | $.3622_{\pm.0000}$ | $.6175_{\pm.0000}$ | $.4462_{\pm.0000}$ | $2.19_{\pm0.25}$ | $1.18_{\pm0.02}$ | $.2512_{\pm.0000}$ | $.5064_{\pm.0000}$ | $.3372_{\pm.0000}$ | $3.59_{\pm0.09}\times10^{1}$ | $5.33_{\pm0.18}$ |
| FINAL | $.4479_{\pm.0000}$ | $.8191_{\pm.0000}$ | $.5743_{\pm.0000}$ | $0.50_{\pm0.01}$ | $1.50_{\pm0.00}$ | $.1920_{\pm.0000}$ | $.6396_{\pm.0000}$ | $.3402_{\pm.0000}$ | $4.61_{\pm0.33}$ | $7.42_{\pm0.11}$ |
| IONE | $.8350_{\pm.0047}$ | $.9218_{\pm.0036}$ | $.8658_{\pm.0042}$ | $1.68_{\pm0.07}\times10^{4}$ | $1.17_{\pm0.02}$ | $.5021_{\pm.0101}$ | $.6829_{\pm.0079}$ | $.5652_{\pm.0094}$ | $1.72_{\pm0.06}\times10^{4}$ | $3.32_{\pm0.18}$ |
| REGAL | $.0110_{\pm.0024}$ | $.0724_{\pm.0047}$ | $.0330_{\pm.0029}$ | $4.83_{\pm0.12}$ | $0.94_{\pm0.00}$ | $.0171_{\pm.0011}$ | $.0780_{\pm.0038}$ | $.0387_{\pm.0017}$ | $1.36_{\pm0.01}\times10^{1}$ | $2.52_{\pm0.04}$ |
| CrossMNA | $.8045_{\pm.0029}$ | $.9179_{\pm.0046}$ | $.8445_{\pm.0028}$ | $6.55_{\pm0.11}\times10^{2}$ | $1.09_{\pm0.02}$ | $.3961_{\pm.0030}$ | $.6964_{\pm.0041}$ | $.5017_{\pm.0028}$ | $7.65_{\pm0.14}\times10^{2}$ | $2.67_{\pm0.00}$ |
| NetTrans | $.5714_{\pm.0011}$ | $.8012_{\pm.0021}$ | $.6459_{\pm.0008}$ | $3.63_{\pm0.18}\times10^{1}$ | $2.16_{\pm0.11}$ | $.4024_{\pm.0014}$ | $.6705_{\pm.0008}$ | $.4918_{\pm.0011}$ | $2.76_{\pm0.57}\times10^{2}$ | $7.34_{\pm4.13}$ |
| WAlign | $.2912_{\pm.0037}$ | $.3877_{\pm.0041}$ | $.3264_{\pm.0038}$ | $1.78_{\pm0.06}$ | $2.37_{\pm0.03}$ | $.2978_{\pm.0090}$ | $.3791_{\pm.0092}$ | $.3282_{\pm.0088}$ | $5.33_{\pm0.24}$ | $7.32_{\pm0.07}$ |
| BRIGHT | $.4828_{\pm.0023}$ | $.6354_{\pm.0037}$ | $.5387_{\pm.0018}$ | $4.95_{\pm0.04}$ | $2.37_{\pm0.05}$ | $.3960_{\pm.0049}$ | $.5870_{\pm.0023}$ | $.4642_{\pm.0008}$ | $9.11_{\pm0.35}\times10^{1}$ | $5.74_{\pm0.03}$ |
| NeXtAlign | $.4576_{\pm.0012}$ | $.6098_{\pm.0019}$ | $.5248_{\pm.0031}$ | $4.65_{\pm0.07}$ | $2.54_{\pm0.01}$ | $.2899_{\pm.0067}$ | $.5263_{\pm.0025}$ | $.3717_{\pm.0056}$ | $5.19_{\pm0.03}\times10^{1}$ | $6.87_{\pm0.04}$ |
| WLAlign | $.7466_{\pm.0005}$ | $.8126_{\pm.0016}$ | $.7682_{\pm.0005}$ | $1.04_{\pm0.07}\times10^{3}$ | $1.74_{\pm0.04}$ | $.4354_{\pm.0006}$ | $.5440_{\pm.0012}$ | $.4710_{\pm.0005}$ | $1.82_{\pm0.03}\times10^{3}$ | $9.40_{\pm0.02}$ |
| PARROT | $.9619_{\pm.0000}$ | $.9926_{\pm.0000}$ | $.9731_{\pm.0000}$ | $1.80_{\pm0.04}\times10^{1}$ | $1.58_{\pm0.03}$ | $.8203_{\pm.0000}$ | $.9373_{\pm.0000}$ | $.8621_{\pm.0000}$ | $2.30_{\pm0.09}\times10^{2}$ | $8.47_{\pm0.03}$ |
| SLOTAlign | $.7398_{\pm.0000}$ | $.8219_{\pm.0000}$ | $.7684_{\pm.0000}$ | $3.98_{\pm0.29}$ | $1.64_{\pm0.02}$ | $.7170_{\pm.0003}$ | $.8097_{\pm.0017}$ | $.7500_{\pm.0011}$ | $1.98_{\pm0.04}\times10^{2}$ | $8.15_{\pm0.03}$ |
| HOT | $.3822_{\pm.0024}$ | $.4667_{\pm.0031}$ | $.2052_{\pm.0011}$ | $5.66_{\pm0.77}\times10^{1}$ | $6.70_{\pm0.00}$ | $.3466_{\pm.0020}$ | $.4724_{\pm.0023}$ | $.1931_{\pm.0011}$ | $8.67_{\pm4.82}\times10^{2}$ | $5.13_{\pm0.44}$ |
| JOENA | $.9856_{\pm.0000}$ | $.9995_{\pm.0000}$ | $.9907_{\pm.0000}$ | $0.54_{\pm0.02}$ | $1.08_{\pm0.02}$ | $.8665_{\pm.0002}$ | $.9583_{\pm.0001}$ | $.9016_{\pm.0000}$ | $1.01_{\pm0.00}\times10^{1}$ | $1.99_{\pm0.00}$ |

| Dataset | DBP15K_JA-EN | | | | | DBP15K_FR-EN | | | | |
|---|---|---|---|---|---|---|---|---|---|---|
| Metrics | Hits@1 | Hits@10 | MRR | Time(s) | Mem.(GB) | Hits@1 | Hits@10 | MRR | Time(s) | Mem.(GB) |
| IsoRank | $.1285_{\pm.0000}$ | $.5095_{\pm.0001}$ | $.2465_{\pm.0001}$ | $1.51_{\pm0.07}\times10^{2}$ | $1.60_{\pm0.00}\times10^{1}$ | $.1072_{\pm.0000}$ | $.5024_{\pm.0001}$ | $.2281_{\pm.0000}$ | $1.50_{\pm0.05}\times10^{2}$ | $1.59_{\pm0.03}\times10^{1}$ |
| FINAL | $.1437_{\pm.0000}$ | $.5580_{\pm.0000}$ | $.2784_{\pm.0000}$ | $1.82_{\pm0.03}\times10^{1}$ | $2.46_{\pm0.00}\times10^{1}$ | $.1414_{\pm.0000}$ | $.5701_{\pm.0000}$ | $.2781_{\pm.0000}$ | $1.81_{\pm0.01}\times10^{1}$ | $2.45_{\pm0.03}\times10^{1}$ |
| IONE | $.0405_{\pm.0037}$ | $.1864_{\pm.0108}$ | $.0903_{\pm.0058}$ | $1.55_{\pm0.02}\times10^{4}$ | $8.83_{\pm0.06}$ | $.0366_{\pm.0010}$ | $.1719_{\pm.0066}$ | $.0836_{\pm.0021}$ | $1.46_{\pm0.01}\times10^{4}$ | $8.72_{\pm0.32}$ |
| REGAL | $.0133_{\pm.0060}$ | $.0453_{\pm.0082}$ | $.0250_{\pm.0067}$ | $3.09_{\pm0.04}\times10^{1}$ | $6.06_{\pm0.11}$ | $.0065_{\pm.0010}$ | $.0247_{\pm.0007}$ | $.0140_{\pm.0003}$ | $3.16_{\pm0.02}\times10^{1}$ | $5.92_{\pm0.02}$ |
| CrossMNA | $.0179_{\pm.0013}$ | $.2932_{\pm.0042}$ | $.1008_{\pm.0020}$ | $1.01_{\pm0.03}\times10^{3}$ | $6.18_{\pm0.04}$ | $.0321_{\pm.0010}$ | $.2888_{\pm.0032}$ | $.1121_{\pm.0013}$ | $1.32_{\pm0.04}\times10^{3}$ | $6.23_{\pm0.04}$ |
| NetTrans | $.3044_{\pm.0011}$ | $.6373_{\pm.0015}$ | $.4103_{\pm.0010}$ | $2.96_{\pm0.70}\times10^{2}$ | $2.14_{\pm0.56}\times10^{1}$ | $.2975_{\pm.0003}$ | $.6457_{\pm.0007}$ | $.4080_{\pm.0004}$ | $2.98_{\pm0.82}\times10^{2}$ | $1.43_{\pm0.45}\times10^{1}$ |
| WAlign | $.2334_{\pm.0030}$ | $.3207_{\pm.0029}$ | $.2673_{\pm.0031}$ | $9.46_{\pm0.14}$ | $1.52_{\pm0.00}\times10^{1}$ | $.1638_{\pm.0053}$ | $.2432_{\pm.0077}$ | $.1946_{\pm.0059}$ | $1.16_{\pm0.04}\times10^{1}$ | $1.60_{\pm0.00}\times10^{1}$ |
| BRIGHT | $.3264_{\pm.0019}$ | $.6255_{\pm.0078}$ | $.4267_{\pm.0011}$ | $3.91_{\pm0.84}\times10^{2}$ | $1.17_{\pm0.00}\times10^{1}$ | $.3143_{\pm.0010}$ | $.6313_{\pm.0018}$ | $.4203_{\pm.0004}$ | $4.31_{\pm0.29}\times10^{2}$ | $1.19_{\pm0.01}\times10^{1}$ |
| NeXtAlign | $.2866_{\pm.0402}$ | $.6001_{\pm.0340}$ | $.3920_{\pm.0387}$ | $2.66_{\pm0.80}\times10^{3}$ | $1.43_{\pm nan}\times10^{1}$ | $.2695_{\pm.0983}$ | $.5981_{\pm.0953}$ | $.3790_{\pm.1004}$ | $2.60_{\pm0.03}\times10^{3}$ | $1.34_{\pm0.07}\times10^{1}$ |
| WLAlign | $.2661_{\pm.0005}$ | $.4378_{\pm.0009}$ | $.3212_{\pm.0005}$ | $3.19_{\pm0.14}\times10^{3}$ | $2.81_{\pm0.00}\times10^{1}$ | $.2764_{\pm.0006}$ | $.4769_{\pm.0008}$ | $.3401_{\pm.0004}$ | $3.63_{\pm0.29}\times10^{3}$ | $2.99_{\pm0.01}\times10^{1}$ |
| PARROT | $.6453_{\pm.0000}$ | $.8600_{\pm.0000}$ | $.7164_{\pm.0000}$ | $9.13_{\pm0.09}\times10^{2}$ | $2.87_{\pm0.00}\times10^{1}$ | $.6999_{\pm.0000}$ | $.9038_{\pm.0000}$ | $.7697_{\pm.0000}$ | $9.01_{\pm0.22}\times10^{2}$ | $2.87_{\pm0.03}\times10^{1}$ |
| SLOTAlign | $.0063_{\pm.0015}$ | $.0294_{\pm.0054}$ | $.0157_{\pm.0029}$ | $2.17_{\pm0.12}\times10^{2}$ | $2.84_{\pm nan}\times10^{1}$ | $.0188_{\pm.0000}$ | $.0725_{\pm.0000}$ | $.0381_{\pm.0000}$ | $1.71_{\pm0.08}\times10^{2}$ | $2.84_{\pm0.03}\times10^{1}$ |
| HOT | $.3512_{\pm.0034}$ | $.6174_{\pm.0045}$ | $.2199_{\pm.0017}$ | $2.08_{\pm0.04}\times10^{3}$ | $1.91_{\pm0.10}\times10^{1}$ | $.3504_{\pm.0029}$ | $.6352_{\pm.0041}$ | $.2210_{\pm.0016}$ | $2.07_{\pm0.06}\times10^{3}$ | $1.62_{\pm0.06}\times10^{1}$ |
| JOENA | $.6278_{\pm.0029}$ | $.8631_{\pm.0047}$ | $.6989_{\pm.0036}$ | $6.42_{\pm0.33}\times10^{2}$ | $2.67_{\pm0.00}\times10^{1}$ | $.7133_{\pm.0053}$ | $.9210_{\pm.0033}$ | $.7739_{\pm.0041}$ | $6.72_{\pm0.33}\times10^{2}$ | $2.67_{\pm0.03}\times10^{1}$ |

| Dataset | Airport | | | | | PeMS08 | | | | |
|---|---|---|---|---|---|---|---|---|---|---|
| Metrics | Hits@1 | Hits@10 | MRR | Time(s) | Mem.(GB) | Hits@1 | Hits@10 | MRR | Time(s) | Mem.(GB) |
| IsoRank | $.1308_{\pm.0000}$ | $.3755_{\pm.0000}$ | $.2074_{\pm.0000}$ | $0.11_{\pm0.00}$ | $0.82_{\pm0.18}$ | $.2537_{\pm.0000}$ | $.7353_{\pm.0000}$ | $.4197_{\pm.0000}$ | $0.23_{\pm0.24}$ | $0.55_{\pm0.02}$ |
| FINAL | $.2117_{\pm.0000}$ | $.6318_{\pm.0000}$ | $.3520_{\pm.0000}$ | $0.16_{\pm0.02}$ | $0.92_{\pm0.18}$ | $.1985_{\pm.0000}$ | $.7574_{\pm.0000}$ | $.3917_{\pm.0000}$ | $0.01_{\pm0.00}$ | $0.64_{\pm0.00}$ |
| IONE | $.4809_{\pm.0054}$ | $.7073_{\pm.0078}$ | $.5575_{\pm.0063}$ | $1.46_{\pm0.10}\times10^{4}$ | $1.03_{\pm0.19}$ | $.3824_{\pm.0252}$ | $.6728_{\pm.0351}$ | $.4833_{\pm.0202}$ | $7.43_{\pm4.40}\times10^{3}$ | $0.84_{\pm0.02}$ |
| REGAL | $.0302_{\pm.0036}$ | $.1477_{\pm.0054}$ | $.0698_{\pm.0031}$ | $1.17_{\pm0.02}$ | $0.81_{\pm0.00}$ | $.0493_{\pm.0115}$ | $.2272_{\pm.0126}$ | $.1167_{\pm.0087}$ | $0.14_{\pm0.00}$ | $0.81_{\pm0.00}$ |
| CrossMNA | $.4304_{\pm.0061}$ | $.7186_{\pm.0178}$ | $.5241_{\pm.0081}$ | $1.40_{\pm0.02}\times10^{2}$ | $0.98_{\pm0.00}$ | $.0066_{\pm.0048}$ | $.0640_{\pm.0178}$ | $.0336_{\pm.0082}$ | $0.10_{\pm0.00}$ | $0.98_{\pm0.00}$ |
| NetTrans | $.4293_{\pm.0034}$ | $.6817_{\pm.0013}$ | $.5154_{\pm.0029}$ | $1.80_{\pm0.51}\times10^{1}$ | $1.47_{\pm0.29}$ | $.3985_{\pm.0141}$ | $.8794_{\pm.0066}$ | $.5621_{\pm.0077}$ | $0.77_{\pm0.44}$ | $1.26_{\pm0.00}$ |
| WAlign | $.2270_{\pm.0113}$ | $.4163_{\pm.0126}$ | $.2924_{\pm.0106}$ | $0.13_{\pm0.00}$ | $2.03_{\pm0.46}$ | $.5846_{\pm.0074}$ | $.8441_{\pm.0042}$ | $.6790_{\pm.0065}$ | $0.13_{\pm0.00}$ | $1.16_{\pm0.34}$ |
| BRIGHT | $.3495_{\pm.0026}$ | $.5667_{\pm.0089}$ | $.4282_{\pm.0028}$ | $0.47_{\pm0.02}$ | $1.37_{\pm0.03}$ | $.4566_{\pm.0060}$ | $.8882_{\pm.0056}$ | $.6123_{\pm.0046}$ | $0.03_{\pm0.00}$ | $1.04_{\pm0.00}$ |
| NeXtAlign | $.2946_{\pm.0078}$ | $.5273_{\pm.0012}$ | $.3748_{\pm.0039}$ | $0.71_{\pm0.02}$ | $1.64_{\pm0.03}$ | $.4596_{\pm.0054}$ | $.8676_{\pm.0123}$ | $.6139_{\pm.0087}$ | $0.04_{\pm0.00}$ | $1.17_{\pm0.04}$ |
| WLAlign | $.4761_{\pm.0018}$ | $.6167_{\pm.0011}$ | $.5252_{\pm.0014}$ | $7.06_{\pm0.71}\times10^{2}$ | $1.33_{\pm0.00}$ | $.3676_{\pm.0069}$ | $.5125_{\pm.0072}$ | $.4203_{\pm.0029}$ | $3.28_{\pm0.55}\times10^{2}$ | $1.33_{\pm0.00}$ |
| PARROT | $.6891_{\pm.0000}$ | $.8687_{\pm.0000}$ | $.7488_{\pm.0000}$ | $1.32_{\pm0.05}$ | $0.91_{\pm0.00}$ | $.7647_{\pm.0000}$ | $.9228_{\pm.0000}$ | $.8209_{\pm.0000}$ | $0.07_{\pm0.00}$ | $0.82_{\pm0.00}$ |
| SLOTAlign | $.6691_{\pm.0008}$ | $.7873_{\pm.0010}$ | $.7098_{\pm.0012}$ | $9.89_{\pm0.32}$ | $1.12_{\pm0.02}$ | $.9118_{\pm.0000}$ | $.9743_{\pm.0000}$ | $.9280_{\pm.0000}$ | $1.54_{\pm0.18}$ | $0.96_{\pm0.00}$ |
| HOT | $.2889_{\pm.0124}$ | $.4648_{\pm.0118}$ | $.1737_{\pm.0059}$ | $1.54_{\pm0.19}\times10^{1}$ | $2.17_{\pm0.17}$ | $.6103_{\pm.0355}$ | $.8566_{\pm.0045}$ | $.3529_{\pm.0134}$ | $0.44_{\pm0.10}$ | $2.10_{\pm0.00}$ |
| JOENA | $.8459_{\pm.0026}$ | $.9637_{\pm.0011}$ | $.8887_{\pm.0012}$ | $0.07_{\pm0.00}$ | $1.03_{\pm0.00}$ | $.9390_{\pm.0134}$ | $.9941_{\pm.0020}$ | $.9568_{\pm.0101}$ | $0.03_{\pm0.00}$ | $1.04_{\pm0.00}$ |

Table 9: Detailed effectiveness results on attributed networks with a training ratio of 20%. The 1st/2nd/3rd best results are marked in red/blue/green respectively. Time denotes the inference time and Mem. denotes the peak memory usage.

| Dataset | Douban | | | | | Flickr-LastFM | | | | |
|---|---|---|---|---|---|---|---|---|---|---|
| Metrics | Hits@1 | Hits@10 | MRR | Time(s) | Mem.(GB) | Hits@1 | Hits@10 | MRR | Time(s) | Mem.(GB) |
| FINAL | .5397±.0012 | .9838±.0013 | .6999±.0032 | 0.90±.01 | 0.87±.02 | .0152±.0002 | .1022±.0004 | .0422±.0000 | 3.66±.80 | 3.70±.00 |
| REGAL | .0352±.0003 | .1525±.0012 | .0758±.0006 | 3.20±.08 | 2.55±.03 | .0086±.0020 | .0580±.0026 | .0283±.0013 | 2.63±.22×10¹ | 2.70±.27 |
| NetTrans | .3274±.0006 | .6145±.0002 | .4226±.0004 | 1.51±.06 | 1.23±.00 | .0041±.0001 | .0401±.0012 | .0201±.0021 | 3.15±.51×10¹ | 2.41±.15×10¹ |
| WAlign | .2855±.0012 | .5698±.0027 | .3798±.0023 | 0.03±.00 | 1.25±.01 | .0169±.0006 | .0710±.0029 | .0416±.0008 | 2.30±.05 | 4.62±.05 |
| BRIGHT | .2813±.0075 | .6095±.0119 | .3966±.0083 | 1.52±.45 | 1.45±.04 | .0345±.0031 | .1019±.0059 | .0602±.0022 | 6.15±.57×10¹ | 3.80±.00 |
| NeXtAlign | .1879±.0045 | .4918±.0012 | .2756±.0022 | 1.48±.01 | 1.47±.00 | .0083±.0038 | .0384±.0141 | .0214±.0070 | 2.04±.26 | 2.00±.10 |
| PARROT | .6413±.0000 | .9408±.0000 | .7481±.0000 | 0.25±.02 | 1.34±.02 | .0442±.0000 | .1064±.0000 | .0701±.0000 | 5.13±.02×10¹ | 1.86±.00 |
| SLOTAlign | .4397±.0000 | .7207±.0000 | .5394±.0000 | 0.01±.00 | 1.31±.01 | .0055±.0000 | .0387±.0000 | .0205±.0000 | 2.58±.02×10¹ | 1.63±.02×10¹ |
| HOT | .3223±.0032 | .6391±.0174 | .2578±.0064 | 8.76±.21 | 1.41±.00 | .0152±.0013 | .0525±.0003 | .0141±.0021 | 4.63±.17×10² | 8.12±.19 |
| JOENA | .6542±.0474 | .9173±.0205 | .7525±.0446 | 0.24±.02 | 1.37±.00 | .0345±.0011 | .1064±.0034 | .0600±.0003 | 1.89±.21×10¹ | 2.63±.02 |

| Dataset | Flickr-MySpace | | | | | Arenas | | | | |
|---|---|---|---|---|---|---|---|---|---|---|
| Metrics | Hits@1 | Hits@10 | MRR | Time(s) | Mem.(GB) | Hits@1 | Hits@10 | MRR | Time(s) | Mem.(GB) |
| FINAL | .0023±.0000 | .0234±.0000 | .0113±.0000 | 1.57±.31 | 2.33±.00 | .4284±.0000 | .9069±.0000 | .5928±.0000 | 0.61±.24 | 0.88±.01 |
| REGAL | .0065±.0030 | .0355±.0061 | .0197±.0040 | 1.45±.12×10¹ | 1.65±.07 | .8961±.0255 | .9815±.0030 | .9281±.0179 | 2.02±.55 | 1.32±.12 |
| NetTrans | .0075±.0019 | .0374±.0052 | .0201±.0014 | 2.58±.09 | 1.32±.00 | .9581±.0036 | .9879±.0013 | .9688±.0026 | 1.03±.05 | 1.17±.00 |
| WAlign | .0112±.0026 | .0505±.0054 | .0316±.0021 | 1.96±.58 | 2.65±.07 | .9808±.0003 | .9978±.0000 | .9886±.0002 | 1.75±1.05 | 2.19±.07 |
| BRIGHT | .0061±.0021 | .0332±.0019 | .0196±.0012 | 3.50±.30×10¹ | 2.38±.00 | .9794±.0005 | .9950±.0006 | .9863±.0004 | 0.51±.21 | 1.05±.00 |
| NeXtAlign | .0037±.0042 | .0243±.0173 | .0162±.0075 | 0.91±.04 | 1.48±.03 | .6684±.2300 | .8230±.1401 | .7244±.1974 | 0.47±.33 | 1.09±.04 |
| PARROT | .0070±.0000 | .0397±.0000 | .0223±.0000 | 1.01±.02×10¹ | 1.40±.00 | .9879±.0000 | .9999±.0000 | .9936±.0000 | 2.56±1.51×10¹ | 0.77±.01 |
| SLOTAlign | .0023±.0000 | .0257±.0000 | .0163±.0000 | 1.03±.01×10¹ | 7.51±.10 | .9891±.0010 | .9999±.0000 | .9942±.0005 | 0.13±.00 | 0.88±.01 |
| HOT | .0000±.0000 | .0257±.0000 | .0024±.0000 | 3.26±.05×10² | 2.71±.00 | .9714±.0026 | .9927±.0015 | .4909±.0009 | 4.20±.62 | 1.15±.00 |
| JOENA | .0117±.0001 | .0584±.0000 | .0345±.0000 | 4.15±.02 | 1.71±.01 | .9873±.0002 | .9999±.0000 | .9929±.0000 | 0.34±.00 | 1.02±.04 |

| Dataset | ACM-DBLP | | | | | Cora | | | | |
|---|---|---|---|---|---|---|---|---|---|---|
| Metrics | Hits@1 | Hits@10 | MRR | Time(s) | Mem.(GB) | Hits@1 | Hits@10 | MRR | Time(s) | Mem.(GB) |
| FINAL | .4054±.0000 | .7980±.0000 | .5366±.0000 | 2.56±.64 | 1.19±.06 | .7891±.0000 | .9026±.0000 | .8332±.0000 | 1.48±.13 | 1.04±.01 |
| REGAL | .4159±.0035 | .6373±.0064 | .4890±.0040 | 1.95±.12×10¹ | 1.78±.05×10¹ | .3576±.0278 | .4705±.0271 | .3986±.0242 | 5.15±1.27 | 6.84±.21 |
| NetTrans | .6874±.0015 | .9300±.0019 | .7716±.0012 | 2.64±2.18×10² | 1.48±.02 | .7907±.4417 | .8005±.4454 | .7955±.4428 | 1.47±.07 | 1.32±.00 |
| WAlign | .6675±.0024 | .9109±.0012 | .7524±.0016 | 6.82±1.00 | 4.85±.03 | .9551±.0012 | .9724±.0010 | .9621±.0011 | 3.38±1.28 | 2.36±.05 |
| BRIGHT | .4858±.0025 | .8740±.0020 | .6163±.0021 | 1.22±.53×10² | 2.35±.00 | .7989±.0051 | .9902±.0012 | .8813±.0029 | 6.07±3.03 | 1.32±.04 |
| NeXtAlign | .3512±.0966 | .7633±.0872 | .4851±.0981 | 1.45±.64×10¹ | 1.47±.15 | .3336±.0418 | .6629±.0291 | .4444±.0376 | 1.56±.22 | 1.15±.02 |
| PARROT | .6867±.0000 | .9437±.0000 | .7770±.0000 | 1.70±.40×10¹ | 1.27±.00 | .9654±.0000 | .9684±.0000 | .9667±.0000 | 1.13±.56 | 1.05±.02 |
| SLOTAlign | .6673±.0011 | .8720±.0003 | .7409±.0009 | 8.34±.04×10¹ | 7.45±.16 | .9949±.0000 | .9974±.0000 | .9974±.0000 | 1.92±.13 | 2.01±.10 |
| HOT | .3893±.0050 | .6180±.0055 | .2350±.0017 | 4.28±.13×10² | 2.85±.01 | .7493±.0040 | .7549±.0040 | .3762±.0021 | 2.17±.59×10¹ | 3.29±.28 |
| JOENA | .7859±.0053 | .9847±.0101 | .8569±.0120 | 9.78±.27 | 1.59±.11 | .9947±.0002 | .9999±.0000 | .9966±.0001 | 0.38±.03 | 1.14±.02 |

| Dataset | PPI | | | | | DBP15K_FR-EN | | | | |
|---|---|---|---|---|---|---|---|---|---|---|
| Metrics | Hits@1 | Hits@10 | MRR | Time(s) | Mem.(GB) | Hits@1 | Hits@10 | MRR | Time(s) | Mem.(GB) |
| FINAL | .0316±.0000 | .1564±.0000 | .0760±.0000 | 1.87±.28 | 0.93±.00 | .1954±.0000 | .4381±.0000 | .2800±.0000 | 4.36±.10×10² | 2.37±.00×10¹ |
| REGAL | .1515±.0080 | .2910±.0072 | .2012±.0078 | 9.30±.42 | 3.89±.12 | .0027±.0004 | .0038±.0004 | .0035±.0004 | 5.30±.18×10¹ | 9.33±.03×10¹ |
| NetTrans | .7356±.0027 | .8581±.0024 | .7783±.0027 | 2.39±.50×10¹ | 1.22±.00 | .4308±.0012 | .7789±.0081 | .5412±.0109 | 1.38±1.03×10¹ | 1.03±.01×10¹ |
| WAlign | .7657±.0021 | .8733±.0007 | .8041±.0009 | 6.84±1.14 | 3.15±.01 | .5077±.0024 | .6492±.0046 | .5572±.0037 | 2.03±.26×10¹ | 1.67±.00×10¹ |
| BRIGHT | .7156±.0030 | .8623±.0036 | .7691±.0026 | 8.76±3.53 | 1.29±.01 | .4393±.0029 | .8020±.0023 | .5652±.0029 | 2.48±.05×10² | 9.55±.03 |
| NeXtAlign | .0306±.0142 | .1059±.0517 | .0579±.0266 | 1.41±.84×10¹ | 1.25±.10 | .4781±.0023 | .8780±.0012 | .6109±.0032 | 2.78±.08×10² | 9.87±.01 |
| PARROT | .9916±.0001 | .9977±.0000 | .9943±.0001 | 5.55±3.14×10¹ | 1.58±.00 | .8737±.0000 | .9550±.0000 | .9040±.0000 | 1.60±.22×10³ | 2.76±.00×10¹ |
| SLOTAlign | .9531±.0000 | .9777±.0000 | .9618±.0000 | 3.70±.01 | 1.71±.00 | .7619±.0012 | .8721±.0029 | .8091±.0012 | 1.70±.12×10² | 2.18±.00×10¹ |
| HOT | .6705±.0032 | .7154±.0054 | .3439±.0018 | 3.84±.99×10¹ | 1.30±.00 | .6172±.0134 | .7980±.0012 | .6512±.0012 | 1.70±.12×10³ | 2.17±.00×10¹ |
| JOENA | .9804±.0012 | .9943±.0041 | .9857±.0024 | 0.73±.12 | 1.10±.00 | .9804±.0012 | .9943±.0041 | .9857±.0024 | 0.73±.12 | 1.10±.00 |

| Dataset | Airport | | | | | PeMS08 | | | | |
|---|---|---|---|---|---|---|---|---|---|---|
| Metrics | Hits@1 | Hits@10 | MRR | Time(s) | Mem.(GB) | Hits@1 | Hits@10 | MRR | Time(s) | Mem.(GB) |
| FINAL | .3897±.0000 | .8267±.0000 | .5414±.0000 | 0.47±.15 | 0.89±.00 | .2132±.0000 | .7610±.0000 | .4051±.0000 | 0.02±.02 | 0.90±.00 |
| REGAL | .0516±.0037 | .2096±.0125 | .1063±.0059 | 2.04±.48 | 1.38±.10 | .3617±.0321 | .5655±.0360 | .4283±.0338 | 0.22±.09 | 1.20±.00 |
| NetTrans | .2430±.0038 | .4667±.0034 | .3192±.0028 | 2.32±.33 | 1.18±.00 | .6353±.0048 | .8963±.0048 | .7127±.0022 | 0.07±.04 | 1.18±.00 |
| WAlign | .3041±.0054 | .5803±.0049 | .3987±.0059 | 1.50±.98 | 2.33±.04 | .6963±.0118 | .9066±.0076 | .7689±.0070 | 2.15±.41 | 2.20±.01 |
| BRIGHT | .3343±.0021 | .5671±.0028 | .4153±.0044 | 1.29±.56 | 1.06±.00 | .5280±.0166 | .8765±.0159 | .6524±.0074 | 0.53±1.04 | 1.06±.00 |
| NeXtAlign | .0792±.0561 | .2551±.1263 | .1398±.0785 | 0.46±.10 | 1.02±.00 | .2992±.0573 | .7162±.1283 | .4373±.0657 | 0.04±.04 | 0.99±.00 |
| PARROT | .8582±.0000 | .9622±.0000 | .8971±.0000 | 2.28±1.91×10¹ | 0.92±.02 | .8787±.0000 | .9596±.0000 | .9071±.0000 | 8.31±4.67 | 0.85±.00 |
| SLOTAlign | .8918±.0000 | .9816±.0000 | .9256±.0000 | 0.15±.01 | 0.98±.00 | .9853±.0000 | .9999±.0000 | .9926±.0000 | 0.01±.00 | 0.90±.00 |
| HOT | .4318±.0052 | .5770±.0116 | .2413±.0027 | 5.33±.24 | 1.16±.00 | .7324±.0175 | .8853±.0120 | .3921±.0063 | 0.35±.04 | 1.16±.00 |
| JOENA | .8734±.0034 | .9748±.0104 | .9117±.0030 | 0.10±.00 | 1.02±.00 | .9999±.0000 | .9999±.0000 | .9999±.0000 | 0.05±.00 | 1.02±.01 |

