# OpenReview forum: "PLANETALIGN: A Comprehensive Python Library for Benchmarking Network Alignment"
_ICLR.cc/2026/Conference — ICLR 2026 Poster_

### Official Review · Reviewer_67gk · 2025-10-24

**Soundness:** 3
**Presentation:** 3
**Contribution:** 2
**Rating:** 4
**Confidence:** 3

**Summary:**

This paper introduces a new python library, PLANETALIGN, for the evaluation of network alignment algorithms. This new framework includes 18 datasets from various domains, like social, biological, knowledge graph, and publication. It supports 14 methods which covers major categories of NA algorithms.
This new framework provides unified APIs, and standardized evaluation metrics for model implementation and evaluation. The author conducted experiments to compare the performance of 14 algorithms across all datasets on effectiveness, efficiency, robustness and sensitivity.
Compare with the existing algorithms, the new framework achieves similar accuracy with 3x faster on efficiency.

**Strengths:**

This new python library integrates a wide range of datasets and algorithms. It provides a useful and fair benchmark for NA researches. It provides a consistent API and unified evaluation pipeline, which makes the method comparisons easier, more consistent and more reliable.
The implementation is efficient and well optimized. The runtime is significantly less the the original baseline.
The experiments shows meaningful empirical findings.
The documentation is clear and easy to follow. Users can ramp up quickly with the tutorial and detailed references.

**Weaknesses:**

The main limitation is lack of extensibility on dataset, functions and customized use cases. This library is mainly focus on algorithm comparisons rather than support the downstream researches.
Through this python library is well designed and easy to use, it is not very clear how will it keep up with the fast iterating NA research field. New datasets and new algorithms emerge quickly.
Currently this framework mainly serves as a benchmark tool with fixed datasets. The paper does not mention whether there is a maintenance plan to expand to new datasets, or support for customized datasets.
This work can be stronger and makes larger impact if the library could be more extensible and more research orientated, supporting NA related researches rather than simply method comparison. For example, add APIs for downstream tasks, customized loss functions, multitask learning, etc. Such improvements will certainly contribute more to the acceleration of new scientific discoveries.

**Questions:**

My questions are mainly related to the limitations I mentioned above.
1. The paper mentioned that the multi-network alignment is not explored in this study. I was wondering, with the current architecture, how difficult would it be to extend PLANETALIGN to multi-graph or cross domain alignment?
2. Are there future plans to extend the support of new datasets or new graph modalities, like heterogeneous graph?
3. If future versions is improved to support new dataset, especially larger datasets, will it include distributed computation or other scalability improvements to handle large graphs?

---

> ### Author Response · Authors · 2025-11-20
> **Rebuttal by Authors 1/3**
>
> We thank the reviewer for the constructive feedback and comments, all of which are helpful to further improve our paper. In the following, we would like to provide point-to-point responses and clarifications.
>
> ### **Limitation 1: Lack of extensibility and long-term usability**
>
> We thank the reviewer for the insightful comments emphasizing the importance of extensibility and long-term usability of PLANETALIGN. We would like to clarify that PLANETALIGN was explicitly designed with extensibility and can serve as a powerful research toolkit for NA beyond benchmarking. Specifically, PLANETALIGN features:
>
> * **Extensible and easy-to-use APIs for customized datasets and algorithms**. As described in Section 4.2 of our paper, PLANETALIGN provides unified base classes *PlanetAlign.data.BaseData* and *PlanetAlign.algorithm.BaseModel*, which allows users to customize their own NA datasets and algorithms while remaining fully compatible with PLANETALIGN's APIs to streamline fair and reproducible evaluation.
>
> * **Comprehensive documentation and tutorials**. We provide detailed documentation and dedicated tutorials at https://planetalign.netlify.app. The "[Working with NA Datasets](https://planetalign.netlify.app/tutorials/datasets)" and "[Working with NA Algorithms](https://planetalign.netlify.app/tutorials/algorithms)" parts of the tutorial explicitly include sections guiding users to build customized datasets and algorithms on top of PLANETALIGN's APIs.
>
> * **Modular design for extensibility beyond benchmarking**. As shown in Figure 3, PLANETALIGN's modular structure supports customized **datasets**, **algorithms**, **training objectives (loss functions)**, and **evaluation metrics** to be directly plugged in without changing other components. The modular design also allows users to fully utilize the intermediate outputs for NA or downstream tasks (e.g., **node embeddings**, **alignment matrix**, **alignment performance**), such as cross-layer dependency inference[1], knowledge integration[2], and cross-KG modality fusion[3]. We have updated Figure 3 and Section 4.2 in the revision to further emphasize PLANETALIGN’s support for diverse customization and standardized outputs.
>
> To further address the reviewer’s concerns regarding extensibility and future maintenance & update plans of PLANETALIGN, we would like to clarify our current design and outline our detailed future roadmap in response to the reviewer’s specific questions below.
>
> ### **Question 1: Extensibility to multi-network and cross-domain settings**
>
> We thank the reviewer for this constructive question. While our paper primarily focuses on benchmarking under pairwise NA settings, the current architecture of PLANETALIGN has already been designed to support multi-network and cross-domain alignment with minimal modification.
>
> * **Support for multi-network alignment.** PLANETALIGN already implements two algorithms designed specifically for multi-network alignment, CrossMNA[4] and HOT[5], demonstrating that the library naturally accommodates multi-graph settings without changing the core codebase.
>
> * **Flexible API design supporting multi-network settings.** As shown in Figure 3, the core data class *PlanetAlign.data.BaseData* accepts a list of PyG graph objects and a tensor of anchor links without length restriction. Similarly, the base model class *PlanetAlign.algorithm.BaseModel* accepts a list of graph IDs via the *gids* argument. Both classes can naturally serve as the bases for pairwise or multi-network alignment datasets and algorithms.
>
> * **Native support for cross-domain alignment.** For cross-domain settings (e.g., aligning text and image networks), as long as graphs fed into the *PlanetAlign.algorithm.BaseModel* are provided as standard PyG (*torch_geometric.data.Data*) objects, PLANETALIGN’s unified APIs ensure direct compatibility. No changes to the APIs are required.
>
> For the future updates of PLANETALIGN, our immediate goal is to include datasets containing multi-network [4, 5] and cross-domain networks [6] (e.g., image-text networks) by collecting from existing works and synthesizing from single networks. Specifically,
>
> * For **multi-network datasets**, we plan to add real-world multi-layered version of ArXiv[7], Twitter[8], and SacchCere[9]. For synthetic datasets, we plan to follow [5] to manually create multi-layered versions of single networks by randomly perturbing edges and features in the networks.
>
> * For **cross-domain datasets**, we plan to follow [6] to construct vision-and-language understanding networks (Flickr30K[10], COCO[11]) and machine translation networks (English-Vietnamese TED-talks corpus[12]), and integrate the resulting cross-domain networks into PLANETALIGN.

---

> ### Author Response · Authors · 2025-11-20
> **Rebuttal by Authors 2/3**
>
> ### **Question 2: Plans for extension to other datasets**
>
> We appreciate the reviewer’s constructive question regarding heterogeneous graph modalities. PLANETALIGN is designed to accommodate heterogeneous graphs through its unified PyG-based data interface, and we plan to extend this capability further in future releases. Specifically, PLANETALIGN has already included and plans to include more heterogeneous graphs of the following types:
>
> * **Heterogeneity within a single graph (Intra-network heterogeneity).** For this type of heterogeneous network, PLANETALIGN has included three variants of the knowledge graphs DBP15K, where each graph contains multiple node and relation types. In the future, we plan to include more graphs with intra-network heterogeneity, such as additional knowledge graphs and heterogeneous biological networks (e.g., Hetionet[13]) where nodes represent different biological entities.
>
> * **Heterogeneity across different graphs (Inter-network heterogeneity).** PLANETALIGN has included real-world networks where the node types across aligned networks are different, e.g., Phone-Email, Foursquare-Twitter, etc. In the future, we also plan to include more graphs with different heterogeneous modalities, such as cross-domain heterogeneous networks (e.g, image-text networks [10, 11]), to improve the comprehensiveness of datasets in PLANETALIGN.
>
> In summary, PLANETALIGN already covers various heterogeneous networks, and we plan to include more heterogeneous networks that capture diverse graph modality in future releases of our library.
>
> ### **Question 3: Scalability optimization for large networks**
>
> We appreciate the reviewer’s question regarding scalability optimization. Extending PLANETALIGN to handle larger datasets is a key direction for improving our library. Specifically, we plan to introduce distributed training and matrix acceleration modules tailored to the computational characteristics of different types of NA algorithms:
>
> * For **embedding-based methods**, we plan to support distributed training on top of existing APIs provided by PyTorch (e.g., torch.distributed, DistributedDataParallel, FSDP, etc.) and PyG (e.g., torch_geometric.distributed). This will enable efficient parallelization of embedding learning across distributed compute nodes with consistent APIs.
>
> * For **consistency-based and OT-based methods** which involve heavy matrix operations, we plan to further incorporate sparse and low-rank matrix techniques, along with APIs for low-rank and sliced optimal transport[14, 15] to reduce both runtime and memory overheads. These modules will allow consistency-based and OT-based NA algorithms built upon PLANETALIGN's APIs capable of handling larger networks.
>
> In addition, the distributed and low-rank modules will be integrated into *PlanetAlign.utils* as optional APIs so that users can utilize these acceleration techniques with minimal code change.

---

> ### Author Response · Authors · 2025-11-20
> **Rebuttal by Authors 3/3**
>
> ### **References**
> [1] Yan, Yuchen, et al. "Dissecting cross-layer dependency inference on multi-layered inter-dependent networks." Proceedings of the 31st ACM International Conference on Information & Knowledge Management. 2022.
>
> [2] Yan, Yuchen, et al. "Dynamic knowledge graph alignment." Proceedings of the AAAI conference on artificial intelligence. Vol. 35. No. 5. 2021.
>
> [3] Chen, Zhuo, et al. "Meaformer: Multi-modal entity alignment transformer for meta modality hybrid." Proceedings of the 31st ACM international conference on multimedia. 2023.
>
> [4] Chu, Xiaokai, et al. "Cross-network embedding for multi-network alignment." The world wide web conference. 2019.
>
> [5] Zeng, Zhichen, et al. "Hierarchical multi-marginal optimal transport for network alignment." Proceedings of the AAAI Conference on Artificial Intelligence. Vol. 38. No. 15. 2024.
>
> [6] Chen, Liqun, et al. "Graph optimal transport for cross-domain alignment." International Conference on Machine Learning. PMLR, 2020.
>
> [7] De Domenico, Manlio, et al. "Identifying modular flows on multilayer networks reveals highly overlapping organization in interconnected systems." Physical Review X 5.1 (2015): 011027.
>
> [8] Omodei, Elisa, Manlio De Domenico, and Alex Arenas. "Characterizing interactions in online social networks during exceptional events." Frontiers in Physics 3 (2015): 59.
>
> [9] De Domenico, Manlio, et al. "Structural reducibility of multilayer networks." Nature communications 6.1 (2015): 6864.
>
> [10] Plummer, Bryan A., et al. "Flickr30k entities: Collecting region-to-phrase correspondences for richer image-to-sentence models." Proceedings of the IEEE international conference on computer vision. 2015.
>
> [11] Lin, Tsung-Yi, et al. "Microsoft coco: Common objects in context." European conference on computer vision. Cham: Springer International Publishing, 2014.
>
> [12] Cettolo, Mauro, et al. "The IWSLT 2016 evaluation campaign." Proceedings of the 13th International Conference on Spoken Language Translation. 2016.
>
> [13] Himmelstein, Daniel S., and Sergio E. Baranzini. "Heterogeneous network edge prediction: a data integration approach to prioritize disease-associated genes." PLoS computational biology 11.7 (2015): e1004259.
>
> [14] Scetbon, Meyer, and Marco Cuturi. "Low-rank optimal transport: Approximation, statistics and debiasing." Advances in Neural Information Processing Systems 35 (2022): 6802-6814.
>
> [15] Liu, Xinran, et al. "Expected sliced transport plans." arXiv preprint arXiv:2410.12176 (2024).

---

> ### Author Response · Authors · 2025-11-28
>
> Dear reviewer 67gk,
>
> Thank you again for your insightful comments!
>
> As the discussion period deadline approaches, we would like to respectfully inquire whether our responses address your concerns properly, and we are more than happy to address any remaining points.
>
> Many thanks for your time and effort!
>
> Sincerely,
>
> Authors

---

### Official Review · Reviewer_rJCh · 2025-10-26

**Soundness:** 3
**Presentation:** 4
**Contribution:** 3
**Rating:** 6
**Confidence:** 4

**Summary:**

This paper introduces PLANETALIGN, a comprehensive Python library for benchmarking Network Alignment (NA) algorithms across multiple domains. The framework integrates 18 datasets spanning six application areas (social, biological, academic, knowledge graph, infrastructure, and communication networks) and implements 14 representative algorithms, including consistency-based, embedding-based, and optimal transport (OT)-based methods. PLANETALIGN provides a unified PyTorch-style API, standardized evaluation metrics, and extensive experiments assessing algorithms on four dimensions—effectiveness, scalability, robustness, and supervision sensitivity. The work fills an important gap in the NA community by establishing a reproducible, standardized evaluation foundation, though it introduces no new alignment algorithm or theoretical framework.

**Strengths:**

(1) Comprehensive benchmark coverage: PLANETALIGN integrates 18 datasets from six domains and 14 representative algorithms, providing the most complete evaluation platform to date for network alignment research.
(2) Systematic multi-dimensional evaluation: The framework assesses algorithms along four complementary dimensions—effectiveness, scalability, robustness, and supervision sensitivity—offering a more holistic perspective than prior work.
(3) High implementation quality and reproducibility: The library provides a clean, modular PyTorch-style API, reproducible experiment settings, and optimized code achieving up to 3× faster runtime compared to original implementations.
(4) Insightful empirical analysis: Experiments yield practical insights, such as the superior stability of OT-based methods and the efficiency-memory trade-offs in embedding-based approaches.
(5) Community impact: By releasing a well-documented, open-source library, the work provides an infrastructure that can serve as a long-term standard for the NA community.

**Weaknesses:**

(1) Limited algorithmic novelty: The paper does not introduce new alignment methods or theoretical developments. Its contribution is primarily infrastructural, which may limit interest for readers expecting methodological innovation.
(2) Incomplete scalability validation: Experiments are limited to graphs with fewer than 100k nodes. Given the scalability claims, evaluation on larger-scale graphs (e.g., >1M nodes) would better support the authors’ argument.
(3) Shallow theoretical motivation: The paper lacks deeper theoretical analysis or justification for why OT-based approaches perform better, relying mainly on empirical evidence.
(4) Restricted scope: The current implementation only supports pairwise alignment. Extending the framework to multi-network or dynamic alignment scenarios could significantly broaden its applicability.

**Questions:**

(1) How does PLANETALIGN ensure fairness when comparing algorithms that rely on different supervision levels (e.g., supervised vs. semi-supervised vs. unsupervised settings)? Are hyperparameter budgets standardized across methods?
(2) The paper claims scalability and efficiency advantages. Could the authors include additional results on large-scale graphs (e.g., >1M nodes) to substantiate this claim?
(3) The framework currently focuses on pairwise alignment. Are there plans or design provisions to extend it toward multi-network or temporal (dynamic) alignment scenarios?
(4) Since the results show that OT-based methods consistently outperform others, can the authors provide more theoretical intuition or ablation evidence to explain why OT-based models are so robust across domains?

---

> ### Author Response · Authors · 2025-11-20
> **Rebuttal by Authors 1/3**
>
> We thank the reviewer for the constructive feedback and comments, all of which are helpful to further improve our paper. In the following, we would like to provide point-to-point responses and clarifications.
>
> ### **Limitation 1: Limited algorithmic novelty**
>
> We appreciate the reviewer’s observation and would like to respectfully clarify the intended scope of our submission. This paper was submitted under the primary area "Datasets and Benchmarks" of ICLR, which emphasizes contributions in terms of high-quality datasets and benchmarking infrastructures, rather than novel algorithmic methodology.
>
> While no new NA algorithms are proposed, PLANETALIGN offers **practical insights** into the **strengths, limitations, and trade-offs** of existing NA methods, and provides a valuable codebase that enables **fair, reproducible evaluation** to facilitate future research in NA.
>
> ### **Limitation 2: Incomplete scalability validation**
>
> We thank the reviewer for raising this important point. We would like to clarify that PLANETALIGN provides a comprehensive collection of the **most widely used** datasets in the NA community, which typically contain up to **20k** nodes per network. As NA is a **multi-network mining task** that requires identifying an alignment for every node across different networks, its computational complexity is inherently higher than that of single-network tasks such as node classification or link prediction.
>
> That said, we agree that testing PLANETALIGN's implementation on large-scale networks is important. To further validate the efficiency of PLANETALIGN, we compare the runtime between the official and PLANETALIGN's implementations on large-scale ER networks with **50K, 75K, and 100K** nodes, **which are much larger than commonly adopted datasets in the NA community**. The results are shown in the table below (Table 6 in the revision), showing that PLANETALIGN's implementations achieve up to **2.7** times speed-up even on large-scale networks. We can also see from Table 6 that most existing NA methods cannot scale to million-node networks due to their intrinsic complexity constraints. PLANETALIGN intentionally preserves the theoretical complexity of existing NA methods to ensure fair and consistent benchmarking.
>
> **Runtime (s) comparison with official implementations. OOM = Out-of-memory.**
>
> | Method      | 50K Official | 50K PLANETALIGN | 50K Speedup | 75K Official | 75K PLANETALIGN | 75K Speedup | 100K Official | 100K PLANETALIGN | 100K Speedup |
> |-------------|--------------|------------------|--------------|--------------|------------------|--------------|----------------|--------------------|---------------|
> | **REGAL**     | 334 | **214** | 1.56 | 583 | **390** | 1.49 | 1.12×10³ | **731** | 1.53 |
> | **CrossMNA**  | 4.12×10³ | **3.02×10³** | 1.36 | 7.36×10³ | **5.13×10³** | 1.43 | 1.37×10⁴ | **8.91×10³** | 1.54 |
> | **WAlign**    | 1.65×10³ | **1.29×10³** | 1.28 | 3.12×10³ | **2.54×10³** | 1.23 | OOM | OOM | OOM |
> | **BRIGHT**    | 2.82×10⁴ | **2.15×10⁴** | 1.31 | 8.95×10⁴ | **6.90×10⁴** | 1.29 | 3.47×10⁵ | **2.49×10⁵** | 1.40 |
> | **NeXtAlign** | 6.55×10⁴ | **6.08×10⁴** | 1.08 | 3.68×10⁵ | **3.41×10⁵** | 1.08 | OOM | OOM | OOM |
> | **WLAlign**   | 2.46×10⁵ | **9.12×10⁴** | 2.70 | OOM | OOM | OOM | OOM | OOM | OOM |
> | **SLOTAlign** | 3.01×10⁴ | **2.75×10⁴** | 1.10 | OOM | OOM | OOM | OOM | OOM | OOM |
> | **HOT**       | 3.14×10³ | **2.87×10³** | 1.10 | 6.17×10³ | **5.74×10³** | 1.07 | 1.08×10⁴ | **9.43×10³** | 1.15 |
> | **JOENA**     | 2.31×10³ | **2.28×10³** | 1.01 | 8.30×10⁴ | **8.12×10⁴** | 1.02 | OOM | OOM | OOM |
>
> In future releases, we plan to introduce more scalability modules within PLANETALIGN, including **1)** distributed training APIs for embedding-based methods built upon PyTorch and PyG's relevant APIs, and **2)** sparse and low-rank matrix techniques, as well as low-rank and sliced OT[1, 2] modules for consistency-based and OT-based methods which typically involve intensive matrix operations. These extensions will be provided as optional APIs, allowing NA algorithms built upon PLANETALIGN to handle large-scale networks with a few simple API calls.

---

> ### Author Response · Authors · 2025-11-20
> **Rebuttal by Authors 2/3**
>
> ### **Limitation 3: Shallow theoretical motivation**
>
> We thank the reviewer for the valuable suggestion. In addition to our empirical findings regarding the effectivness of OT-based methods, we provide below a more in-depth theoretical analysis of their superior performance.
>
> OT-based methods formulate the NA problem as an optimization problem that minimizes the total effort of transporting the node distribution of one graph to another under a set of pre-defined or learnable cost functions. Compared with consistency-based methods, which are restricted by local consistency principles, OT-based methods go beyond local assumptions by solving a globally constrained optimization problem. Compared with embedding-based methods, which infer alignment from noisy embedding similarities, OT-based methods directly learns a robust alignment matrix from transportation cost, thanks to the marginal constraints that naturally encourage one-to-one node alignment. Empowered by **constrained optimization** and **informative transportation cost** encoded by powerful graph proximity measures or learnable node embeddings, OT-based methods learns **robust**, **deterministic**, and **global-structure-aware** alignment.
>
> We have included the above analysis in Section 5.2 of the revision to enhance the theoretical depth of our paper.
>
>
> ### **Limitation 4: Restricted scope without multi-network and dynamic alignment**
>
> We thank the reviewer for this insightful comment. We agree that extending PLANETALIGN to support multi-network and dynamic alignment is an important future direction.
>
> * For **multi-network alignment**, we would like to **first** clarify that PLANETALIGN **already includes** two algorithms designed specifically for multi-network alignment: CrossMNA[3] and HOT[4], demonstrating that the library naturally accommodates multi-graph settings without changing the core codebase. **Secondly**, as shown in Figure 3, the core data class *PlanetAlign.data.BaseData* accepts a list of PyG graph objects and a tensor of anchor links without length restriction. Similarly, the base model class *PlanetAlign.algorithm.BaseModel* accepts a list of graph IDs via the *gids* argument. Both classes can naturally serve as the bases for pairwise or multi-network alignment tasks. We are also actively collecting multi-network alignment datasets (e.g, multi-layered version of ArXiv[5], Twitter[6], and SacchCere[7]) and algorithms, and will include them in the future release of PLANETALIGN.
>
> * For **dynamic/temporal network alignment**, to incorporate such datasets into PLANETALIGN, we plan to first introduce unified APIs (e.g., *PlanetAlign.data.TemporalData*) that support different evolving graph modalities, and add dynamic datasets[8, 9] into PLENATALIGN. We also plan to incorporate existing dynamic network alignment algorithms (e.g., DynaMAGNA++[8], DINGA[9], etc.) based on unified APIs provided by PLANETALIGN.
>
>
> ### **Question 1: How to ensure fairness when comparing algorithms that rely on different supervision levels / Hyperparamter budget**
>
> We thank the reviewer for raising this important question. We follow standard practice in the NA community[10, 11, 12] to ensure that all semi-supervised and unsupervised methods are compared under consistent and fair conditions, and we maintain a consistent hyperparameter budget for all baselines. Specifically,
>
> * For **unsupervised methods evaluated under semi-supervised settings**, we inject supervision by concatenating anchor-informed embeddings (e.g., one-hot embeddings) with the original node features. This approach incorporates supervision into unsupervised methods without modifying the original algorithmic structure[10, 11, 12].
> * For **semi-supervised methods evaluated under unsupervised settings**, we generate pseudo anchor links based on degree and node features similarity to emulate supervision[10, 11].
>
> For hyperparameter tuning, we maintain a consistent search budget of 5 configurations per key hyperparameter across all methods. Hyperparameters are tuned based on the default values and the hyperparameter study in the original papers of different NA methods. Detailed hyperparameter search space can be found in Appendix C in the revision.
>
>
> ### **Question 2: Additional results on scalability**
>
> We thank the reviewer for this constructive question and would like to refer the reviewer to our response to **Limitation 2** which addresses this point in detail.
>
> ### **Question 3: Extension to multi-network and dynamic settings**
>
> We thank the reviewer for this constructive question and would like to refer the reviewer to our response to **Limitation 4** which addresses this point in detail.
>
> ### **Question 4: Insights into the effectiveness of OT-based methods**
>
> We thank the reviewer for this constructive question and would like to refer the reviewer to our response to **Limitation 3** which addresses this point in detail.

---

> ### Author Response · Authors · 2025-11-20
> **Rebuttal by Authors 3/3**
>
> ### **References**
> [1] Scetbon, Meyer, and Marco Cuturi. "Low-rank optimal transport: Approximation, statistics and debiasing." Advances in Neural Information Processing Systems 35 (2022): 6802-6814.
>
> [2] Liu, Xinran, et al. "Expected sliced transport plans." arXiv preprint arXiv:2410.12176 (2024).
>
> [3] Chu, Xiaokai, et al. "Cross-network embedding for multi-network alignment." The world wide web conference. 2019.
>
> [4] Zeng, Zhichen, et al. "Hierarchical multi-marginal optimal transport for network alignment." Proceedings of the AAAI Conference on Artificial Intelligence. Vol. 38. No. 15. 2024.
>
> [5] De Domenico, Manlio, et al. "Identifying modular flows on multilayer networks reveals highly overlapping organization in interconnected systems." Physical Review X 5.1 (2015): 011027.
>
> [6] Omodei, Elisa, Manlio De Domenico, and Alex Arenas. "Characterizing interactions in online social networks during exceptional events." Frontiers in Physics 3 (2015): 59.
>
> [7] De Domenico, Manlio, et al. "Structural reducibility of multilayer networks." Nature communications 6.1 (2015): 6864.
>
> [8] Vijayan, Vipin, Dominic Critchlow, and Tijana Milenković. "Alignment of dynamic networks." Bioinformatics 33.14 (2017): i180-i189.
>
> [9] Yan, Yuchen, et al. "Dynamic knowledge graph alignment." Proceedings of the AAAI conference on artificial intelligence. Vol. 35. No. 5. 2021.
>
> [10] Zhang, Si, and Hanghang Tong. "Final: Fast attributed network alignment." Proceedings of the 22nd ACM SIGKDD international conference on knowledge discovery and data mining. 2016.
>
> [11] Zeng, Zhichen, et al. "Parrot: Position-aware regularized optimal transport for network alignment." Proceedings of the ACM web conference 2023. 2023.
>
> [12] Yu, Qi, et al. "Joint optimal transport and embedding for network alignment." Proceedings of the ACM on Web Conference 2025. 2025.

---

> ### Author Response · Authors · 2025-11-28
>
> Dear reviewer rJCh,
>
> Thank you again for your insightful comments!
>
> As the discussion period deadline approaches, we would like to respectfully inquire whether our responses address your concerns properly, and we are more than happy to address any remaining points.
>
> Many thanks for your time and effort!
>
> Sincerely,
>
> Authors

---

### Official Review · Reviewer_CB1K · 2025-10-27

**Soundness:** 3
**Presentation:** 3
**Contribution:** 3
**Rating:** 6
**Confidence:** 2

**Summary:**

This paper introduces PLANETALIGN, an open-source Python library aimed at providing a standardized platform for evaluating and developing network alignment (NA) methods. Network alignment is essential for identifying node correspondences across multiple networks, a task central to applications in various domains, including social network analysis, bioinformatics, and knowledge graph fusion. Despite the importance of this task, there has been a lack of a comprehensive benchmarking tool for NA methods. PLANETALIGN addresses this gap by providing a unified library that integrates 18 datasets from 6 domains, supports 14 NA methods, and offers a rich set of evaluation metrics. The library features easy-to-use APIs and a flexible design, allowing users to benchmark and develop new methods for network alignment. The authors claim that their library outperforms previous solutions in terms of speed and provides in-depth comparative studies that highlight the strengths and limitations of existing NA methods.

**Strengths:**

1. The paper offers a large collection of 18 datasets from diverse domains and 14 NA methods, providing a solid foundation for comparison across different scenarios.

2. PLANETALIGN emphasizes reproducibility by using standardized evaluation metrics (Hits@K, MRR) and consistent dataset splits. The inclusion of time and memory usage metrics makes it easier to assess the scalability and efficiency of algorithms.

3. PLANETALIGN has demonstrated significant improvements in execution time, with some methods being up to 3 times faster than their official counterparts.

**Weaknesses:**

1. Many datasets are synthetically generated or permuted from a single network, which may not fully capture the heterogeneity and noise of real-world multi-network environments.

2. The evaluation focuses heavily on quantitative metrics, but does not address the interpretability or explainability of the alignment results—a critical aspect in domains like bioinformatics or fraud detection.

3. The paper emphasizes the library’s ease of use and API design but provides no user study or external feedback to validate these claims.

**Questions:**

1. While PLANETALIGN supports a few OT-based methods, could you elaborate on potential future plans to incorporate more OT methods, especially those that handle non-convex optimizations better?

2. How do the methods in PLANETALIGN perform when applied to networks with millions of nodes and edges? Have you tested the library's performance on very large-scale graphs?

3. Could you discuss potential plans for updating the library, such as adding new NA methods, datasets, or advanced evaluation techniques in future releases?

---

> ### Author Response · Authors · 2025-11-20
> **Rebuttal by Authors 1/3**
>
> We thank the reviewer for the constructive feedback and comments, all of which are helpful to further improve our paper. In the following, we would like to provide point-to-point responses and clarifications.
>
> ### **Limitation 1: Synthetic datasets lack heterogeneity**
>
> We appreciate the reviewer’s thoughtful comment. While some synthetic homogeneous datasets are included for comprehensive benchmarking, we would like to clarify that the **majority (11 out of 18)** of the benchmarking datasets in PLANETALIGN are **real-world NA datasets**, spanning **all 6 domains**. Most of these real-world datasets already capture either **intra-network heterogeneity** (i.e., different node/edge types within a single network, such as DBP15K) or **inter-network heterogeneity** (i.e., different node/edge types across networks to be aligned, such as Phone-Email and Foursquare-Twitter). Therefore, existing built-in datasets in PLANETALIGN already cover **both homogeneous and heterogeneous** networks that capture real-world NA environments comprehensively.
>
> That said, we fully agree that adding synthetic heterogeneous networks will further improve the comprehensiveness of PLANETALIGN. In future releases, we plan to generate more heterogeneous datasets, such as text–image graphs, to further improve the diversity of datasets in our library.
>
> ### **Limitation 2: Lack of interpretability or explainability of alignment**
>
> We thank the reviewer for this valuable comment. We agree that interpretability and explainability are crucial for applying NA to downstream tasks. While PLANETALIGN is designed to be agnostic to specific downstream tasks, we have taken the following steps to improve the interpretability and explainability of the alignment results:
>
> * **Empirical and theoretical insights.** In the experiments section, we analyze NA performance spanning four key dimensions, including effectiveness, scalability, robustness, and sensitivity to supervision, discussing the **strengths, limitations, and trade-offs** of existing NA methods comprehensively from **both empirical and theoretical** views. These analyses naturally improve the interpretability and explainability of alignment results by revealing the **influence of different factors (e.g., network scale, anchors, node attributes, graph noises)**, and clarifying **why certain alignment behaviors emerge** under varying levels of **supervision levels and graph noises**.
>
> * **Visualization and monitoring tools.** PLANETALIGN provides built-in utility functions in *PlanetAlign.logger* and *PlanetAlign.utils* for **inspecting** the training process (e.g., loss and hits curves) and **visualizing** the alignment matrix (e.g., *PlanetAlign.utils.plot_alignment_matrix*). These tools help users to better understand the **convergence behavior, stability, and characteristics of the alignment**.
>
> * **Accessible intermediate outputs.** As shown in Figure 3 in the revision, PLANETALIGN also supports exporting intermediate results during alignment, such as **node embeddings,  alignment matrix, and alignment performance**. These outputs enable users to easily utilize various outputs during alignment into downstream tasks, facilitating interpretability and extensibility beyond benchmarking.
>
> ### **Limitation 3: No external feedback for usability**
>
> We thank the reviewer for this helpful comment. We acknowledge that PLANETALIGN is a relatively new library and is submitted under a double-blind policy, thus a formal user study has not yet been conducted.
>
> That said, the structure and API design of PLANETALIGN follow standard PyTorch and PyG conventions, emphasizing both usability and extensibility. We are already collecting feedback from collaborators within the research community and will continue to update and refine the library based on user feedback. As the number of users grows, we plan to conduct a formal user study to systematically guide future improvement of PLANETALIGN.

---

> ### Author Response · Authors · 2025-11-20
> **Rebuttal by Authors 2/3**
>
> ### **Question 1: Future plans to incorporate more OT methods**
>
> We thank the reviewer for this constructive question. We agree that incorporating more OT-based methods is an important future direction. To accelerate the development and integration of future OT-based methods, we plan to add the following utility modules to PLANETALIGN.
>
> * **General utility function for OT**, such as numerically stable Sinkhorn algorithm for solving entropy-regularized OT problem, a convex approximation for the original Kantorovich OT problem[1].
> * **Optimization tools for OT**, such as convex approximation and proximal point methods for solving complex OT problems (e.g., Fused Gromov-Wasserstein formulation), as well as low-rank and sliced OT[2,3] approximation for solving large-scale OT problems.
>
> While PLANETALIGN has already included the most recent published OT-based method to our knowledge (JOENA [4]), we will continue monitoring newly proposed OT-based approaches and integrate them into PLANETALIGN in a timely manner.
>
> ### **Question 2: Performance of PLANETALIGN on large graphs**
>
> We thank the reviewer for raising this thoughtful point. We would like to clarify that PLANETALIGN provides a comprehensive collection of the **most widely used** datasets in the NA community, which typically contain up to **20k** nodes per network. As NA is a **multi-network mining task** that requires identifying an alignment for every node across different networks, its computational complexity is inherently higher than that of single-network tasks such as node classification or link prediction.
>
> That said, we agree that testing PLANETALIGN's implementation on large-scale networks is important. To further validate the efficiency of PLANETALIGN, we compare the runtime between the official and PLANETALIGN's implementations on large-scale ER networks with **50K, 75K, and 100K** nodes, **which are much larger than commonly adopted datasets in the NA community**. The results are shown in the table below (Table 6 in the revision), showing that PLANETALIGN's implementations achieve up to **2.7** times speed-up even on large-scale networks. We can also see from Table 6 that most existing NA methods cannot scale to million-node networks due to their intrinsic complexity constraints. PLANETALIGN intentionally preserves the theoretical complexity of existing NA methods to ensure fair and consistent benchmarking.
>
> **Runtime (s) comparison with official implementations. OOM = Out-of-memory.**
>
> | Method      | 50K Official | 50K PLANETALIGN | 50K Speedup | 75K Official | 75K PLANETALIGN | 75K Speedup | 100K Official | 100K PLANETALIGN | 100K Speedup |
> |-------------|--------------|------------------|--------------|--------------|------------------|--------------|----------------|--------------------|---------------|
> | **REGAL**     | 334 | **214** | 1.56 | 583 | **390** | 1.49 | 1.12×10³ | **731** | 1.53 |
> | **CrossMNA**  | 4.12×10³ | **3.02×10³** | 1.36 | 7.36×10³ | **5.13×10³** | 1.43 | 1.37×10⁴ | **8.91×10³** | 1.54 |
> | **WAlign**    | 1.65×10³ | **1.29×10³** | 1.28 | 3.12×10³ | **2.54×10³** | 1.23 | OOM | OOM | OOM |
> | **BRIGHT**    | 2.82×10⁴ | **2.15×10⁴** | 1.31 | 8.95×10⁴ | **6.90×10⁴** | 1.29 | 3.47×10⁵ | **2.49×10⁵** | 1.40 |
> | **NeXtAlign** | 6.55×10⁴ | **6.08×10⁴** | 1.08 | 3.68×10⁵ | **3.41×10⁵** | 1.08 | OOM | OOM | OOM |
> | **WLAlign**   | 2.46×10⁵ | **9.12×10⁴** | 2.70 | OOM | OOM | OOM | OOM | OOM | OOM |
> | **SLOTAlign** | 3.01×10⁴ | **2.75×10⁴** | 1.10 | OOM | OOM | OOM | OOM | OOM | OOM |
> | **HOT**       | 3.14×10³ | **2.87×10³** | 1.10 | 6.17×10³ | **5.74×10³** | 1.07 | 1.08×10⁴ | **9.43×10³** | 1.15 |
> | **JOENA**     | 2.31×10³ | **2.28×10³** | 1.01 | 8.30×10⁴ | **8.12×10⁴** | 1.02 | OOM | OOM | OOM |
>
> In future releases, we plan to introduce more scalability modules within PLANETALIGN, including 1) distributed training APIs for embedding-based methods built upon PyTorch and PyG's relevant APIs, and 2) sparse and low-rank matrix techniques, as well as low-rank and sliced OT[1, 2] modules for consistency-based and OT-based methods which typically involve intensive matrix operations. These extensions will be provided as optional APIs, allowing NA algorithms built upon PLANETALIGN to handle large-scale networks with a few simple API calls.

---

> ### Author Response · Authors · 2025-11-20
> **Rebuttal by Authors 3/3**
>
> ### **Question 3: Plans for updating the library**
>
> We thank the reviewer for raising this important point. We are closely monitoring the NA community and will update new datasets, algorithms, and evaluation techniques upon release. Specifically,
>
> * For **NA datasets**, our immediate goal is to include multi-network (e.g., multi-layered version of ArXiv[5], Twitter[6], and SacchCere[7]) and dynamic networks (e.g., synthetic networks from [8, 9]). While our current API design naturally supports multi-network settings, we plan to add another API *PlanetAlign.data.TemporalData* to support integration of dynamic networks.
>
> * For **NA algorithms**, we plan to include multi-network alignment algorithms[10, 11], dynamic NA algorithms[8, 9], and domain-specific alignment algorithms such as entity alignment methods[12, 13]. Additionally, we plan to add more utility functions that facilitate the development and integration of future NA methods. Specifically, we plan to include general utility functions and optimization tools for consistency and OT-based methods. We also plan to include distributed training APIs and customizable ranking loss functions for embedding-based methods, potentially built upon existing APIs of PyTorch and PyG to ensure compatibility and maintain the usability of our library.
>
> * For **evaluation techniques**, in addition to effectivness, scalability, robustness, and sensitivity metrics to supervision, we plan to add uncertainty metrics of the alignment into PLANETALIGN, such as entropy-based or Bayesian measures[14], which are critical for developing active or self-improving NA methods highlighted as important future directions in our paper.
>
> We have added specific plans for updating PLANETALIGN in Section 7 of the revision.
>
> ### **References**
> [1] Peyré, Gabriel, and Marco Cuturi. "Computational optimal transport: With applications to data science." Foundations and Trends® in Machine Learning 11.5-6 (2019): 355-607.
>
> [2] Scetbon, Meyer, and Marco Cuturi. "Low-rank optimal transport: Approximation, statistics and debiasing." Advances in Neural Information Processing Systems 35 (2022): 6802-6814.
>
> [3] Liu, Xinran, et al. "Expected sliced transport plans." arXiv preprint arXiv:2410.12176 (2024).
>
> [4] Yu, Qi, et al. "Joint optimal transport and embedding for network alignment." Proceedings of the ACM on Web Conference 2025. 2025.
>
> [5] De Domenico, Manlio, et al. "Identifying modular flows on multilayer networks reveals highly overlapping organization in interconnected systems." Physical Review X 5.1 (2015): 011027.
>
> [6] Omodei, Elisa, Manlio De Domenico, and Alex Arenas. "Characterizing interactions in online social networks during exceptional events." Frontiers in Physics 3 (2015): 59.
>
> [7] De Domenico, Manlio, et al. "Structural reducibility of multilayer networks." Nature communications 6.1 (2015): 6864.
>
> [8] Vijayan, Vipin, Dominic Critchlow, and Tijana Milenković. "Alignment of dynamic networks." Bioinformatics 33.14 (2017): i180-i189.
>
> [9] Yan, Yuchen, et al. "Dynamic knowledge graph alignment." Proceedings of the AAAI conference on artificial intelligence. Vol. 35. No. 5. 2021.
>
> [10] Chu, Xiaokai, et al. "Cross-network embedding for multi-network alignment." The world wide web conference. 2019.
>
> [11] Zeng, Zhichen, et al. "Hierarchical multi-marginal optimal transport for network alignment." Proceedings of the AAAI Conference on Artificial Intelligence. Vol. 38. No. 15. 2024.
>
> [12] Liu, Xiaoze, et al. "Unsupervised entity alignment for temporal knowledge graphs." Proceedings of the ACM web conference 2023. 2023.
>
> [13] Chen, Zhuo, et al. "Meaformer: Multi-modal entity alignment transformer for meta modality hybrid." Proceedings of the 31st ACM international conference on multimedia. 2023.
>
> [14] Zhou, Qinghai, et al. "Attent: Active attributed network alignment." Proceedings of the Web Conference 2021. 2021.

---

> > ### Comment · Reviewer_CB1K · 2025-11-21
> >
> > Thanks for the rebuttal. I appreciate your work and have no further questions.

---

> > > ### Author Response · Authors · 2025-11-21
> > >
> > > Thank you for taking the time to review our rebuttal. We are glad to hear your positive view of our paper.

---

### Official Review · Reviewer_2QXK · 2025-11-01

**Soundness:** 4
**Presentation:** 4
**Contribution:** 3
**Rating:** 8
**Confidence:** 3

**Summary:**

PlanetAlign collects and organizes under clean and standardized APIs, datasets, methods and benchmarks for the network alignment problem: 18 datasets, 14 methods, standard evaluation metrics (Hits@k, Mean Reciprocal Rank (MRR)),  diverse benchmarking settings,  emphasizing effectiveness (e.g. supervision regimes), scalability (e.g. logging), robustness (e.g. controlled noise injection). PlanetAlign is put into a comprehensive test: empirically evaluating all methods over collected datasets (semi-supervised setting), reporting on metrics and drawing insights on what type of methods work better than others or their particular requirements).

**Strengths:**

- This is an excellently organized manuscript, describing the design and practical usage of a solid, software engineering work.

- The engineering contribution is substantial and noteworthy: links to related, mature artifacts (code repository, documentation) are conveniently provided. Abstractions identified are well thought-of and disciplined.

- It includes reports on extensive empirical studies: it would be hard to imagine comparing so "easily", so many and diverse methods for network alignment over multiple (and multiple-type) networks prior to this work. PlanetAlign makes a Herculean task, manageable: it certainly accelerates "network alignment" research. Figure 3 provides a compelling summary of the development simplicity made possible.

- Nicely reuses and integrates organization ideas and abstractions from other, highly-popular, open-source projects (e.g. pytorch-geometric).

- Impressive performance comparison agreement of PlanetAlign and "official" implementations of methods.

**Weaknesses:**

Relevance to ICLR audience is the major weakness of this effort:
- PlanetAlign is an (excellently engineered) collection of existing artifacts for network alignment methods, so  - although by "design" - the novelty dimension is limited, in particular for ML audience expecting emphasis on deep/representation learning (although there is an embeddings-related component part).
- This work would be potentially a best fit for a venue focusing on networks.

**Questions:**

- Are there provisions/facilities for the user community to easily "extend" PlanetAlign (i.e. integrated methods/datasets into it) - rather than the authors expanding it as in Appendix E?
- Any thoughts for a kind of leaderboard for the network alignment community?

---

> ### Author Response · Authors · 2025-11-20
> **Rebuttal by Authors 1/1**
>
> We thank the reviewer for the constructive feedback and comments, all of which are helpful to further improve our paper. In the following, we would like to provide point-to-point responses and clarifications.
>
> ### **Limitation 1: Limited relevance ICLR's audience**
>
> We thank the reviewer for this thoughtful comment. Firstly, we would like to respectfully clarify that our paper was submitted to the "Datasets and Benchmarks" track of ICLR, which emphasizes contributions in terms of high-quality datasets, comprehensive evaluations, and mature benchmarking infrastructures, rather than novel algorithmic methodology.
>
> Secondly, PLANETALIGN is also highly relevant to the ICLR audience. Specifically, embedding-based NA methods directly build on **graph representation learning**, which is a central topic at ICLR. OT-based methods explore applications of **optimal transport**, which is actually listed explicitly among the relevant topics at the [homepage](https://iclr.cc) of ICLR. The NA problem itself also falls into the broader area of **"learning on graphs"**, which is also listed as a relevant topic on the ICLR [call for papers](https://iclr.cc/Conferences/2026/CallForPapers).
>
>
> ### **Question 1: Guidelines for extending PLANETALIGN**
>
> We thank the reviewer for the constructive question. Yes, PLANETALIGN was explicitly designed with extensibility and can serve as a powerful research toolkit for NA even beyond benchmarking. Specifically, PLANETALIGN features:
>
> * **Extensible and easy-to-use APIs for customized datasets and algorithms**. As described in Section 4.2 of our paper, PLANETALIGN provides unified base classes *PlanetAlign.data.BaseData* and *PlanetAlign.algorithm.BaseModel*, which allows users to customize their own NA datasets and algorithms while remaining fully compatible with PLANETALIGN's APIs to streamline fair and reproducible evaluation.
>
> * **Comprehensive documentation and tutorials**. We provide detailed documentation and dedicated tutorials at https://planetalign.netlify.app. The "[Working with NA Datasets](https://planetalign.netlify.app/tutorials/datasets)" and "[Working with NA Algorithms](https://planetalign.netlify.app/tutorials/algorithms)" parts of the tutorial explicitly include sections guiding users to build customized datasets and algorithms on top of PLANETALIGN's APIs.
>
> ### **Question 2: Leaderboard for NA**
>
> We thank the reviewer for this excellent suggestion. We fully agree that such a platform would greatly benefit the NA community.
>
> Following this suggestion, we plan to develop a public leaderboard as part of future PLANETALIGN releases. Specifically, the leaderboard will host standardized datasets, evaluation metrics, and testing scripts from PLANETALIGN to ensure consistent and reproducible comparison. The platform will allow researchers to submit their model outputs for automatic evaluation across multiple tracks (e.g., semi-supervised, unsupervised, etc.), and provide comprehensive evaluation across effectiveness, runtime/memory usage, robustness, etc.

---

> ### Author Response · Authors · 2025-11-28
>
> Dear reviewer 2QXK,
>
> Thank you again for your insightful comments!
>
> As the discussion period deadline approaches, we would like to respectfully inquire whether our responses address your concerns properly, and we are more than happy to address any remaining points.
>
> Many thanks for your time and effort!
>
> Sincerely,
>
> Authors

---

### Author Response · Authors · 2025-12-03
**Rebuttal Summary to AC**

Dear AC,

Thank you for overseeing the review process. We would like to briefly summarize the main contributions of our work and our effort in addressing the concerns & questions of reviewers.

We introduce a comprehensive Python library for network alignment (NA) that features a rich collection of built-in datasets, methods, and evaluation pipelines with easy-to-use APIs, facilitating standardized benchmarking and algorithmic development in the field. Through extensive benchmarking studies, we reveal valuable insights into the strengths and limitations of existing NA methods. **Across the reviews, there are explicit, broad, and consistent recognitions of the strengths of our work, including**,

1) **Comprehensive coverage of datasets, algorithms, and evaluation dimensions**:
    * **2QXK** - "extensive empirical studies: it would be hard to imagine comparing so easily."
    * **CB1K** - "offers a large collection of 18 datasets from diverse domains and 14 NA methods, providing a solid foundation for comparison across different scenarios."
    * **rJCh** - "Comprehensive benchmark coverage; Systematic multi-dimensional evaluation"
    * **67gk** - "This new python library integrates a wide range of datasets and algorithms."
2) **Strong engineering quality and reproducibility**:
    * **2QXK** - "Abstractions identified are well thought-of and disciplined. Nicely reuses & integrates organization ideas and abstractions from highly-popular projects"
    * **CB1K** - "PLANETALIGN emphasizes reproducibility by using standardized evaluation metrics and consistent dataset splits."
    * **rJCh** - "provides a clean, modular PyTorch-style API, reproducible experiment settings."
    * **67gk** - "It provides a consistent API and unified evaluation pipeline, which makes the method comparisons easier, more consistent and more reliable"
3) **Highly optimized and efficient implementations**:
    * **2QXK** - "Impressive performance comparison of PlanetAlign and official implementations of methods."
    * **CB1K** - "PLANETALIGN has demonstrated significant improvements in execution time."
    * **rJCh** - "optimized code achieving up to 3× faster runtime"
    * **67gk** - "achieves similar accuracy with 3x faster on efficiency."
4) **High community impact through a well-documented library**:
    * **2QXK** - "PlanetAlign makes a Herculean task, it certainly accelerates network alignment research; links to related, mature artifacts (code repository, documentation) are conveniently provided"
    * **rJCh** - "By releasing a well-documented, open-source library, the work provides an infrastructure that can serve as a long-term standard for the NA community."
    * **67gk** - "The documentation is clear and easy to follow."


Across all reviewers, we received constructive questions and concerns regarding (1) extensibility of our library; (2) plans for future maintenance & updates; (3) scalability to large-scale networks; (4) concerns for algorithmic novelty and relevance to ICLR; (5) concerns for interpretability and theothcial analysis; (6) inclusion of additional graph modalities; (7) additional experimental details; (8) formal community feedback. **In response, we provided point-by-point clarifications and added new analyses and experiments, including**:

1) Clarification about our **extensible and modular API design**, as well as **comprehensive documentation and tutorials**, all of which demonstrate high customizability and extensibility of our library to benefit NA research **beyond benchmarking**.
2) **Detailed and actionable future plans** for extending our library to include new datasets, algorithms, benchmark settings, and utility functions.
3) Justification for the scale of networks we adopt and **additional experiments on larger-scale networks**.
4) Clarification about the intended scope of our submission and relevance to ICLR's audience.
5) Clarification about the **inclusion of utility functions** in our library that **facilitate interpretability** of alignment, as well as **additional theoretical insights** into OT-based methods.
6) Explanation that our library **already captures various graph modalities**, as well as **detailed future plans** for adding more graph modalities.
7) **More detailed description** of the hyperparameter budget and setup.
8) Clarification about the recency of our library and the double-blind policy of ICLR, as well as our plan for building a NA leaderboard.

Unfortunately, most of the reviewers had not yet responded before the unfortunate incident happened, except for reviewer CB1K, who expressed satisfaction with our rebuttal. Nevertheless, we are confident that our comprehensive rebuttal addresses most, if not all, of their concerns.

**We hope this summary, which is based on objective information from the reviews and rebuttal, helps facilitate your decision. Once again, we thank you and all reviewers for your engagement and thoughtful evaluations.**

Best Regards,

The Authors

---

### Meta-Review · Area_Chair_rkFd · 2026-01-04

**Summary:**

The paper presents PLANETALIGN, an open-source Python library for network alignment (NA). It aggregates 18 datasets and 14 representative NA methods, exposes clean PyTorch-style APIs for development of NA methods, and enforces reproducible evaluation with metrics like Hits@K and MRR under multiple regimes (effectiveness, scalability, robustness, supervision sensitivity). The authors conduct extensive experiments empirically evaluating all methods over collected datasets. Compare with the existing algorithms, the new framework achieves similar accuracy with 3x faster on efficiency.

The authors resolved the key concerns from reviewers, and the rest are not critical.

**Reviewer Concerns:**

The concerns from 2QXK (Relevance to ICLR) were resolved. The paper complies with the datasets and benchmarks track of ICLR.

The concern (Datasets may not capture the heterogeneity and noise of real-world multi-network environments) from CB1K was resovled.  The remaining concerns about lack of interpretability in alignment results and absence of user studies are not critical.

The concern (The paper lacks deeper theoretical analysis for why OT-based approaches perform better) from rJCh was resolved. The concern (The framework should be extended to support multi-network and dynamic alignment scenarios.) was partially resolved (PLANETALIGN supports multi-network scenarios and implements CrossMNA and HOT) and the support of dynamic alignment scenarios is a future work. The concern of limited algorithmic novelty is not critical, as the paper is for datasets and benchmarks track. The concern of incomplete scalability validation is also not a critical concern, as most of NA methods can not scale to graphs > 100k nodes. But this can be an important future work.

The reviewer 67gk's concern about lack of extensibility for datasets, functions, and customized use cases was resolved and confirmed by the reviewer.

**Reviewer Scores:**

Three reviewers gave positive scores (two 6s and one 8). One reviewer gave a negative score of 4, but would like to raise the score.

---

### Decision · Program_Chairs · 2026-01-26

Accept (Poster)